# A somato-cognitive action network alternates with effector regions in motor cortex

Evan M. Gordon[1]✉, Roselyne J. Chauvin[2], Andrew N. Van[2,3], Aishwarya Rajesh[1], Ashley Nielsen[2], Dillan J. Newbold[2,4], Charles J. Lynch[5], Nicole A. Seider[2,6], Samuel R. Krimmel[2], Kristen M. Scheidter[2], Julia Monk[2], Ryland L. Miller[2,6], Athanasia Metoki[2], David F. Montez[2], Annie Zheng[2], Immanuel Elbau[5], Thomas Madison[7], Tomoyuki Nishino[6], Michael J. Myers[6], Sydney Kaplan[2], Carolina Badke D'Andrea[1,6,8], Damion V. Demeter[8], Matthew Feigelis[8], Julian S. B. Ramirez[9], Ting Xu[9], Deanna M. Barch[1,6,10], Christopher D. Smyser[1,2,11], Cynthia E. Rogers[5,11], Jan Zimmermann[12], Kelly N. Botteron[6], John R. Pruett[6], Jon T. Willie[2,5,13], Peter Brunner[3,13], Joshua S. Shimony[1], Benjamin P. Kay[2], Scott Marek[1], Scott A. Norris[1,2], Caterina Gratton[14], Chad M. Sylvester[6], Jonathan D. Power[5], Conor Liston[5], Deanna J. Greene[8], Jarod L. Roland[13], Steven E. Petersen[1,2,3,10,15], Marcus E. Raichle[1,2,3,10,15], Timothy O. Laumann[6], Damien A. Fair[7,16,17] & Nico U. F. Dosenbach[1,2,3,10,11,18]✉

Motor cortex (M1) has been thought to form a continuous somatotopic homunculus extending down the precentral gyrus from foot to face representations[1,2], despite evidence for concentric functional zones[3] and maps of complex actions[4]. Here, using precision functional magnetic resonance imaging (fMRI) methods, we find that the classic homunculus is interrupted by regions with distinct connectivity, structure and function, alternating with effector-specific (foot, hand and mouth) areas. These inter-effector regions exhibit decreased cortical thickness and strong functional connectivity to each other, as well as to the cingulo-opercular network (CON), critical for action[5] and physiological control[6], arousal[7], errors[8] and pain[9]. This interdigitation of action control-linked and motor effector regions was verified in the three largest fMRI datasets. Macaque and pediatric (newborn, infant and child) precision fMRI suggested cross-species homologues and developmental precursors of the inter-effector system. A battery of motor and action fMRI tasks documented concentric effector somatotopies, separated by the CON-linked inter-effector regions. The inter-effectors lacked movement specificity and co-activated during action planning (coordination of hands and feet) and axial body movement (such as of the abdomen or eyebrows). These results, together with previous studies demonstrating stimulation-evoked complex actions[4] and connectivity to internal organs[10] such as the adrenal medulla, suggest that M1 is punctuated by a system for whole-body action planning, the somato-cognitive action network (SCAN). In M1, two parallel systems intertwine, forming an integrate–isolate pattern: effector-specific regions (foot, hand and mouth) for isolating fine motor control and the SCAN for integrating goals, physiology and body movement.

Beginning in the 1930s, Penfield and colleagues mapped human M1 with direct cortical stimulation, eliciting movements from about half of sites, mostly of the foot, hand and mouth[1]. Although representations for specific body parts overlapped substantially[11], these maps gave rise to the textbook view of M1 organization as a continuous homunculus, from head to toe.

In non-human primates, organizational features inconsistent with the motor homunculus have been described. Structural connectivity studies divided M1 into anterior, gross motor, 'old' M1 (few direct projections to spinal motor neurons) and posterior, fine motor, 'new' M1[12,13] (many direct motoneuronal projections). Non-human primate stimulation studies showed the body to be represented in anterior M1[14], and the motor effectors (tail, foot, hand and mouth) in posterior M1. Such studies also suggested that the limbs are represented in concentric functional zones progressing from the digits at the centre, to the shoulders on the periphery[3]. Moreover, stimulations could elicit increasingly complex and multi-effector actions when moving from posterior to anterior M1[4].

During natural behaviour, voluntary movements are part of goal-directed actions, initiated and controlled by executive regions in the CON[5]. Neural activity preceding voluntary movements can first be detected in the rostral cingulate zone[15] within dorsal anterior cingulate cortex (dACC), then in the pre-supplementary motor area (pre-SMA) and supplementary motor area[16] (SMA), followed by M1. These regions all project to the spinal cord[17], with M1 as the main transmitter of motor commands down the corticospinal tract[18]. Efferent motor copies are received by primary somatosensory cortex[19] (S1), cerebellum[20] and striatum[21] for online correction, learning[20] and inhibition of competing movements[22]. Tracer injections in non-human primates demonstrated projections from anterior M1/CON to internal organs (such as adrenal medulla) for preparatory sympathetic arousal in anticipation of action[10]. Post-movement error and pain signals are relayed primarily to insular and cingulate regions of the CON, which update future action plans[8,9].

Resting-state functional connectivity (RSFC) fMRI noninvasively maps the brain's functional networks[23]. Precision functional mapping (PFM) studies rely on large amounts of multi-modal data (such as RSFC and tasks) to map individual-specific brain organization in the greatest possible detail[24,25]. Early PFM studies identified separate foot, hand and mouth M1 regions[24] with their respective cerebellar and striatal targets[26,27]. These foot, hand and mouth motor circuits were characterized by strong within-circuit connectivity and effector specificity in task fMRI[24], consistent with myeloarchitectonic evidence for distinct cortical fields[28]. However, these circuits were relatively isolated and did not include functional connections with control networks such as CON that could support the integration of movement with global behavioural goals. A recent study showed that prolonged dominant arm immobilization strengthened functional connectivity between disused M1 and the CON[29,30], suggesting that the role of CON may extend beyond abstract action control and into movement coordination.

Here we used the latest iteration of PFM with higher resolution (2.4 mm) and greater amounts of fMRI (RSFC: 172–1,813 min per participant; task: 353 min per participant), and diffusion data, to map M1 and its connections with the highest detail. The results were verified in group-averaged data from the three largest fMRI studies (Human Connectome Project (HCP), Adolescent Brain Cognitive Development (ABCD) study, UK Biobank (UKB); total $n$ of approximately 50,000). Furthermore, we placed our findings in cross-species (macaque versus human), developmental (neonate, infant, child and adult) and clinical (perinatal stroke) contexts using PFM data.

## Two networks alternate in motor cortex

Advanced PFM revealed connectivity that differed markedly from the canonical homuncular organization of M1. Two contrasting patterns of functional connectivity alternated in M1 (Fig. 1a and Supplementary Video 1). The expected pattern, as previously described for M1 foot, hand and mouth representations[24,31], comprised three regions (per hemisphere) for which cortical connectivity was restricted to homotopic contralateral M1 and adjacent S1 (Fig. 1a, seeds 1, 3 and 5). This set of RSFC-defined regions corresponded with task-evoked activity during foot, hand and tongue movements (Fig. 1b; see Extended Data Fig. 1d for other participants).

Interleaved between the known foot, hand and mouth M1 regions lay three areas that were strongly functionally connected to each other, both contralaterally and ipsilaterally, forming a previously unrecognized interdigitated chain down the precentral gyrus (Fig. 1a, seeds 2, 4, 6). The motif of three M1 inter-effector regions was observed in every highly sampled adult (Extended Data Fig. 1a and Supplementary Table 1) and replicated within-individual in separate data from the same participants (Extended Data Fig. 1b). Of note, the inter-effector pattern was also evident in all large $n$ group-averaged data (UKB ($n = 4,000$), ABCD ($n = 3,928$), HCP ($n = 812$) and WU120 ($n = 120$); Extended Data

Fig. 1c). The M1 inter-effector functional connectivity motif was most apparent in individual-specific maps, but once recognized, was also clearly identifiable in group-averaged data when visualized using stringent connectivity thresholds (Fig. 1c).

The inter-effector regions were evident relatively early in development. Whereas PFM data from a human newborn did not reveal the inter-effector motif, it was detectable in an 11-month-old infant, and was almost adult-like in a 9-year-old child (Extended Data Fig. 2a–e). Inter-effector regions could even be identified in an individual with preserved motor function despite suffering severe bilateral perinatal strokes that destroyed large portions of M1 (Extended Data Fig. 2f; see ref. 32 for clinical details).

## Inter-effectors link to control network

In addition to being interconnected, the three inter-effector regions were functionally connected to multiple regions of the CON, thought to be important for goal-oriented cognitive control. Connectivity was very strong with SMA and a region in dACC[15] (caudal cingulate zone) (Fig. 2a; see Extended Data Fig. 3 and Supplementary Table 1 for all participants) but was also evident with anterior prefrontal cortex (aPFC) and insula (Supplementary Fig. 1). In striatum, inter-effector regions were most strongly connected to dorsolateral putamen. In thalamus, connectivity peaked in the centromedian (CM) nucleus, with additional strong connectivity observed in ventral intermediate (VIM), ventral posteriomedial (VPM) and ventral posterior inferior (VPI) nuclei. Inter-effector regions were strongly connected to cerebellar areas (Fig. 2a) surrounding but distinct from effector-specific cerebellar regions (Extended Data Fig. 1e).

In all highly sampled individuals ($n = 7$), the inter-effector regions had stronger connections to CON than did any of the foot, hand or mouth regions (Fig. 2b; Supplementary Fig. 2a for all participants); across participants: all two-tailed paired $t > 4.75$, $P < 0.01$ false discovery rate (FDR) corrected, for inter-effector versus foot, versus hand, and versus mouth (Extended Data Fig. 4a). The inter-effector versus foot, hand and mouth difference was larger for CON than for any of the other 10 networks (all two-tailed paired $t > 3.5$; all $P < 0.05$, FDR-corrected; Fig. 2b). In network space, inter-effector regions were positioned between CON and the foot, hand and mouth regions (Fig. 2c; Supplementary Fig. 2b for all participants). Inter-effector regions were also more strongly connected to: middle insula, known to process pain[9] and interoceptive signals[33] (Extended Data Fig. 4b; all two-tailed paired $t > 2.7$; all $P < 0.05$, FDR-corrected); lateral cerebellar lobule V and vermis Crus II, lobule VIIb and lobule VIIIa (all two-tailed paired $t > 3.7$, all $P < 0.05$, FDR-corrected; Extended Data Fig. 4c); dorsolateral putamen, critical for motor function (all two-tailed paired $t > 3.7$; all $P < 0.01$, FDR-corrected, Extended Data Fig. 4d); and sensory-motor regions of thalamus (VIM, CM and VPM; all two-tailed paired $t > 3.0$, all $P < 0.03$, FDR-corrected; Extended Data Fig. 4e–g). Searching for differences between the three inter-effector regions revealed that the middle inter-effector region consistently exhibited stronger functional connectivity to extrastriate visual cortex than did the superior and inferior inter-effector regions (Extended Data Fig. 5; Supplementary Fig. 3 for all participants).

Comparing the relative timing of resting-state fMRI signals (lag structure[34]) showed that infra-slow (<0.1 Hz) fMRI signals in both the CON and the inter-effector network lagged behind those in effector-specific regions (Fig. 2d; CON versus foot: two-tailed paired $t = 2.38$, $P = 0.055$, uncorrected; versus hand and mouth: all two-tailed paired $t > 2.84$, all $P < 0.03$, uncorrected; inter-effector versus foot, hand and mouth: all two-tailed paired $t > 2.5$, all $P < 0.05$, uncorrected). Inter-regional lags in infra-slow (<0.1 Hz) signals are associated with propagation of higher-frequency delta activity (0.5–4 Hz) in the opposite direction[35], suggesting that high-frequency signals may occur earlier in CON than in M1—consistent with electrical recordings during voluntary

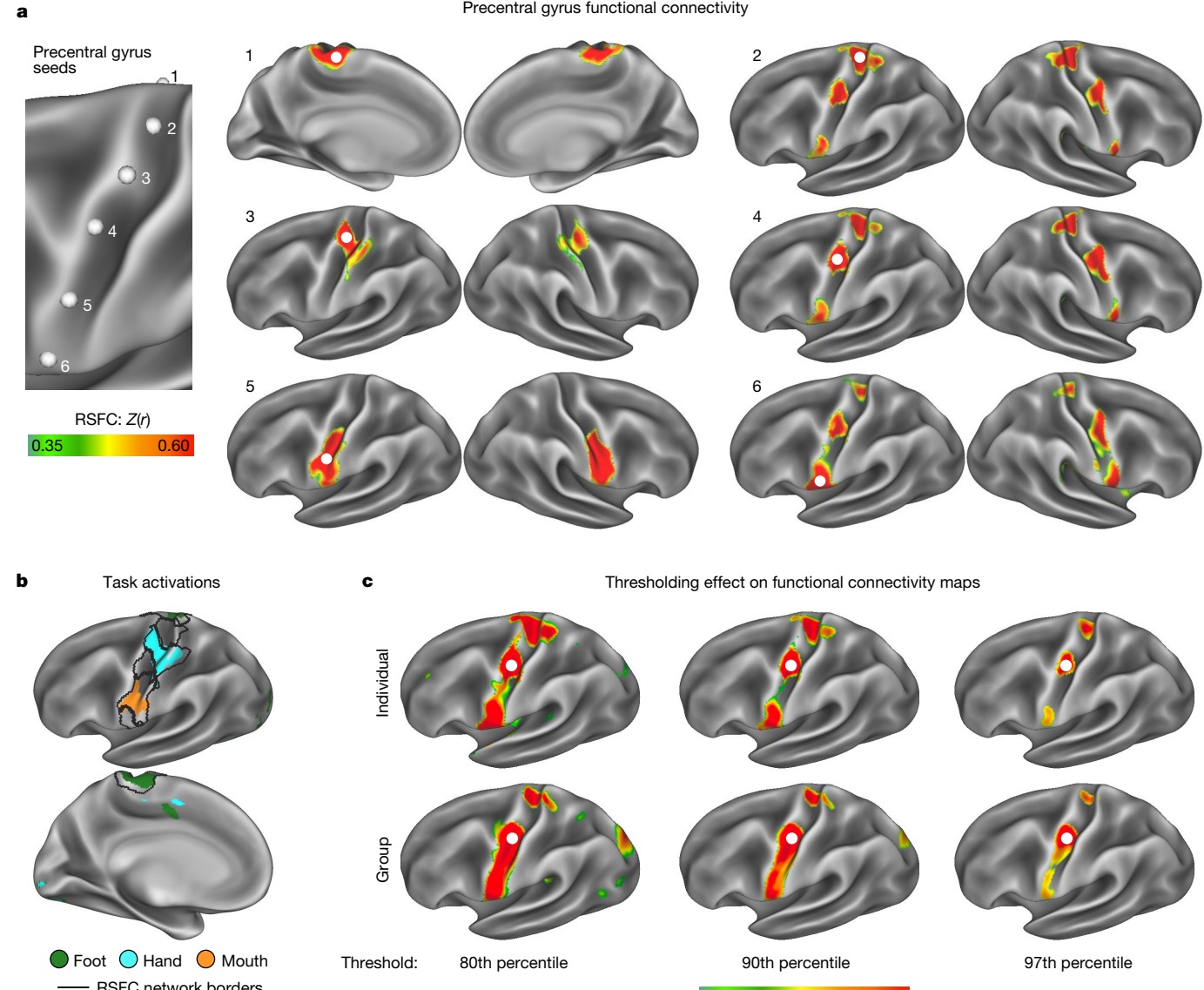

**a** Precentral gyrus functional connectivity

Precentral gyrus seeds

RSFC: Z(r)
0.35 — 0.60

**b** Task activations

Foot  Hand  Mouth
—— RSFC network borders

**c** Thresholding effect on functional connectivity maps

Individual

Group

Threshold: 80th percentile     90th percentile     97th percentile

− RSFC: Z(r) +

**Fig. 1 | Precision functional mapping of primary motor cortex. a**, RSFC seeded from a continuous line of cortical locations in the left precentral gyrus in a single exemplar participant (P1; 356 min resting-state fMRI). The six exemplar seeds shown represent all distinct connectivity patterns observed (see Supplementary Video 1 for complete mapping). Functional connectivity seeded from these locations illustrated classical M1 connectivity of regions representing the foot (1), hand (3) and mouth (5), as well as an interdigitated set of strongly interconnected regions (2, 4 and 6). See Extended Data Fig. 1a and Supplementary Video 2 for all highly sampled participants, Extended Data Fig. 1b for within-participant replications, and Extended Data Fig. 1c for group-averaged data. **b**, Discrete functional networks were demarcated using a whole-brain, data-driven, hierarchical approach (Methods) applied to the resting-state fMRI data, which defined the spatial extent of the networks observed in Fig. 1 (black outlines). Regions defined by RSFC were functionally labelled using a classic block-design fMRI motor task involving separate movement of the foot, hand and tongue (following ref. 31; see ref. 29 for details). The map illustrates the top 1% of vertices activated by movement of the foot, hand and mouth in the exemplar participant (P1; see Extended Data Fig. 1d for other participants). **c**, The inter-effector connectivity pattern became more distinct from surrounding effector-specific motor regions as connectivity thresholding increased from the 80th to the 97th percentile. RSFC thresholds required to detect the inter-effector pattern were lower in individual-specific data (top) than in group-averaged data (bottom; ABCD study, n = 3,928).

movement[36]—but that such signals reach the inter-effectors earlier than the foot, hand and mouth regions.

As expected, the M1 foot, hand and mouth regions were strongly functionally connected with adjacent S1 (Fig. 1a and Extended Data Fig. 6a), consistent with known functional connections between M1 and S1[37]. By contrast, inter-effector regions exhibited lower connectivity with adjacent S1 (Extended Data Fig. 4h; all two-tailed paired $t > 3.2$, all $P < 0.02$, FDR-corrected). More specifically, inter-effector functional connectivity extended into the fundus of the central sulcus (Extended Data Fig. 6b; Brodmann area (BA) 3a), which represents proprioception[38], but not to the postcentral gyrus (BA1, BA2 and BA3b), representing cutaneous tactile stimuli.

Convergent with these functional differences, metrics of brain structure systematically differed between inter-effector and effector-specific regions. Inter-effector regions exhibited lower cortical thickness (all two-tailed paired $t > 3.6$; all $P \le 0.01$, FDR-corrected; Fig. 2e), more similar to prefrontal cortex[39], but higher fractional anisotropy (2 mm beneath cortex; all two-tailed paired $t > 5.3$; all $P < 0.05$, FDR-corrected; Extended Data Fig. 4j). Intracortical myelin content was higher in inter-effector regions than in foot regions (two-tailed

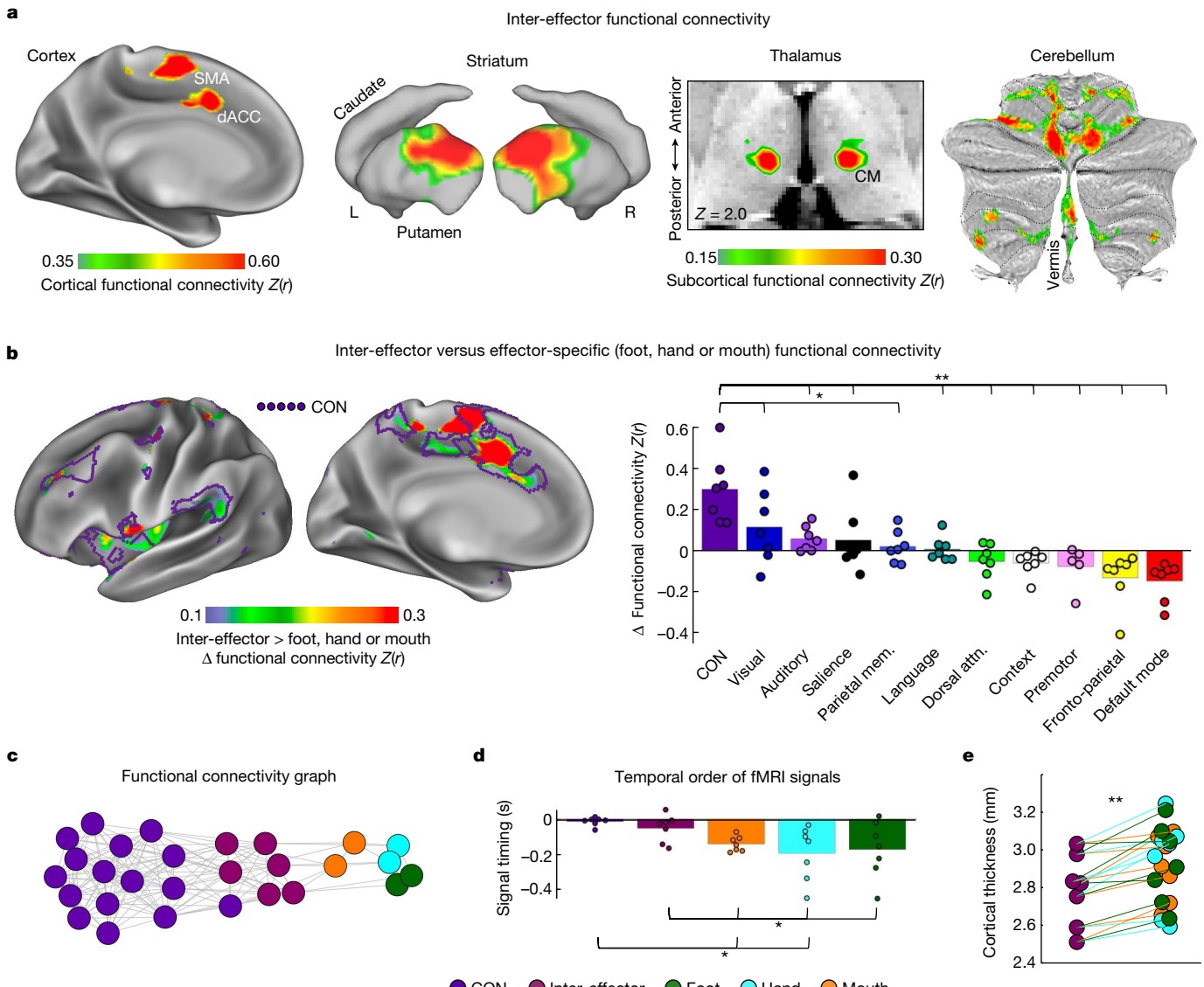

**Fig. 2 | Functional connectivity and cortical thickness of the M1 inter-effector motif. a**, Brain regions with the strongest functional connectivity to the left middle inter-effector region (exemplar seed) in cortex, striatum, thalamus (horizontal slice; CM nucleus) and cerebellum (flat map) in the exemplar participant (P1). See Extended Data Fig. 3 for other participants. **b**, Left, brain regions more strongly functionally connected to inter-effectors than to any foot, hand or mouth regions (P1; Supplementary Fig. 2a for other participants). Purple outlines show the CON (individual-specific). Central sulcus is masked as it exhibits large differences by definition. Right, connectivity was calculated between every network and both the inter-effector and effector-specific M1 regions. The plot shows the smallest difference between inter-effector and any effector-specific connectivity, averaged across participants. This difference was larger for CON than for any other network (two-tailed paired *t*-tests, *$P < 0.05$, FDR-corrected; **$P < 0.01$, FDR-corrected). Coloured circles represent individual participants. **c**, Inter-network relationships visualized in network space using a spring-embedding plot, in which connected regions are pulled together and disconnected regions are pushed apart. Connecting lines indicate a functional connection ($Z(r) > 0.2$) (P1; see Supplementary Fig. 2b for all participants). **d**, Inter-effector and effector-specific regions were tested for systematic differences in the temporal ordering of their infra-slow fMRI signals[34] (<0.1 Hz). The plot shows signal ordering in CON, inter-effector and effector-specific regions, averaged across participants (standard error bars; two-tailed paired *t*-test *$P < 0.05$, uncorrected). Coloured circles represent individual participants. Prior electrophysiology work suggests that later infra-slow activity (here, CON) corresponds to earlier delta-band (0.5–4 Hz) activity[35]. **e**, In each participant (filled circles), inter-effector regions exhibited lower cortical thickness than all effector-specific regions (two-tailed paired *t*-test **$P \le 0.01$, FDR-corrected). Attn., attention; mem., memory.

paired $t = 6.8$, $P < 0.005$, FDR-corrected) but lower than in hand regions (two-tailed paired $t = 4.8$, $P < 0.005$, FDR-corrected; Extended Data Fig. 4k), suggesting myeloarchitectonic differences similar to those described in ref. 28.

## Concentric motor and body-action zones

To better understand the functions of the inter-effector motif, we collected fMRI data during blocked performance of 25 different

movements in 2 highly sampled individuals (64 runs; 244 min per participant) and during a novel event-related task with separate planning and execution phases for coordinated hand and foot movements (12 runs; 132 min per participant). According to the homuncular model of M1, activation when moving a given body part should exhibit a single peak within the precentral gyrus. If M1 is instead organized into concentric functional zones, all movements except those at the centres (that is, toes, fingers, tongue) should exhibit two peaks (above and below). Within each of the three effector-specific regions, the topography of

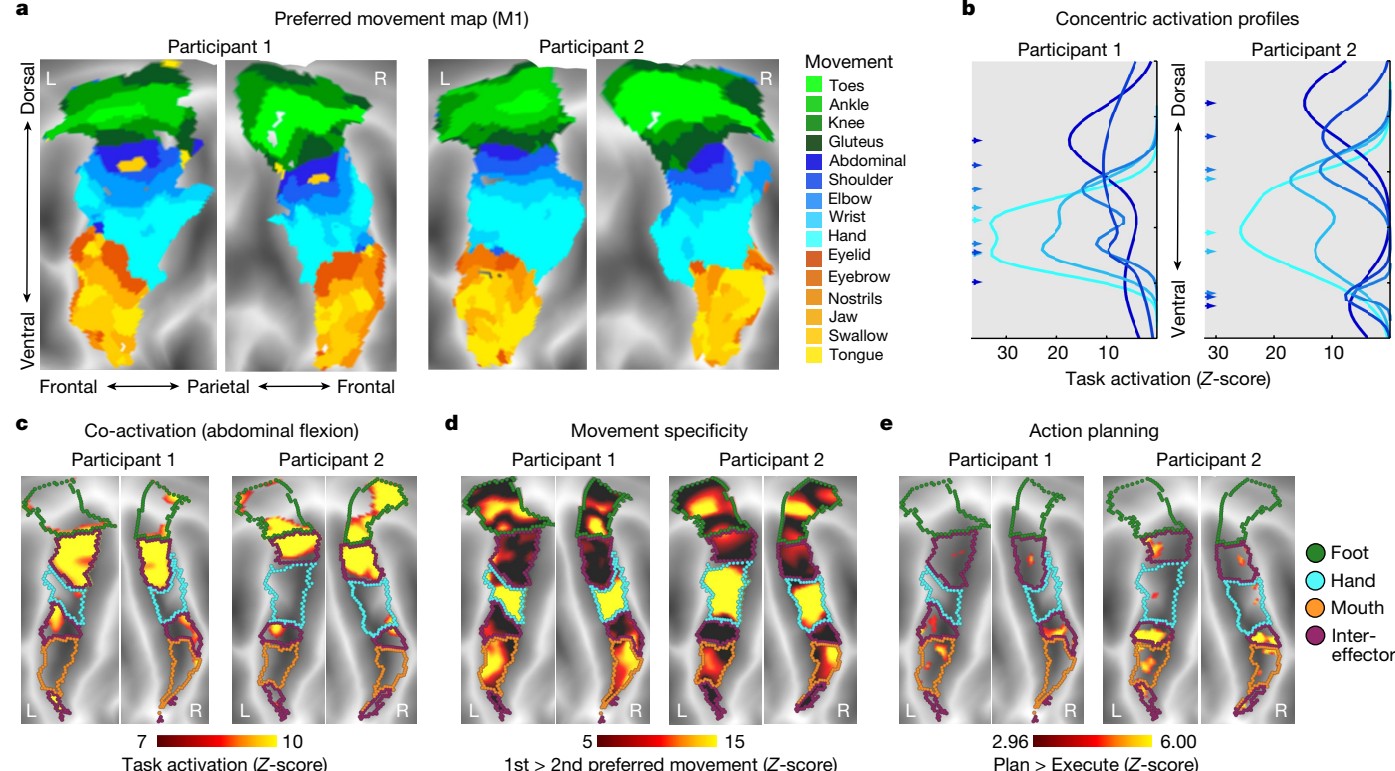

**Fig. 3 | Individual-specific task activations in M1. a**, Task fMRI activations (P1 and P2) during a movement task battery, including movement of the toes, ankles, knees, gluteus, abdominals, shoulders, elbows, hands, eyebrows, eyelids, tongue and swallowing (244 min per participant). Each cortical vertex is coloured according to the movement that elicited the strongest task activation (winner takes all) and is shown on a flattened representation of the cortical surface. Background shading indicates sulcal depths. **b**, Activation strength for each movement was computed along the dorsal–ventral axis within M1. A two-peak Gaussian curve was fitted to each movement activation (Methods). Fitted curves are shown for movement of abdominals, shoulder, elbow, wrist and hand. Peak locations (arrows on left) were arranged concentrically around the hand peak. See Extended Data Fig. 7 and Supplementary Fig. 4 for all movements. **c**, Inter-effector regions were co-activated during abdominal contraction. **d**, Inter-effector regions exhibited more generalized evoked activity during movements. Movement specificity was computed as the activation difference between the first- and second-most preferred movements for the six conditions that most activated each discrete region (toes, abdominal, hand, eyelid, tongue and swallowing). **e**, Event-related task fMRI data during an action planning task with separate planning and execution phases for movements of the hands and feet (Methods). M1 activity in the planning phase was higher than in the execution phase in the inter-effector but not the effector-specific regions.

preferred movements—the movement eliciting greatest activation in each vertex (Fig. 3a)—was more consistent with a concentric organization (distal-proximal; for example, toes in the centre, with surrounding concentric zones of ankle–knee–hip)[3] than with the canonical, linear toes-to-face homuncular model[1].

To formally test for a concentric organization, we fit one- and two-peak Gaussian curves to the task activation profiles along the dorsomedial-to-ventrolateral axis of M1. Two-peak curve fits were significantly better for all movements (*F*-test for comparing models (Methods): all $F > 6.9$, all $P < 0.001$, FDR-corrected) except hand in P2 ($F \cong 0, P \cong 1$) (Supplementary Fig. 4). The curve fits revealed concentric activation zones centred around activation peaks for distal movements (hand (Fig. 3b), toes and tongue (Supplementary Fig. 4)) and expanding outward to more proximal movements (shoulder, gluteus and jaw). Concentric rings of activation from separate foot, hand and mouth centres intersected in the superior and middle inter-effector regions (Extended Data Fig. 7).

Some movements requiring less fine motor control, such as isometric contraction of the abdominals (Fig. 3c) or raising the eyebrow co-activated multiple inter-effector regions and the CON (Extended Data Figs. 7 and 8a,b,e). By contrast, movements of the foot and hand only activated the corresponding effector-specific regions (Extended Data Figs. 7 and 8c–e). Unlike effector-specific regions, the inter-effectors exhibited weak movement specificity, with minimal activation differences between their preferred and non-preferred movements (Fig. 3d) and at least some activation observed across most movements (Extended Data Fig. 7).

To verify that inter-effector function is not specific to vocalization[40], we also collected task fMRI data while participants repeatedly made an 'ee' sound, to isolate movement of the larynx while minimizing respirations and jaw and tongue motion. We observed a dual laryngeal representation that was confined to the mouth area rather than extending into the inter-effector regions (Supplementary Fig. 5), consistent with ref. 41 and a concentric functional zone organization.

Regions in CON instantiate action plans, suggesting that the CON-to-inter-effector connection could carry general action planning signals. Across foot and hand movements in a novel coordination task, the inter-effectors showed greater activity during action planning than movement execution but the effector-specific regions did not (Fig. 3e), suggesting that the implementation of action plans may be enabled in part by the inter-effector regions in M1.

## Macaque homologue of body/action network

To link these neuroimaging findings to decades of detailed motor mapping in non-human primates, we searched for inter-effector homologues in macaques using fMRI. Seeds placed in macaque M1 revealed foot, hand and mouth effector-specific functional connectivity patterns consistent with those seen in humans[24] (Extended Data Fig. 9, rows 2–4). Seeding putative CON homologues in dACC (see Supplementary Table 2

for seed locations), revealed strong connectivity with lateral frontal cortex, insula and supramarginal gyrus, similar to the human CON, and with two regions in anterior central sulcus potentially homologous to the superior and middle inter-effectors (Extended Data Fig. 9, row 1).

Distinct patterns of corticospinal connectivity are known to distinguish separable regions of macaque M1[12,13]. Phylogenetically newer, posterior M1 represents the effectors[14], projects contralaterally—mainly to the cervical and lumbar enlargements of the spinal cord[13]—and contains more projections that synapse directly onto muscle-innervating spinal neurons[12] for fine motor control. By contrast, older anterior M1 represents the body[14], projects bilaterally throughout the spinal cord[13], and connects to internal organs such as the adrenal medulla[10] and stomach[42]. Notably, the spatial distribution of adrenal connectivity[10] converges with the proposed inter-effector homologues and connected medial wall regions (SMA and dACC).

Direct stimulation studies in macaques have evoked complex, multi-effector actions by applying longer stimulation trains (500 ms) to motor cortex[4]. These actions range from feeding behaviours to climbing and defensive postures—movements that are purposeful and coordinated rather than isolated, involving integration of muscles across the classic foot, hand and mouth divisions. The inter-effector regions, which are connected to action planning areas (Fig. 2) and are active during a wide range of foot, hand and mouth movements (Extended Data Fig. 7), represent candidate human homologues to the macaque multi-effector action sites.

## Effector isolation versus action integration

Penfield conceptualized his direct stimulation findings in M1 as a continuous map of the human body—the homunculus—an organizational principle that has been dominant for almost 100 years (Fig. 4a). On the basis of novel and extant data, we instead propose a dual-systems, integrate–isolate model of behavioural control, in which effector isolating and whole-organism action implementation regions alternate (Fig. 4b). This model better fits the human imaging data presented here demonstrating contrasting structural, functional and connectivity patterns within M1 (Extended Data Fig. 4). The inter-effector patterning emerges in infancy and is preserved even in the presence of substantial perinatal cortical injury (Extended Data Fig. 2). In the integrate–isolate model, the regions for foot, hand and mouth fine motor skill are organized somatotopically as three concentric functional zones with distal parts of the effector (toes, fingers and tongue) at the centre and proximal ones (knee, shoulder and larynx) on the perimeter (Fig. 3a,b, Extended Data Fig. 7 and Supplementary Fig. 4). It has been suggested that this concentric organization extends to the ordering of fingers within the hand representation[43]. Effector-specific regions activate strongly for preferred movements and are commonly deactivated for non-preferred movements (Fig. 3d and Extended Data Fig. 7). The inter-effector regions at the edges of the effector zones coordinate with each other and with the CON (Extended Data Figs. 7 and 8; see also ref. 44) to accomplish holistic, whole-body functions in the service of performing actions (Fig. 3e). The present work suggests that these functions include action implementation, as well as postural and gross motor control of axial muscles, and prior work in humans and non-human primates suggests that these circuits may also regulate arousal[7], coordinate breathing with speech and other complex actions[45], and control internal processes and organs (such as, blood pressure[6], stomach[42] and adrenal medulla[10]), consistent with circuits for whole-body, metabolic and physiological control. Minor connectivity (Extended Data Fig. 5) and activation (Extended Data Fig. 7) differences between the superior, middle and inferior inter-effector regions probably reflect some degree of functional specialization within this integrated system. The middle region's relatively stronger connectivity to visual cortex, for example, could suggest a potential role in hand–eye coordination during reach-and-grasp motions[4].

Thus, the inter-effector system fulfils the role of a somato-cognitive action system (SCAN). The SCAN forms part of an integrated action control system, in conjunction with the CON's upstream executive control operations, to coordinate gross movements and muscle groups (such as torso and eyebrow) and enact top-down control of posture and internal physiology, while preparing for and implementing actions. These proposed functions converge with the concept of allostatic regulation by which the brain anticipates upcoming changes in physiological demands on the basis of planned actions and exerts top-down preparatory control over the body[46].

## Human electrophysiology evidence

Penfield proposed the homunculus as an approximation of group-averaged, intraoperative direct electrocortical stimulation data, which showed significant overlap across patients and body parts. He later described his artistic rendering of the homunculus as "an aid to memory […] a cartoon of representation in which scientific accuracy is impossible"[2]. Re-examination of extant human stimulation data raises doubts about the veracity of the homunculus in individuals[11] and reveals an equal or better fit with the integrate–isolate model. In some individuals, a distal-to-proximal concentric organization was documented for the upper limb, just as in non-human primates[47], whereas face movements could be elicited from areas dorsal to the hand representation[48]. In addition to focal movements, several other response types are routinely elicited with M1 stimulation, all of which can be better accounted for by whole-organism control regions. Individuals have reported the urge to move while being aware that they are holding still; they have reported a sense of moving even though no movement is detectable; or they have moved but denied having done so[2]—effects consistent with modulation of a system also representing action goals. These responses are similar to those typically elicited in CON regions such as dACC[49] and anterior parietal cortex[50].

Stimulations almost never produce isolated torso or shoulder movements[47], and a common outcome of stimulation is no reported response at all[2]. Historically, stimulations that did not elicit movement were not documented. However, we re-analysed motor stimulations from a recent large study[51] by mapping them onto cortex, revealing a region that never elicited movement in any individual, corresponding to the middle inter-effector region (Extended Data Fig. 10a). These results suggest that stimulation strengths deemed safe in humans may not typically elicit movements in the M1 SCAN regions, akin to higher-order lateral and medial premotor regions[52].

Human brain–computer interface (BCI) recordings in M1, near the superior SCAN node, have also demonstrated whole-body movement tuning[53], possibly reflecting inter-effector activity and suggesting that the inter-effector motif could provide a target for whole-body BCI. Human speech BCI studies have suggested that the precentral gyrus between the hand and mouth effectors is essential for phonological-motoric aspects of speech planning[54], and that speech can also be decoded from a region at the bottom of precentral gyrus[55]. Following our identification of the SCAN, human depth electrode recordings verified that a portion of M1 between the foot and hand effector regions (superior SCAN) is active during foot, hand and mouth movements, strongly supporting its integrative, whole-body function[44].

## Evidence from clinical neurology

Brain lesion data further support the existence of dual systems for movement isolation and action integration, with partial redundancy in M1. Motor deficits after middle cerebral artery strokes are unilateral, more severe in most distal effectors, and without significant global organismal control deficits[56]. By contrast, lesions of SCAN-linked CON regions (dACC, anterior insula and aPFC) can cause isolated volitional

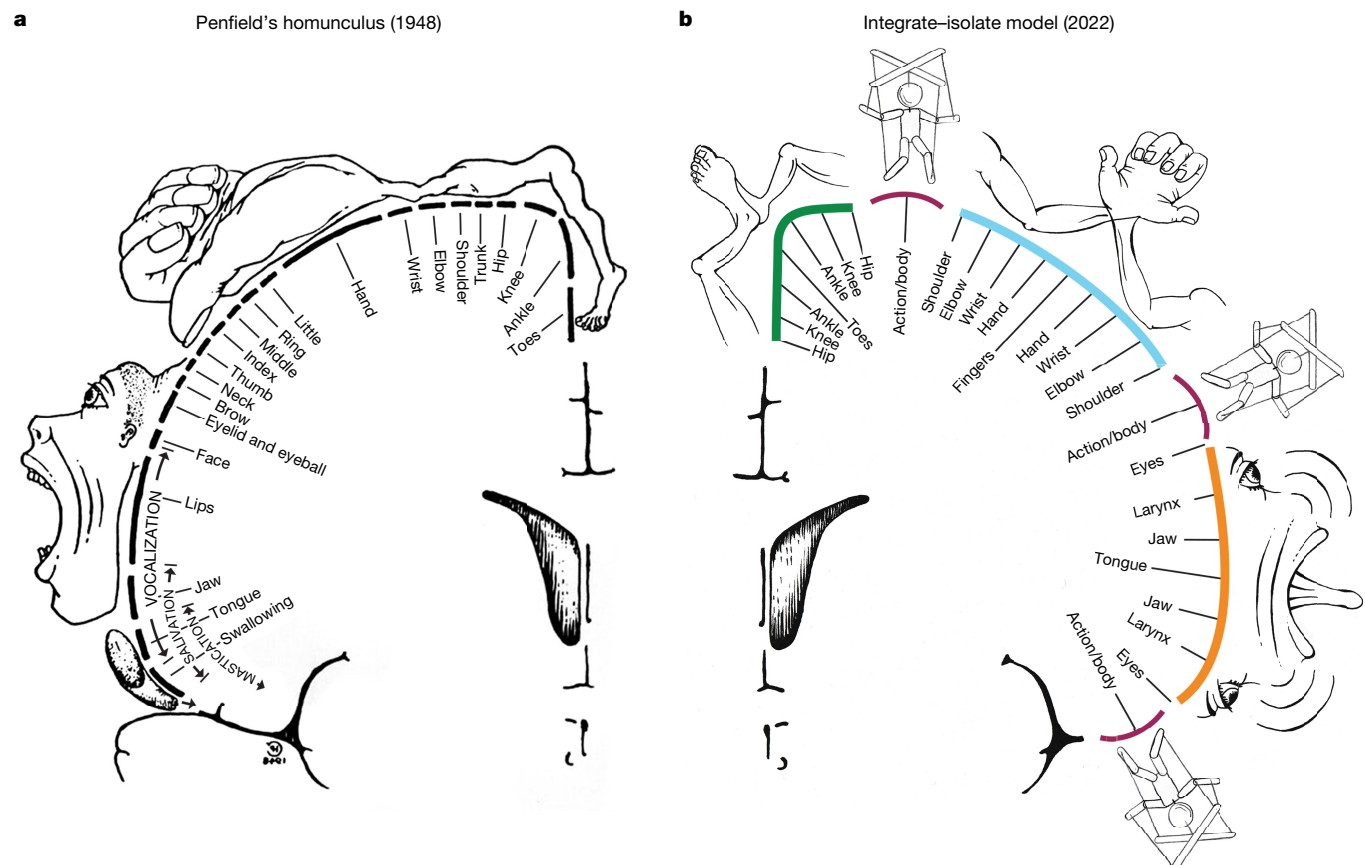

**a** Penfield's homunculus (1948)

**b** Integrate–isolate model (2022)

**Fig. 4 | The interrupted homunculus, an integrate–isolate model of action and motor control. a**, Penfield's classical homunculus (adapted from ref. 2), depicting a continuous map of the body in primary motor cortex. **b**, In the integrate–isolate model of M1 organization, effector-specific—foot (green), hand (cyan) and mouth (orange)—functional zones are represented by concentric rings with proximal body parts surrounding the relatively more isolatable distal ones (toes, fingers and tongue). Inter-effector regions (maroon) sit at the intersecting points of these fields, forming part of a somato-cognitive action network for integrative, allostatic whole-body control. As with Penfield's original drawing, this diagram is intended to illustrate organizational principles, and must not be over-interpreted as a precise map.

deficits ranging from decreased fluency to abulia to akinetic mutism, with preserved motor abilities but little self-generated movement[57]. Similarly, anterior motor lesions in macaques can spare visually guided movements while selectively disrupting internally generated actions[58], whereas posterior lesions preserve intentionality but disrupt execution[59]. Animals with lesions in effector M1 typically recover gross effector control very quickly[60], whereas fine finger movement deficits persist longer[59,61]. More rapid recovery of gross motor abilities may be in part caused by proximal functions being taken up by the contra-lesional SCAN circuits, enabled by their bilateral spinal cord connectivity. Persistent deficits may therefore be more likely in functions uniquely supported by the effector-specific circuitry.

In an individual with extensive bilateral perinatal strokes but typical motor ability, extensive post-stroke reorganization maintained the SCAN patterning at the cost of part of the already reduced M1 hand area[32]. The top third of M1 was destroyed, and surviving cortex contained an M1 hand area that was ventrally shifted and much smaller than in typical control brains. Surprisingly, SCAN regions were identified both above and below the surviving effector-specific hand region (Extended Data Fig. 2f), highlighting the importance of the SCAN for typical motor ability.

With specific connections to thalamic motor nuclei used as targets for clinical intervention (VIM and CM), the CON-linked SCAN may be relevant for a variety of movement disorders, including dystonia or essential tremor (Supplementary Information). Of particular note,

many symptoms of Parkinson's disease span motor, physiological and volitional domains (for example, postural instability, autonomic dysfunction and reduced self-initiated activity, among many others[62]), mirroring SCAN connections to regions relevant for postural control (cerebellar vermis), volition and physiological regulation (CON[6,7,57]).

## Similarities to sensory systems

Many of the organizational features of M1 described here have clear parallels in sensory systems. Similar to the concentric somatotopic organization with fine finger movements at the centre, primary visual cortex over-represents higher acuity processing at the centre, concentrically transitioning to lower acuity in the periphery[63]. Similar to our integrate–isolate dual-systems model, visual processing streams are parallel and separated in thalamus, early visual cortex and higher-order visual processing streams, with each level of processing maintaining segregation of different types of information (for example, early: eccentricity versus angle[64]; late: faces versus objects[65]). Auditory processing may have similar features, as acoustic signals are processed at least partially in parallel for hearing and speech perception in superior temporal gyrus[66]. These findings suggest shared organizational principles across the brain's input and output processing streams. It is possible that S1 also includes concentric organizational elements, which should be explored in future work.

## A network for mind–body integration

Two behavioural control systems are interleaved in human M1. One well-known system consists of effector-specific circuits for precise, isolated movements of highly specialized appendages—fingers, toes and tongue—the type of dexterous motion needed for speaking or manipulating objects. A second, integrative output system, the SCAN, is more important for controlling the organism as a whole. The SCAN integrates body control (motor and autonomic) and action planning, consistent with the idea that aspects of higher-level executive control might derive from movement coordination[67]. The SCAN includes specific regions of M1, SMA, thalamus (VIM and CM), posterior putamen and the postural cerebellum, and is functionally connected to dACC regions linked to free will[68], parietal regions representing movement intentions[50], and insular regions for processing somatosensory, pain[9] and interoceptive visceral signals[33]. The apparent relative expansion of SCAN regions in humans could suggest a role in complex actions specific to humans, such as coordinating breathing for speech, and integrating hand, body and eye movement for tool use. A common factor across this wide range of processes is that they must be integrated if an organism is to achieve its goals through movement while avoiding injury and maintaining physiological allostasis[46]. The SCAN provides a substrate for this integration, enabling pre-action anticipatory postural, breathing, cardiovascular and arousal changes (such as shoulder tension, increased heart rate or 'butterflies in the stomach'). The finding that action and body control are melded in a common circuit could help explain why mind and body states so often interact.

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

¹Mallinckrodt Institute of Radiology, Washington University School of Medicine, St Louis, MO, USA. ²Department of Neurology, Washington University School of Medicine, St Louis, MO, USA. ³Department of Biomedical Engineering, Washington University in St. Louis, St Louis, MO, USA. ⁴Department of Neurology, New York University Langone Medical Center, New York, NY, USA. ⁵Department of Psychiatry, Weill Cornell Medicine, New York, NY, USA. ⁶Department of Psychiatry, Washington University School of Medicine, St Louis, MO, USA. ⁷Department of Pediatrics, University of Minnesota, Minneapolis, MN, USA. ⁸Department of Cognitive Science, University of California San Diego, La Jolla, CA, USA. ⁹Center for the Developing Brain, Child Mind Institute, New York, NY, USA. ¹⁰Department of Psychological and Brain Sciences, Washington University in St. Louis, St Louis, MO, USA. ¹¹Department of Pediatrics, Washington University School of Medicine, St Louis, MO, USA. ¹²Department of Neuroscience, University of Minnesota, Minneapolis, MN, USA. ¹³Department of Neurosurgery, Washington University School of Medicine, St Louis, MO, USA. ¹⁴Department of Psychology, Florida State University, Tallahassee, FL, USA. ¹⁵Department of Neuroscience, Washington University School of Medicine, St Louis, MO, USA. ¹⁶Masonic Institute for the Developing Brain, University of Minnesota, Minneapolis, MN, USA. ¹⁷Institute of Child Development, University of Minnesota, Minneapolis, MN 55455, United States. ¹⁸Program in Occupational Therapy, Washington University in St. Louis, St Louis, MO, USA. ✉e-mail: egordon@wustl.edu; dosenbachn@wustl.edu

## Methods

### Washington University adult participants

Data were collected from three healthy, right-handed, adult participants (aged 35, 25 and 27 years; 1 female) as part of a study investigating effects of arm immobilization on brain plasticity (previously published data[29,30,69]). Written informed consent was obtained from all participants. The study was approved by the Washington University School of Medicine Human Studies Committee and Institutional Review Board. The primary data employed here were collected either prior to the immobilization intervention (participants 1 and 3) or two years afterwards (participant 2). Data collected immediately after the intervention are presented for within-participant replication in Extended Data Fig. 1b. For details concerning data acquisition and processing, see ref. 29.

For two participants (participants 1 and 2), we collected additional fMRI data using the same sequence during performance of two motor tasks: a somatotopic mapping task and a motor control task.

### Movement task battery.
A block design was adapted from the motor task in ref. 31. In each run, the participant was presented with visual cues that directed them to perform one of five specific movements. Each block started with a 2.2-s cue indicating which movement was to be made. After this cue, a centrally presented caret replaced the instruction and flickered once every 1.1 s (without temporal jittering). Each time the caret flickered, participants executed the proper movement. Twelve movements were made per block. Each block lasted 15.4 s, and each task run consisted of 2 blocks of each type of movement as well as 3 blocks of resting fixation. Movements conducted within each run were as follows:

Run type 1: Close left (L) hand/Close right (R) hand/Flex L foot/Move tongue L and R (participant 1: 24 runs; participant 2: 20 runs).

Run type 2: Flex L elbow/Flex R elbow/Flex L wrist/Flex R wrist/Lift bilateral shoulders (participant 1: 10 runs; participant 2: 11 runs).

Run type 3: Flex L gluteus/Flex R gluteus/Tense abdomen/Open and close mouth/Swallow (participant 1: 10 runs; participant 2: 11 runs).

Run type 4: Flex L ankle/Flex R ankle/Bend L knee/Bend R knee/Flex bilateral toes (participant 1: 10 runs; participant 2: 11 runs).

Run type 5: Lift L eyebrow/Lift R eyebrow/Wink L eyelid/Wink R eyelid/Flare nostrils (participant 1: 10 runs; participant 2: 11 runs).

### Action control and coordination task.
An event-related design implemented using JSpsych toolbox v6.3 was used to discriminate planning and execution of limb movement. See Supplementary Fig. 6 for an illustration of this task. Within the run, the participant is prompted to move either a single limb or to simultaneously move two limbs. There are four possible motions—open–close of fingers or toes, left–right flexion of the wrist or ankle, clockwise rotation of the wrist or ankle, and anticlockwise rotation of the wrist or ankle—each of which may be executed by any of the four extremities (left or right upper or lower extremity). Each motion–extremity combination may be required in isolation, or in combination with a second simultaneous motion. The participant is cued to prepare the movement(s) when they see one or two movement symbols placed on a body shape in a grey colour (planning phase), and is then cued to execute the movement(s) when the grey symbol or symbols turn green (execution phase). Using a pseudorandom jitter, the planning phase can last from 2 to 6.5 s followed by 4 to 8.5 s of movement execution. Each movement trial (planning and execution) is followed by a jittered fixation of up to 5 s. A rest block of 8.6 s is implemented every 12 movements. Two possible movements are requested during the task run and practiced before the task. The movement pair is changed for each task run. 48 trials were collected in each run. Twelve total runs were acquired per participant.

### Laryngeal mapping task.
For the same two participants, as well as for one additional participant (participant 8, 40 years of age, from whom written informed consent was obtained) additional fMRI data were collected during performance of a laryngeal mapping task using a multiband five-echo blood oxygen level-dependent (BOLD) contrast sensitive gradient echo-planar sequence (flip angle = 68°, resolution = 2.0 mm isotropic, TR = 1,761 ms, multiband 6 acceleration, $TE_1$: 14.20 ms, $TE_2$: 38.93 ms, $TE_3$: 63.66 ms, $TE_4$: 88.39 ms, and $TE_5$: 113.12 ms), with each run lasting 3 min 52 s. A pair of spin-echo echo-planar images (EPI) with opposite phase-encoding directions (anterior→posterior (AP) and posterior→anterior (PA)) but identical geometrical parameters to the BOLD sequence were acquired. In participant 8, 15 min of resting-state fMRI were also acquired using the sequence above.

An additional set of ten movement task runs adapted from[31] were collected to localize laryngeal phonation. In each run, the participant was presented with visual cues that directed them to perform one of six specific movements: Left hand, Right hand, Left foot, Right foot, Tongue or Voice. In the Voice condition, participants were required to briefly make the noise "eeee" without moving their jaw. Each block started with a 3.0 s cue indicating which movement was to be made. After this cue, a centrally presented caret replaced the instruction and flickered once every 1.0 s (without temporal jittering). Each time the caret flickered, participants executed the proper movement. Ten movements were made per block. Each block lasted 15.0 s, and each task run consisted of 2 blocks of each type of movement as well as 2 blocks of resting fixation. Each participant completed 10 runs.

### Cornell adult participants

Data were collected from four healthy adult participants (ages 29, 38, 24 and 31; all male) as part of a previously published study[70]. The study was approved by the Weill Cornell Medicine Institutional Review Board. Written informed consent was provided by each participant.

For details concerning data acquisition and processing, see ref. 70.

### Neonatal participant

Data were collected from one sleeping, healthy full-term neonatal participant beginning 13 days after birth, corresponding to 42 weeks post-menstrual age. The study was approved by the Washington University School of Medicine Human Studies Committee and Institutional Review Board. Written informed consent was provided by a parent.

### MRI acquisition.
The participant was scanned while asleep over the course of 4 consecutive days using a Siemens Prisma 3T scanner on the Washington University Medical Campus. Every session included collection of a high-resolution T2-weighted spin-echo image (TE = 563 ms, TR = 3,200 ms, flip angle = 120°, 208 slices with 0.8 × 0.8 × 0.8 mm voxels). In each session, a number of 6 min 45 s multi-echo resting-state fMRI runs were collected as a five-echo BOLD contrast sensitive gradient echo-planar sequence (flip angle = 68°, resolution = 2.0 mm isotropic, TR = 1,761 ms, multiband 6 acceleration, $TE_1$: 14.20 ms, $TE_2$: 38.93 ms, $TE_3$: 63.66 ms, $TE_4$: 88.39 ms, and $TE_5$: 113.12 ms). The number of BOLD runs collected in each session depended on the ability of the neonate to stay asleep during that scan; across the 4 days, 23 runs were collected in total. A pair of spin-echo EPI images with opposite phase-encoding directions (AP and PA) but identical geometrical parameters and echo spacing were acquired between every three BOLD runs or any time the participant was removed from the scanner.

### MRI processing.
Structural and functional processing followed the pipeline used for the Washington University dataset, with two exceptions. First, segmentation, surface delineation and atlas registration were conducted using a T2-weighted image (the single highest quality T2 image, as assessed via visual inspection) rather than a T1-weighted image, due to the inverted image contrast observed in neonates.

Second, after the multi-echo BOLD data were unwarped and normalized to atlas space, it was optimally combined before nuisance regression and mapping to cifti space. All fMRI scans from the second day of scanning were excluded due to registration abnormalities.

### Infant participant

Data were collected from one healthy sleeping infant aged 11 months. The study was approved by the Washington University School of Medicine Human Studies Committee and Institutional Review Board. Written informed consent was provided by a parent.

**MRI acquisition.** The participant was scanned while asleep over the course of three sessions using a Siemens Prisma 3T scanner on the Washington University Medical Campus. The first session included collection of a high-resolution T1-weighted MP-RAGE (TE = 2.24 ms, TR = 2,400 ms, flip angle = 8°, 208 slices with 0.8 × 0.8 × 0.8 mm voxels) and a T2-weighted spin-echo image (TE = 564 ms, TR = 3200 ms, flip angle = 120°, 208 slices with 0.8 × 0.8 × 0.8 mm voxels). The second and third sessions included collection of 26 total runs of resting-state fMRI, each collected as a 6 min 49 s-long BOLD contrast sensitive gradient echo-planar sequence (flip angle = 52°, resolution = 3.0 mm isotropic, TE = 30 ms, TR = 861 ms, multiband 4 acceleration). For each run, a pair of spin-echo EPI images with opposite phase-encoding directions (AP and PA) but identical geometrical parameters and echo spacing were acquired to correct spatial distortions.

**MRI processing.** Structural processing followed the DCAN Labs processing pipeline found in the ABCD BIDS Community Collection (ABCC; NDA Collection 3165) (https://github.com/DCAN-Labs/abcd-hcp-pipeline)[71], which we found performed the best surface segmentation at this age. Functional processing followed the pipeline used for the Washington University adult dataset.

### Child participant

Data were collected from one healthy awake male child age 9 years. The study was approved by the Washington University School of Medicine Human Studies Committee and Institutional Review Board. Written informed consent was provided by a parent and assent was given by the participant.

**MRI acquisition.** The participant was scanned repeatedly over the course of 12 sessions using a Siemens Prisma 3T scanner on the Washington University Medical Campus. These sessions included collection of 14 high-resolution T1-weighted MP-RAGE images (TE = 2.90 ms, TR = 2,500 ms, flip angle = 8°, 176 slices with 1 mm isotropic voxels), 14 T2-weighted spin-echo images (TE = 564 ms, TR = 3200 ms, flip angle = 120°, 176 slices with 1 mm isotropic voxels), and 26 total runs of resting-state fMRI, each collected as a 10 min-long BOLD contrast sensitive gradient echo-planar sequence (flip angle = 84°, resolution = 2.6 mm isotropic, 56 slices, TE = 33 ms, TR = 1,100 ms, multiband 4 acceleration). In each session, a pair of spin-echo EPI images with opposite phase-encoding directions (AP and PA) but identical geometrical parameters and echo spacing were acquired to correct spatial distortions in the BOLD data.

**MRI processing.** Structural and functional processing followed the DCAN Labs processing pipeline found in the ABCD BIDS Community Collection (ABCC; NDA Collection 3165)[71] (https://github.com/DCAN-Labs/abcd-hcp-pipeline).

### Participant with perinatal stroke

PS1, a left-handed, 13-year-old male who played for a competitive youth baseball team, was referred to an orthopaedic physician because of difficulty using his right arm effectively. Ulnar neuropathy was considered and he was referred for physical therapy. However, PS1 was first seen by a child neurologist (N.U.F.D.) for further evaluation. Structural brain MRI revealed unexpectedly extensive bilateral cystic lesions consistent with perinatal infarcts. Review of PS1's medical history revealed that the injury occurred in the perinatal period.

Data acquisition from PS1 were performed with the approval of the Washington University Institutional Review Board. Written informed consent was provided by PS1's mother and assent was given by PS1 at the time of data acquisition.

For additional details regarding clinical history, neuropsychological evaluations, motor assessments, or MR image acquisition or processing, see ref. 32.

### UMN macaque

Data were collected from a sedated adult female macaque monkey (*Macaca fascicularis*) aged 6 years. Experimental procedures were carried out in accordance with the University of Minnesota Institutional Animal Care and Use Committee and the National Institute of Health standards for the care and use of non-human primates. The subject was fed ad libitum and pair-housed within a light- and temperature-controlled colony room. The animal was not water restricted. The subject did not have any prior implant or cranial surgery. The animal was fasted for 14–16 h prior to imaging. On scanning days, anaesthesia was first induced by intramuscular injection of atropine (0.5 mg kg$^{-1}$), ketamine hydrochloride (7.5 mg kg$^{-1}$), and dexmedetomidine (13 μg kg$^{-1}$). The subject was transported to the scanner anteroom and intubated using an endotracheal tube. Initial anaesthesia was maintained using 1.0%–2% isoflurane mixed with oxygen (1 l min$^{-1}$ during intubation and 2 l m$^{-1}$ during scanning to compensate for the 12-m length of the tubing used). For functional imaging, the isoflurane level was lowered to 1%. The subject was placed onto a custom-built coil bed with integrated head fixation by placing stereotactic ear bars into the ear canals. The position of the animal corresponds to the sphinx position. Experiments were performed with the animal freely breathing. Continuous administration of 4.5 μg kg$^{-1}$ h$^{-1}$ dexmedetomidine using a syringe pump was administered during the procedure. Rectal temperature (~37.6 °C), respiration (10–15 breaths per min), end-tidal $CO_2$ (25–40), electro-cardiogram (70–150 bpm), and peripheral capillary oxygen saturation (SpO$_2$) (>90%) were monitored using an MRI compatible monitor (IRAD-IMED 3880 MRI Monitor). Temperature was maintained using a circulating water bath as well as chemical heating pads and padding for thermal insulation.

**MRI acquisition.** Data were acquired on a Siemens Magnetom 10.5 T Plus. A custom in-house built and designed RF coil was used with an 8-channel transmit/receive end-loaded dipole array of 18-cm length combined with a close-fitting 16- channel loop receive array head cap, and an 8-channel loop receive array of 50 × 100 mm$^2$ size located under the chin[72]. A B1+ (transmit B1) field map was acquired using a vendor provided flip angle mapping sequence and then power calibrated for each individual. Following B1+ transmit calibration, 3–5 averages (23 min) of a T1-weighted MP-RAGE were acquired for anatomical processing (TR = 3300 ms, TE = 3.56 ms, TI = 1,140, flip angle = 5°, slices = 256, matrix = 320×260, acquisition voxel size = 0.5 × 0.5 × 0.5 mm$^3$, in-plane acceleration GRAPPA = 2). A resolution and field of view-matched T2-weighted 3D turbo spin-echo sequence was run to facilitate B1 inhomogeneity correction. Five images were acquired in both phase-encoding directions (R→L and L→R) for offline EPI distortion correction. Six runs of fMRI time series, each consisting of 700 continuous 2D multiband EPI[73–75] functional volumes (TR = 1,110 ms; TE = 17.6 ms; flip angle = 60°, slices = 58, matrix = 108 × 154; field of view = 81 × 115.5 mm ; acquisition voxel size = 0.75 × 0.75 × 0.75 mm) were acquired with a left–right phase-encoding direction using in-plane acceleration factor GRAPPA = 3, partial Fourier = 7/8, and MB factor = 2. Since the macaque was scanned in sphinx position, the orientations noted here are what is consistent with a (head first supine) typical

human brain study (in terms of gradients) but translate differently to the actual macaque orientation.

**MRI processing.** Processing followed the DCAN Labs non-human primate processing pipeline (http://github.com/DCAN-Labs/nhp-abcd-bids-pipeline), with minor modifications. Specifically, we observed that distortion from the 10T scanner was so extensive that the field maps did not fully correct it. Therefore, instead of field map-based unwarping, we used the computed field map-based warp as an initial starting point for Synth, a field map-less distortion correction algorithm that creates synthetic undistorted BOLD images for registration to anatomical images[76]. Synth substantially reduced residual BOLD image distortion.

## PRIME-DE macaque

Raw structural and functional data were provided from the Oxford dataset of the PRIMatE Data Exchange (PRIME-DE) consortium (https://fcon_1000.projects.nitrc.org/indi/PRIME/oxford.html)[77,78]. The full dataset consisted of 19 (age 4.1 ± 0.98 years, weight 6.61 ± 2.94 kg) rhesus macaques (*Macaca mulatta*). The animal care, anaesthesia and MRI protocols were carried out in accordance with the UK Animals (Scientific Procedures) Act of 1986. Animals in the study were group-housed prior to scanning. Ketamine (10 mg kg$^{-1}$) was administered via intramuscular injection for induction, along with either xylazine (0.125–0.25 mg kg$^{-1}$), midazolam (0.1 mg kg$^{-1}$), or buprenorphine (0.01 mg kg$^{-1}$). Additionally, injections of atropine (0.05 mg kg$^{-1}$, intramuscular injection), meloxicam (0.2 mg kg$^{-1}$, intravenous injection), and ranitidine (0.05 mg kg$^{-1}$) were administered. A minimum of 15 min prior to being placed in the stereotaxic frame, animals also received local anaesthetics (5% lidocaine/prilocaine cream and 2.5% bupivacaine injected subcutaneously around ears). Finally, anaesthesia was maintained with isoflurane, and scanning began 1.5–2 h after the initial ketamine induction.

**MRI acquisition.** Anaesthetized animals were placed in the sphinx position into a stereotactic frame (Crist Instrument) and scanned in a horizontal 3T MRI scanner using a four-channel phased-array coil (Windmiller Kolster Scientific, Fresno, CA). Each animal received 53.33 min (1,600 volumes) of resting-state data, which was acquired at a 2.0 mm isotropic voxel resolution (TR = 2,000 ms, TE = 19 ms, Flip angle = 90°). A T1-weighted MP-RAGE sequence was used to acquire anatomical data (TR = 2,500 ms, TE = 4.01 ms, TI = 1,100, flip angle = 8°, acquisition voxel size = 0.5 × 0.5 × 0.5 mm, 128 slices).

**MRI processing.** Processing for structural data followed the DCAN Labs non-human primate processing pipeline (https://github.com/DCAN-Labs/nhp-abcd-bids-pipeline). Smoothing was applied with FWHM = 1.5 mm in both volume and surface space. The surface data were then down-sampled to a 10k surface to create the preprocessed cifti data. Finally, each animal's data was closely visually inspected for quality. Following these inspections, data from 11 animals were excluded due to the presence of artefact in or near the central sulcus, leaving eight animals in the final data. This sample size of eight was chosen to include all available artefact-free data. No randomization or blinding was performed.

## Group-averaged datasets

Resting-state fMRI data was averaged across participants within each of five large datasets.

## UK Biobank

A group-averaged weighted eigenvectors file from an initial batch of 4,100 UKB participants aged 40–69 years (53% female) scanned using resting-state fMRI for 6 min was downloaded from https://www.fmrib.ox.ac.uk/ukbiobank/. This file consisted of the top 1,200 weighted spatial eigenvectors from a group-averaged principal component analysis. See ref. 79 and documentation at https://biobank.

ctsu.ox.ac.uk/crystal/ukb/docs/brain_mri.pdf for details of the acquisition and processing pipeline. This eigenvectors file was mapped to the Conte69 surface template atlas[80] using the ribbon-constrained method in Connectome Workbench[81], and the eigenvector time courses of all surface vertices were cross-correlated.

## Adolescent Brain Cognitive Development Study

Twenty minutes (4 × 5-min runs) of resting-state fMRI data, as well as high-resolution T1-weighted and T2-weighted images, were collected from 3,928 9- to 10-year-old participants (51% female), who were selected as the participants with at least 8 min of low-motion data from a larger scanning sample. Data collection was performed across 21 sites within the USA, harmonized across Siemens, Philips and GE 3T MRI scanners. See ref. 82 for details of the acquisition parameters. Data processing was conducted using the ABCD-BIDS pipeline found in the ABCD BIDS Community Collection (ABCC; NDA Collection 3165) (https://github.com/DCAN-Labs/abcd-hcp-pipeline)[71]; see ref. 83 for details.

## Human Connectome Project

A vertexwise group-averaged functional connectivity matrix from the HCP 1200 participants release was downloaded from https://db.humanconnectome.org. This matrix consisted of the average strength of functional connectivity across all 812 participants aged 22–35 years (410 female) who completed 4× 15-min resting-state fMRI runs and who had their raw data reconstructed using the newer recon 2 software. See refs. 81,84–86 for details of the acquisition and processing pipeline.

## Washington University 120

Data were collected from 120 healthy young adult participants recruited from the Washington University community during relaxed eyes-open fixation (60 females, ages 19–32). Scanning was conducted using a Siemens TRIO 3.0T scanner and included collection of high-resolution T1-weighted and T2-weighted images, as well as an average of 14 min of resting-state fMRI. See ref. 87 for details of the acquisition and processing pipeline.

## Neonates

Mothers were recruited during the second or third trimester from two obstetrics clinics at Washington University as part of the Early Life Adversity, Biological Embedding, and Risk for Developmental Precursors of Mental Disorders (eLABE) study. This study was approved by the Human Studies Committees at Washington University in St. Louis and written informed consent was obtained from mothers. Neuroimaging was conducted in full-term, healthy neonate offspring shortly after birth (average post-menstrual age of included participants 41.4 weeks, range 38–45 weeks). Of the 385 participants scanned for eLABE, 262 were included in the current analyses (121 female). See ref. 88 for additional details of the participants, criteria for exclusion, scanning acquisition protocol and parameters, and processing pipeline.

## Analyses

**Functional connectivity.** For each single-participant dataset, a vertex or voxelwise functional connectivity matrix was calculated from the resting-state fMRI data as the Fisher-transformed pairwise correlation of the time series of all vertices/voxels in the brain. In the ABCD, Washington University 120, eLABE and PRIME-DE datasets, vertex and voxelwise group-averaged functional connectivity matrices were constructed by first calculating the vertex or voxelwise functional connectivity within each participant as the Fisher-transformed pairwise correlation of the time series of all vertices or voxels in the brain, and then averaging these values across participants at each vertex or voxel.

**Seed-based functional connectivity.** We defined a continuous line of seeds down the left precentral gyrus by selecting every vertex in a continuous straight line on the cortical surface between the most ventral aspect of the medial motor area (approximate MNI coordinates (−4, −31, 54)) and the ventral lip of the precentral gyrus right above the operculum (approximate MNI coordinates (−58, 4, 8)). For each seed, we

examined its map of functional connectivity as the Fisher-transformed correlation between that vertex's time course and that of every other vertex or voxel in the brain.

**Network detection in somatomotor cortex.** To define the somatomotor regions that were visually identified from the seed-based connectivity analysis in an unbiased fashion for further exploration, we entered each individual adult human participant's data into a data-driven network detection algorithm designed to identify network subdivisions that are hierarchically below the level of classic large-scale networks (for example, those that produce hand/foot divisions in somatomotor cortex;[23,37]). We have previously described how this approach identifies sub-network structures that converge with task-activated regions[89] and with known neuroanatomical systems[90].

In each adult participant, this analysis clearly identified network structures corresponding to motor representation of the foot, hand and mouth; and it additionally identified network structures corresponding exactly to the previously unknown connectivity pattern identified from the seed-based connectivity exploration as the inter-effector regions. For simplicity, we manually grouped all inter-effector subnetworks together as a single putative network structure (labelled as inter-effector) for further analysis.

Finally, to identify classic large-scale networks in each participant, we repeated the Infomap algorithm on matrices thresholded at a series of denser thresholds (ranging from 0.2% to 5%), and additionally identified individual-specific networks corresponding to the default, medial and lateral visual, cingulo-opercular, fronto-parietal, dorsal attention, language, salience, parietal memory, and contextual association networks following procedures described in ref. 24. See Supplementary Fig. 6 for these individual-specific networks.

**Differences in functional connectivity between inter-effector and foot, hand or mouth regions.** Within each adult human participant, we calculated an inter-effector connectivity map as the Fisher-transformed correlation between the average time course of all cortical inter-effector vertices and the time course of every other vertex or voxel in the brain. We then repeated this procedure to calculate a connectivity map for the foot, hand and mouth areas.

To identify brain regions more strongly connected to inter-effector regions than to other motor regions, we computed the smallest positive difference in each voxel or vertex between inter-effector connectivity and any foot, hand or mouth connectivity. That is, we calculated (inter-effector – max[foot, hand, mouth]). This represents a conservative approach that only identifies regions of the brain for which the inter-effector regions are more strongly connected than any of the other motor areas.

**Differences in functional connectivity among inter-effector regions.** Within each adult human participant, as well as in the HCP group-averaged data, we computed a connectivity map for each of the three distinct inter-effector regions (superior, middle and inferior) as the Fisher-transformed correlation between the average time course of all cortical vertices in the two bilateral regions in each position and the time course of every other vertex or voxel in the brain.

To identify brain regions more strongly connected to one of the inter-effector regions than the other two, we computed the smallest positive difference in each voxel or vertex between that region's connectivity and either of the other two regions' connectivity. That is, we calculated (superior inter-effector – max[middle, inferior inter-effector]), (middle inter-effector – max[superior, inferior inter-effector]), and (inferior inter-effector – max[superior, middle inter-effector]). This represents a conservative approach that only identifies regions of the brain for which one inter-effector region is more strongly connected than either of the other two regions.

**Functional connectivity with CON.** Within each adult human participant, we calculated the functional connectivity between each of the foot, hand, mouth, and inter-effector regions and the CON. This was computed as the Fisher-transformed correlation between (1) the average time course across all vertices in the motor region and (2) the average time course across all vertices in the CON. We conducted paired *t*-tests across subjects comparing the inter-effector connectivity with CON against each of the foot, hand and mouth connectivity strengths, FDR-correcting for the three tests conducted.

We then calculated the functional connectivity between the inter-effector regions and every other large-scale cortical network in the brain (visual, auditory, salience, premotor, fronto-parietal, default mode, dorsal attention, language, contextual association, and parietal memory). The strength of connectivity between the inter-effector network and the CON was compared against the strength of its connectivity to each of these other networks using paired *t*-tests, FDR-correcting for the ten tests conducted.

**Motor and CON network visualization.** Visualization of network relationships was conducted using spring-embedded plots[23], as implemented in Gephi (https://gephi.org/). In each individual adult human participant, nodes were defined as congruent clusters of foot, hand, mouth, inter-effector, and CON networks larger than 20 mm². Pairwise connectivity between nodes was calculated as the Fisher-transformed correlation of their mean time courses. For visualization purposes, graphs were constructed by thresholding the pairwise node-to-node connectivity matrices at 40% density (the general appearance of the graphs did not change across a range of densities).

**Functional connectivity with adjacent postcentral gyrus.** In each adult human participant, we defined the pre- and postcentral gyri based on the individual-specific Brodmann areal parcellation produced by Freesurfer, which was deformed into fs_LR_32k space to match the functional data. Precentral gyrus was considered to be the vertices labelled as BA 4a and 4p, and postcentral gyrus was the vertices labelled as BA 3b and 2. BA 3a (fundus of central sulcus) was not considered for this analysis. Because the medial aspect of somatomotor cortex (corresponding to representation of the leg and foot) was always classified by Freesurfer as BA 4a, we defined the medial postcentral gyrus as the cortical vertices with *y*-coordinates farther posterior than the median *y*-coordinate of the foot region (from the network mapping above).

Within the participant's precentral gyrus, we labelled vertices as representing foot, hand, mouth or inter-effector according to their labels from the network mapping procedure. We then partitioned the postcentral gyrus into foot, hand, mouth and inter-effector areas depending on which precentral region each vertex was physically closest to. Finally, within each partition (foot, hand, mouth and inter-effector) we calculated the average connectivity between the pre and postcentral gyrus as the Fisher-transformed correlation between the average time courses of all vertices in each area. We then conducted paired *t*-tests across subjects comparing the inter-effector connectivity with adjacent S1 against each of the foot/hand/mouth connectivity strengths with S1, FDR-correcting for the three tests conducted.

**Functional connectivity with middle insula.** In each adult human participant, we defined the middle insula based on the individual-specific Freesurfer gyral parcellation using the Destrieux atlas[91], which was deformed into fs_LR_32k space to match the functional data. Middle insula was considered to be the vertices labelled as the superior segment of the circular sulcus of the insula or as the short insular gyrus. We then calculated the functional connectivity between each of the bilateral foot, hand, mouth, and inter-effector regions and the bilateral middle insula. We conducted paired *t*-tests across subjects comparing the inter-effector connectivity with middle insula against each of the foot, hand and mouth connectivity strengths, FDR-correcting for the number of tests conducted.

**Functional connectivity with cerebellum.** In each adult human participant, we calculated the functional connectivity between each of the foot, hand, mouth and inter-effector regions with each voxel of the cerebellum. Cerebellar connectivity strengths calculated this way were then mapped onto a cerebellar flat map using the SUIT

toolbox[92]. Connectivity strengths were averaged within each of 27 atlas regions[93]. For each region, we conducted three paired t-tests comparing inter-effector connectivity strength against foot, hand and mouth connectivity strength, FDR-correcting for the total number of tests conducted. Regions were reported if the inter-effector connectivity strength was significantly higher than the connectivity strength of all other motor regions.

**Functional connectivity with putamen.** In each adult human participant, we divided each unilateral putamen in each hemisphere into quarters by splitting it based on the median of its $y$ (anterior-posterior) and $z$ (dorsal–ventral) coordinates. We then calculated the functional connectivity between each of the foot, hand, mouth and inter-effector regions and each putamen quarter.

For each putamen division, we conducted paired t-tests across subjects comparing the inter-effector connectivity with that putamen division against each of the foot, hand and mouth connectivity strengths, FDR-correcting for the number of tests conducted. We reported divisions in which the inter-effector connectivity was significantly different from all three effector-specific connectivities.

**Functional connectivity with thalamus.** To investigate subregions of thalamus, we employed the DISTAL atlas v1.1[94], which contains a number of histological thalamic subregions identified by[95]. This atlas was down-sampled into the 2-mm isotropic space of the functional data. Functional connectivity maps seeded from the foot, hand, mouth, and inter-effector regions in each adult human participant were computed, and mean connectivity values were calculated within each atlas region. The atlas specifies multiple subregions for many nuclei; these subregions were combined and treated as single nuclei for the purposes of connectivity calculation.

For each adult human participant, we averaged the connectivity seeded from the inter-effector regions and from each of the foot, hand and mouth regions across all voxels within each thalamic nucleus. For each thalamic nucleus, we conducted paired t-tests across subjects comparing the inter-effector with the mean of the foot, hand and mouth connectivity strengths, FDR-correcting for the number of thalamic nuclei tested.

**Lag structure of RSFC.** We used a previously published method for estimating relative time delays (lags) in fMRI data[34,96]. In brief, for each session in each adult human participant, we computed a lagged cross-covariance function (CCF) between each pair of vertex or voxel time courses within the motor system and CON in the cortex. Lags were more precisely determined by estimating the cross-covariance extremum of the session-level CCF using three-point parabolic interpolation. The resulting set of lags was assembled into an antisymmetric matrix capturing all possible pairwise time delays (TD matrix) for each session, which was averaged across sessions to yield participant-level TD matrices. Finally, each participant's TD matrix was averaged across rows to summarize the average time-shift from one vertex to all other vertices. Average time lag was then averaged across all vertices with each of the precentral gyrus foot, hand, mouth and inter-effector regions, and the CON.

We then conducted paired t-tests across subjects comparing (1) the mean lag in inter-effector regions against the mean lags in each of the foot, hand and mouth regions, and (2) the mean lag in CON regions against the mean lags in each of the foot, hand and mouth regions.

**Macaque RSFC.** We placed connectivity seeds continuously along area 4p in the left hemisphere of each macaque, as well as continuously running from the dorsal cingulate motor area to the rostral cingulate motor area in the dACC (area 24). See Supplementary Fig. 8a for medial cortex seed locations and Supplementary Fig. 8b for all functional connectivity maps from all medial seeds.

**Structural MRI. Cortical thickness.** Within each adult human participant, the map of cortical thickness generated by the Freesurfer segmentation was deformed into fs_LR_32k space to match the functional data. Precentral gyrus foot, hand, mouth and inter-effector regions were defined as above, and mean cortical thickness was calculated within each region. We then conducted paired t-tests across subjects comparing the inter-effector thickness against each of the foot, hand and mouth thicknesses, correcting for the three tests conducted.

**Fractional anisotropy.** White matter fibres tracked from separate areas of M1 using diffusion imaging quickly converge into the internal capsule and become difficult to dissociate. As such, we tested for fractional anisotropy differences in the white matter immediately below the precentral gyrus.

To calculate fractional anisotropy beneath the cortex, we first constructed fs_LR_32k-space surfaces 2 mm below each grey-white surface in adult human participants 1–3. To accomplish this, for each vertex on the surface, we computed the 3D vector between corresponding points on the fs_LR_32k pial and the grey-white surfaces, and we extended that vector an additional 2 mm beyond the grey-white surface in order to create a lower surface. We then mapped the fractional anisotropy values using the using the ribbon-constrained method, mapping between the grey-white and the 2 mm-under surfaces. The result is fractional anisotropy values mapped to a lower surface within white matter that is in register to the existing fs_LR_32k surfaces on which the functional data is mapped and the motor regions defined.

Precentral gyrus foot, hand, mouth and inter-effector regions were defined as above, and we calculated mean fractional anisotropy beneath each cortical region.

We then conducted paired t-tests across subjects comparing the mean fractional anisotropy beneath the inter-effector regions against mean fractional anisotropy beneath each of the foot, hand and mouth regions.

**Myelin density.** Within each adult human participant, we created vertexwise maps of intracortical myelin content following methods described in refs. 81,97. Precentral gyrus was defined as above. Across participants, we found that baseline myelin density values (both in precentral gyrus and in the whole-brain myelin density map) varied wildly across participants in different datasets, likely based on differences in the T1- and T2-weighted sequences employed. Thus, for optimal visualization of results, in each participant we normalized the myelin density values by dividing the calculated vertexwise myelin densities in precentral gyrus by the mean myelin density across the whole precentral gyrus. Finally, precentral gyrus foot, hand, mouth and inter-effector regions were defined as above, and mean normalized myelin density was calculated within each region. We then conducted paired t-tests across subjects comparing the inter-effector myelin density against each of the foot, hand and mouth myelin densities, correcting for the three tests conducted.

**Task fMRI. Movement task battery analysis.** Basic analysis of the movement task battery data was conducted using within-participant block designs. To compute the overall degree of activation in response to each motion, data from each run was entered into a first-level analysis within FSL's FEAT[98] in which each motion block was modelled as an event of duration 15.4 s, and the combined block waveform for each motion condition was convolved with a haemodynamic response function to form a separate regressor in a generalized linear model (GLM) analysis testing for the effect of the multiple condition regressors on the time course of activity within every vertex or voxel in the brain. Beta value maps for each condition were extracted for each run and entered into a second-level analysis, in which run-level condition betas were tested against a null hypothesis of zero activation in a one-sample t-test across runs (within participant). The resulting t-values from each motion condition tested in this second-level analysis were converted to Z-scores. Z-score activation maps were smoothed with a geodesic 2D (for surface data) or Euclidean 3D (for volumetric data) Gaussian kernel of $\sigma = 2.55$ mm.

**Movement task battery winner take all.** For each vertex within the broad central sulcus area, we identified the movement that produced the greatest activation strength ($Z$-score from second-level analysis, above) in that vertex, and we assigned that motion to that vertex.

**Movement task battery curve fitting.** For each vertex within precentral gyrus, we first computed its position along the dorsal–ventral axis of left hemisphere M1. This was done by identifying the closest point within the continuous line of points running down precentral gyrus (defined in 'Seed-based functional connectivity'), and assigning that closest point's ordered position within the line to the vertex.

For every movement, we then plotted that dorsal–ventral M1 position against $Z$-score activation in each vertex. We then fit two curves to each of these relationships. The first curve was a single-Gaussian model of the form:

Activation = $a_1 \times \exp(-((\text{position-}b_1)/c_1)^2)$.

The second curve was a double-Gaussian model of the form:

Activation = $a_1 \times \exp(-((\text{position-}b_1)/c_1)^2) + a_2 \times \exp(-((\text{position-}b_2)/c_2)^2)$.

The $a_1$ and $a_2$ parameters in each model were constrained to be positive (to enforce positive-going peaks). Curve fitting was constrained to be conducted within the general vicinity of the activated area in order to avoid fitting negative activations observed in distant portions of M1. For lower extremity movements, this meant excluding the bottom third of M1; for upper extremity movements, the bottom third of M1 plus the medial wall; for face movements, the top third of M1.

Finally, we tested whether the one- or two-peak models better fit the data. This was done by conducting an $F$-test between the models, computed as:

$$F = ((\text{SSE}_{1\text{peak}} - \text{SSE}_{2\text{peaks}})/(\text{df}_{1\text{peak}} - \text{df}_{2\text{peaks}}))/(\text{SSE}_{2\text{peaks}}/\text{df}_{2\text{peaks}}).$$

where SSE represents the sum of squared errors from the model and df represents the degrees of freedom in the model.

The $P$ value was computed from this $F$ by employing the $F$-statistic continuous distribution function (fcdf.m) in Matlab and using ($\text{df}_{1\text{peak}} - \text{df}_{2\text{peaks}}$) and $\text{df}_{2\text{peaks}}$ as the numerator and denominator degrees of freedom, respectively.

**Movement task battery curve visualization.** For each movement, the complete dorsal–ventral M1 position versus $Z$-score activation profile (from above) was visualized more clearly by fitting a LOWESS curve. These LOWESS curves recapitulated the two-peak activation fits while also revealing additional task responsive cortex.

**Movement selectivity.** Based on results from the above winner-take-all analysis, we identified the movement that was most preferred at the centre of each the three effector-specific (toe movement, hand movement and tongue movement) and inter-effector regions (abdominal movement, eyelid movement and swallowing). The centre-most movements were selected to avoid issues with spreading, overlapping activation near the borders of effector-specific and inter-effector regions. For every vertex within the precentral gyrus, we compared the strength of activation between the most preferred of the six movements at that vertex against the activation of the second-most preferred movements. The differences between these activation strengths was taken to be the movement selectivity of that vertex.

**Movement coactivation.** For each region among the six resting-state-derived foot, hand, mouth, and inter-effector regions in the precentral gyrus, we calculated the average activation within that region for each movement, producing a profile of motor activation strengths for that region. We also calculated the average activation within all CON vertices for each movement. To determine the degree to which various regions were coactive across movements, we then correlated each foot, hand, mouth and inter-effector cluster's profile of activation strengths with that of all other clusters, and with that of the CON. Note: visualization of activation maps revealed some striping, suggesting that the Open and close mouth and the Bend L knee conditions were partially distorted by head motion; therefore, these conditions were excluded from analysis, although their inclusion did not change results.

**Laryngeal motor mapping task analysis.** As with the movement task battery, analysis of the laryngeal mapping task data was conducted using within-participant block designs. To compute the overall degree of activation in response to each motion, data from each run was entered into a first-level analysis within FSL's FEAT[98] in which each motion block was modelled as an event of duration 15.0 s, and the combined block waveform for each motion condition was convolved with a hemodynamic response function to form a separate regressor in a GLM analysis testing for the effect of the multiple condition regressors on the time course of activity within every vertex or voxel in the brain. Beta value maps for each condition were extracted for each run and entered into a second-level analysis, in which run-level condition betas were tested against a null hypothesis of zero activation in a one-sample $t$-test across runs (within participant). The resulting $t$-values from each motion condition tested in this second-level analysis were converted to $Z$-scores.

Note that the laryngeal mapping data was not included in the movement task battery analysis because it was collected on a different scanner with a different sequence, and so would not be directly comparable.

**Action control and coordination task analyses.** Analysis of the action control task was conducted using within-participant event-related designs. For each separate run, a GLM model was constructed in FEAT[98] in which separate regressors described the initiation of (1) planning and (2) execution of each type of movement (4 movements × 4 limbs). Each regressor was constructed as a 0-length event convolved with a canonical haemodynamic response, and beta values for each regressor were estimated for every voxel in the brain. These beta value maps for each condition were thus computed for each run and entered into a second-level analysis, in which a $t$-test across runs contrasted the run-level planning betas against the run-level execution betas.

**Human direct electrocortical stimulation site mapping.** Each stimulation location reported in ref. 51 was separately mapped into the MNI-space Conte69 atlas pial cortical surface[80] by identifying the vertex with the minimal Euclidean distance to the stimulation site's MNI coordinates. Movements resulting from each site were classified as 'lower extremity', 'upper extremity' or 'face' and coloured accordingly (although no lower extremity movements were reported in the displayed left hemisphere).

### Reporting summary

Further information on research design is available in the Nature Portfolio Reporting Summary linked to this article.

### Data availability

Data from individual subjects participants 1–3 are available in the openneuro repository: https://openneuro.org/datasets/ds002766/versions/3.0.0. Data from the individual perinatal stroke subject are available in the openneuro repository: https://openneuro.org/datasets/ds004498/versions/1.0.0. Data from the UKB dataset used here are available at https://www.fmrib.ox.ac.uk/ukbiobank/. The ABCD data used in this report came from ABCD BIDS Community Collection (ABCC; NDA Collection 3165) and the Annual Release 2.0: https://doi.org/10.15154/1503209. Data from the HCP dataset used here is available at www.humanconnectome.org. Users must agree to data use terms for the HCP before being allowed access to the data and ConnectomeDB, details are provided at https://www.humanconnectome.org/study/hcp-young-adult/data-use-terms. Data from the WU120 dataset is available in the openneuro repository at https://openneuro.org/datasets/ds000243/versions/00001. Data from the PRIME-DE Oxford macaque dataset used in this report are available at https://fcon_1000.projects.nitrc.org/indi/PRIME/oxford.html. Users register with NITRC

and with the 1000 Functional Connectomes Project website on NITRC to gain access to the PRIME-DE datasets. Data from the UMN macaque will be publicly available via the PRIME-DE website (see above) by the end of 2023, after data collection of a larger sample is complete. Data from individual subjects participant 7 and 8, the individual neonate, infant and child participants, as well as those from the group average infant datasets, are available on reasonable request from C.J.L., E.M.G., J.R.P., C.M.S. and D.J.G. and C.D.S. They are not yet available through public databases because data collection is still ongoing. The DISTAL atlas is available from https://www.lead-dbs.org/helpsupport/knowledge-base/atlasesresources/distal-atlas/. The SUIT atlas is available from https://www.diedrichsenlab.org/imaging/suit.htm.

## Code availability

Task stimuli were presented using JSpsych toolbox v6.3, available from https://www.jspsych.org/6.3/. Data processing code for the ABCD data as well as for the child participant can be found at https://github.com/DCAN-Labs/abcd-hcp-pipeline. Data processing code for the HCP data can be found at https://github.com/Washington-University/HCPpipelines. Data processing code for participants 1–3 and 8, and the neonate, infant and perinatal stroke participants can be found at https://gitlab.com/DosenbachGreene/. Data processing code for participants 4–7 can be found at https://github.com/cjl2007/Liston-Laboratory-MultiEchofMRI-Pipeline. Data processing code for the macaque datasets can be found at https://github.com/DCAN-Labs/nhp-abcd-bids-pipeline. Code specific to the analyses in this manuscript can be found at https://gitlab.com/DosenbachGreene/SCAN/. Software packages incorporated into the above pipelines for data analysis included: Matlab R2020b, https://www.mathworks.com/; Connectome Workbench 1.5, http://www.humanconnectome.org/software/connectome-workbench.html; Freesurfer v6.2, https://surfer.nmr.mgh.harvard.edu/; FSL 6.0, https://fsl.fmrib.ox.ac.uk/fsl/fslwiki; 4dfp tools, https://4dfp.readthedocs.io/en/latest/; and Infomap, https://www.mapequation.org.

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

**Acknowledgements** The authors thank P. Strick for extensive comments and discussions that allowed critical conceptualization of results; M. Graziano for helpful comments and suggestions; and J. Sallet, R. Mars and M. Rushworth for sharing their data with the PRIMatE Data Exchange consortium. This work was supported by NIH grants NS110332 (D.J.N.), MH120989 (C.J.L.), MH100019 (N.A.S.), MH129493 (D.M.B.), MH113883 (C.E.R.), MH128177 (J.Z.), EB031765 (J.Z.), DA048742 (J.Z.), MH120194 (J.T.W.), NS123345 (B.P.K.), NS098482 (B.P.K.), MH121518 (S.M.), MH128696 (T.X.), NS124789 (S.A.N.), MH118370 (C.G.), MH118362 (J.R.P.), HD088125 (J.R.P.), HD055741 (J.R.P.), MH121462 (J.R.P.), MH116961 (J.R.P.), MH129426 (J.R.P.), HD103525, MH120194 (J.T.W.), MH122389 (C.M.S.), DA047851 (C.J.L.), MH118388 (C.J.L.), MH114976 (C.J.L.), MH129616 (T.O.L.), DA041148 (D.A.F.), DA04112 (D.A.F.), MH115357 (D.A.F.), MH096773 (D.A.F. and N.U.F.D.), MH122066 (E.M.G., D.A.F. and N.U.F.D.), MH121276 (E.M.G., D.A.F. and N.U.F.D.), MH124567 (E.M.G., D.A.F. and N.U.F.D.), NS129521 (E.M.G., D.A.F. and N.U.F.D.), and NS088590 (N.U.F.D.); by NSF grant CAREER BCS-2048066 (C.G.); by Center for Brain Research in Mood Disorders; by Eagles Autism Challenge; by the Dystonia Medical Research Foundation (S.A.N.); by the National Spasmodic Dysphonia Association (E.M.G. and S.A.N.); by the Taylor Family Foundation (C.M.S. and T.O.L.); by the Intellectual and Developmental Disabilities Research Center (D.J.G. and N.U.F.D.); by the Kiwanis Foundation (N.U.F.D.); by the Washington University Hope Center for Neurological Disorders (E.M.G., B.P.K. and N.U.F.D.); and by Mallinckrodt Institute of Radiology pilot funding (D.J.G., E.M.G. and N.U.F.D.).

**Author contributions** Conception: E.M.G. and N.U.F.D. Design: E.M.G., R.J.C., T.O.L. and N.U.F.D. Data acquisition, analysis and interpretation: E.M.G., R.J.C., A.N.V., A.R., A.N., D.J.N., C.J.L., N.A.S., S.R.K., K.M.S., J.M., R.L.M., A.M., D.F.M., A.Z., I.E., T.M., T.N., M.J.M., S.K., C.B.D., D.V.D., M.F., J.S.B.R., T.X., D.M.B., C.D.S., C.E.R., J.Z., K.N.B., J.R.P., J.T.W., P.B., J.S.S., B.P.K., S.M., S.A.N., C.G., C.M.S., J.D.P., C.L., D.J.G., J.L.R., S.E.P., M.E.R., T.O.L., D.A.F. and N.U.F.D. Manuscript writing and revision: E.M.G., D.J.N., D.M.B., S.M., S.A.N., C.G., C.M.S., J.D.P., D.J.G., S.E.P., M.E.R., T.O.L., D.A.F., and N.U.F.D. Participant 1 was author A.N; participant 2 was author N.U.F.D; participant 4 was author C.J.L; participant 5 was author J.D.P; and participant 8 was author E.M.G.

**Competing interests** D.A.F., N.U.F.D. and N.A.S. have a financial interest in Turing Medical Inc. and may benefit financially if the company is successful in marketing FIRMM motion monitoring software products. A.N.V., D.A.F. and N.U.F.D. may receive royalty income based on FIRMM technology developed at Washington University School of Medicine and Oregon Health and Sciences University and licensed to Turing Medical Inc. D.A.F. and N.U.F.D. are co-founders of Turing Medical Inc. These potential conflicts of interest have been reviewed and are managed by Washington University School of Medicine, Oregon Health and Sciences University and the University of Minnesota. N.A.S. is now an employee of Turing Medical Inc. C.M.S receives research support from Sage Therapeutics. C.L. is listed as an inventor for Cornell University patent applications on neuroimaging biomarkers for depression that are pending or in preparation. C.L. has served as a scientific advisor or consultant to Compass Pathways PLC, Delix Therapeutics, Magnus Medical and Brainify.AI. The other authors declare no competing interests.

**Additional information**
**Correspondence and requests for materials** should be addressed to Evan M. Gordon or Nico U. F. Dosenbach.

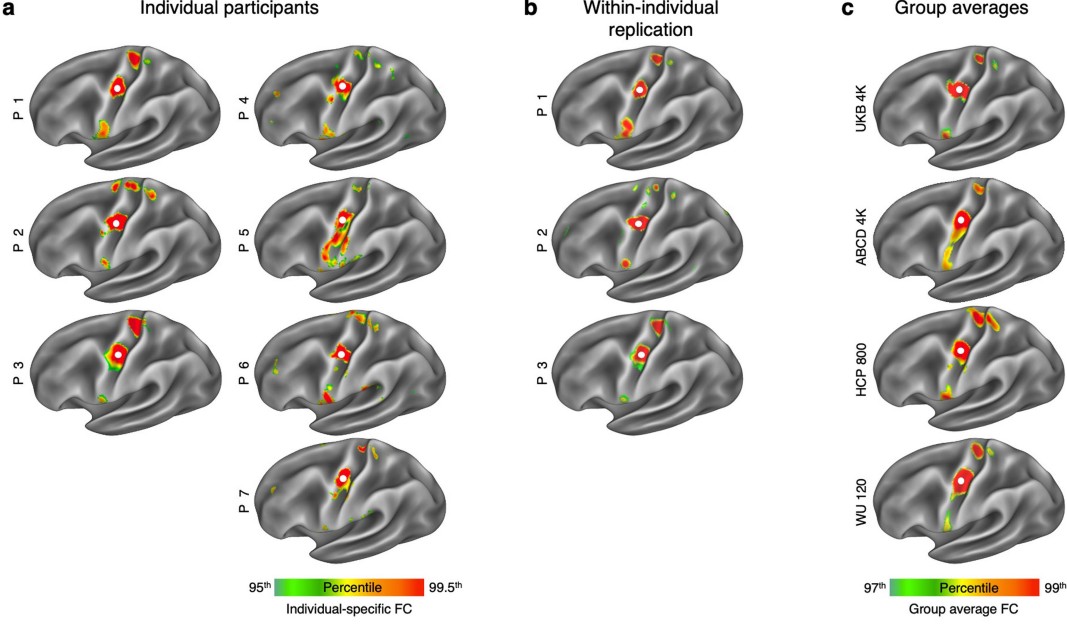

**a** Individual participants

P 1 P 4

P 2 P 5

P 3 P 6

P 7

95th Percentile 99.5th
Individual-specific FC

**b** Within-individual replication

P 1

P 2

P 3

**c** Group averages

UKB 4K

ABCD 4K

HCP 800

WU 120

97th Percentile 99th
Group average FC

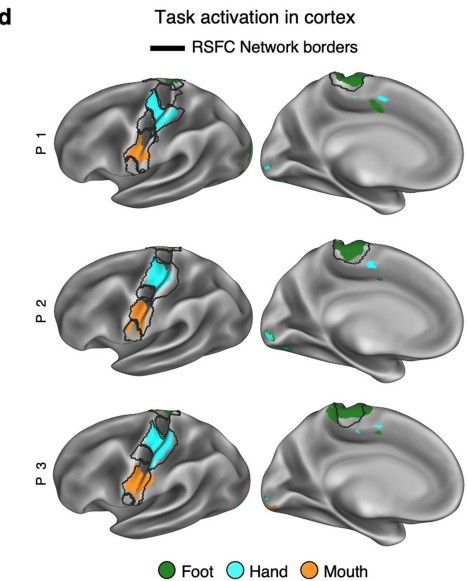

**d** Task activation in cortex

— RSFC Network borders

P 1

P 2

P 3

● Foot ● Hand ● Mouth

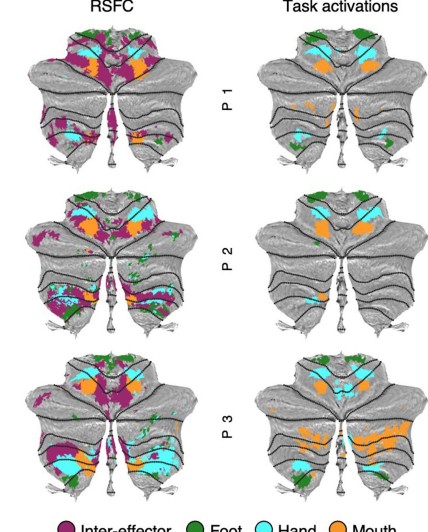

**e** Cerebellum connectivity and task activation

RSFC Task activations

P 1

P 2

P 3

● Inter-effector ● Foot ● Hand ● Mouth

**Extended Data Fig. 1 | Consistency of the inter-effector motif across datasets.** Connectivity patterns seeded from a continuous line down the left precentral gyrus revealed that the interleaved motor functional connectivity pattern was consistent across **a**, seven highly-sampled individual participants (172–356 min of data); **b**, replication data (416–1,114 min) collected in P1–P3; and **c**, multiple independent sets of group data averaged across cohorts of varying size. Here, functional connectivity is shown seeded from the middle inter-effector region for each individual participant and group-averaged dataset (see Supplementary Video 2 for all seeds). Thresholds for connectivity maps were scaled to the 95th percentile of map values in individuals, and to the 97th percentile of values in groups, to account for differences in data acquisition and processing strategies across datasets. **d**, Discrete functional networks were demarcated within each subject in M1 and S1 using a whole-brain, data-driven hierarchical approach applied to the resting-state fMRI data (see Fig. S7), which defined the spatial extent of the networks observed in Fig. 1 (black outlines). In P1-P3, regions defined by resting state functional connectivity (RSFC) were functionally labeled using a classic block-design fMRI motor task involving separate movement of the foot, hand, and tongue (following[31]; see[29] for details). The map illustrates the top 1% of vertices activated by movement of the foot (green), hand (cyan), and mouth (orange). **e**, Left: preferential connectivity of each motor division to the cerebellum. Right: activations during the fMRI motor task described in panel d. The map illustrates the top 5% of vertices within cerebellum active during movement of the foot (green), hand (cyan), and mouth (orange).

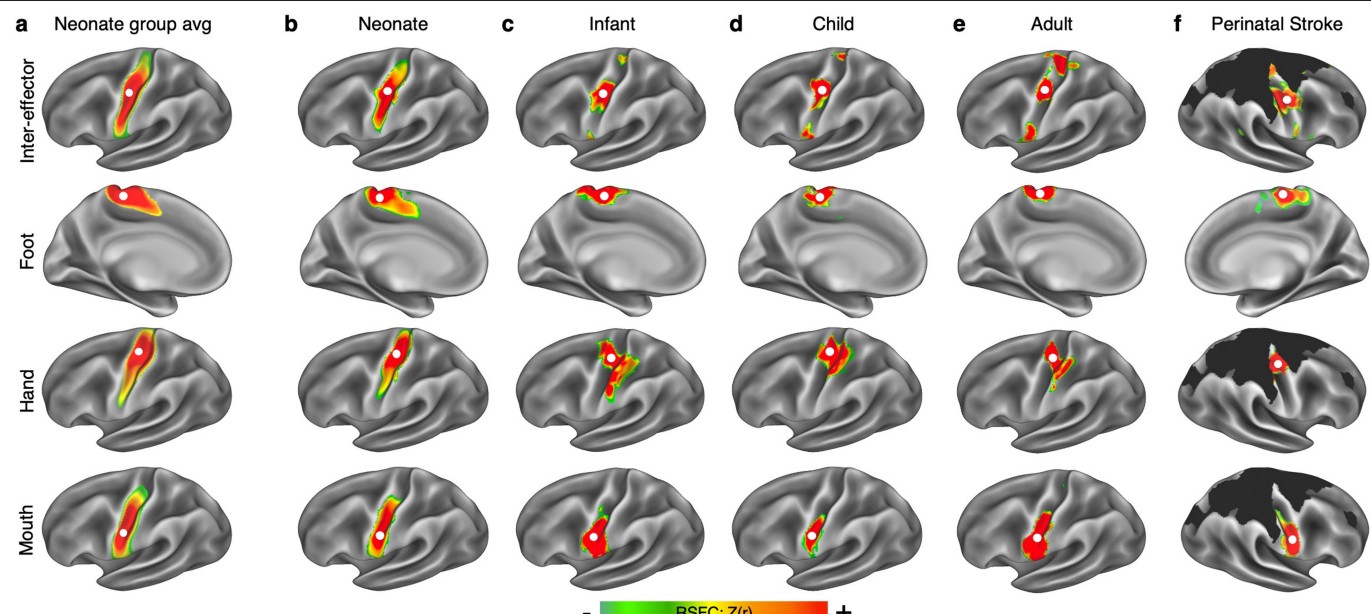

**Extended Data Fig. 2 | Motor cortex functional connectivity in pediatric participants and perinatal stroke.** Functional connectivity maps were seeded from a continuous line of points down precentral gyrus in fMRI data from **a**, data averaged across 262 human neonates, all scanned shortly after birth; **b**, a neonate scanned 13 days after birth; **c**, an 11-month old infant; **d**, a 9-year old child; **e**, adult participant P1 (from Fig. 1); and **f**, an adolescent who had experienced extensive cortical reorganization after severe bilateral perinatal strokes (destroyed cortex in black). Right hemisphere is shown in the stroke patient because left hemisphere M1 was entirely lost. Example seed maps shown here illustrate observed inter-effector (row 1) and effector-specific connectivity (rows 2-4). Inter-effector and effector-specific regions exhibited clear boundaries within M1 in the infant, child, the adults, and the stroke patient, but not in the neonates. Visualization thresholds varied between $Z(r) > 0.3$ and $Z(r) > 0.5$ across datasets due to differences in data collection and processing, as well as differences inherent to the populations.

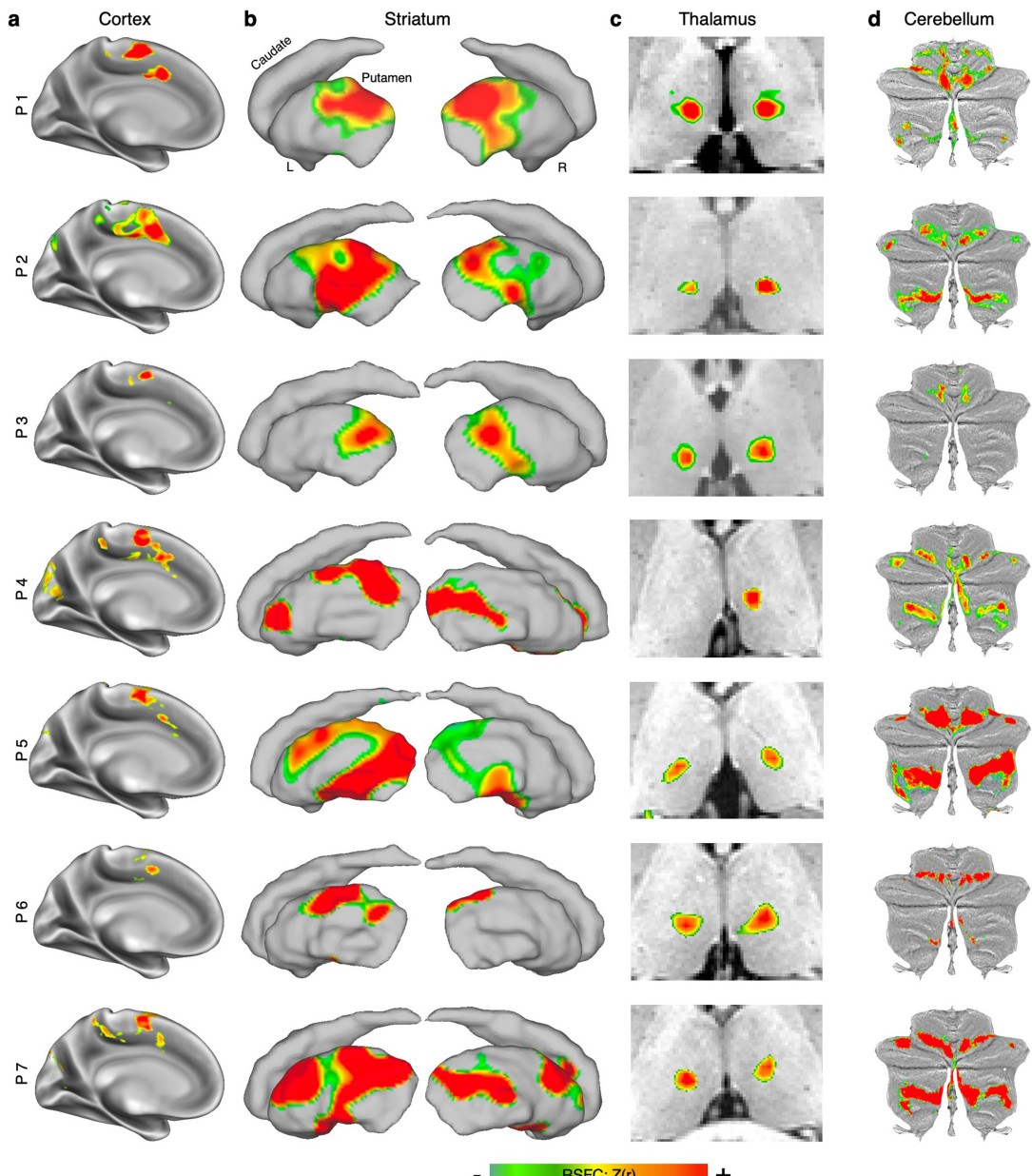

**Extended Data Fig. 3 | Whole brain functional connectivity of inter-effector motif across participants.** Brain regions with the strongest functional connectivity to the middle inter-effector region in **a**, medial cortex, **b**, striatum (lateral view of left and right striatum), **c**, thalamus (axial view), and **d**, cerebellum. Functional connectivity values are thresholded at $Z(r) > 0.35$ in cortex.

Subcortical functional connectivity values are thresholded at different levels in each subject due to variation in subcortical signal-to-noise ratios across individuals. Thresholds were chosen to illustrate the strongest subcortical connections. Specific thresholds shown here: P1 - $Z(r) > 0.15$; P3, 4, 6, 7 - $Z(r) > 0.1$; P2 - $Z(r) > 0.04$; P5 - $Z(r) > 0.03$.

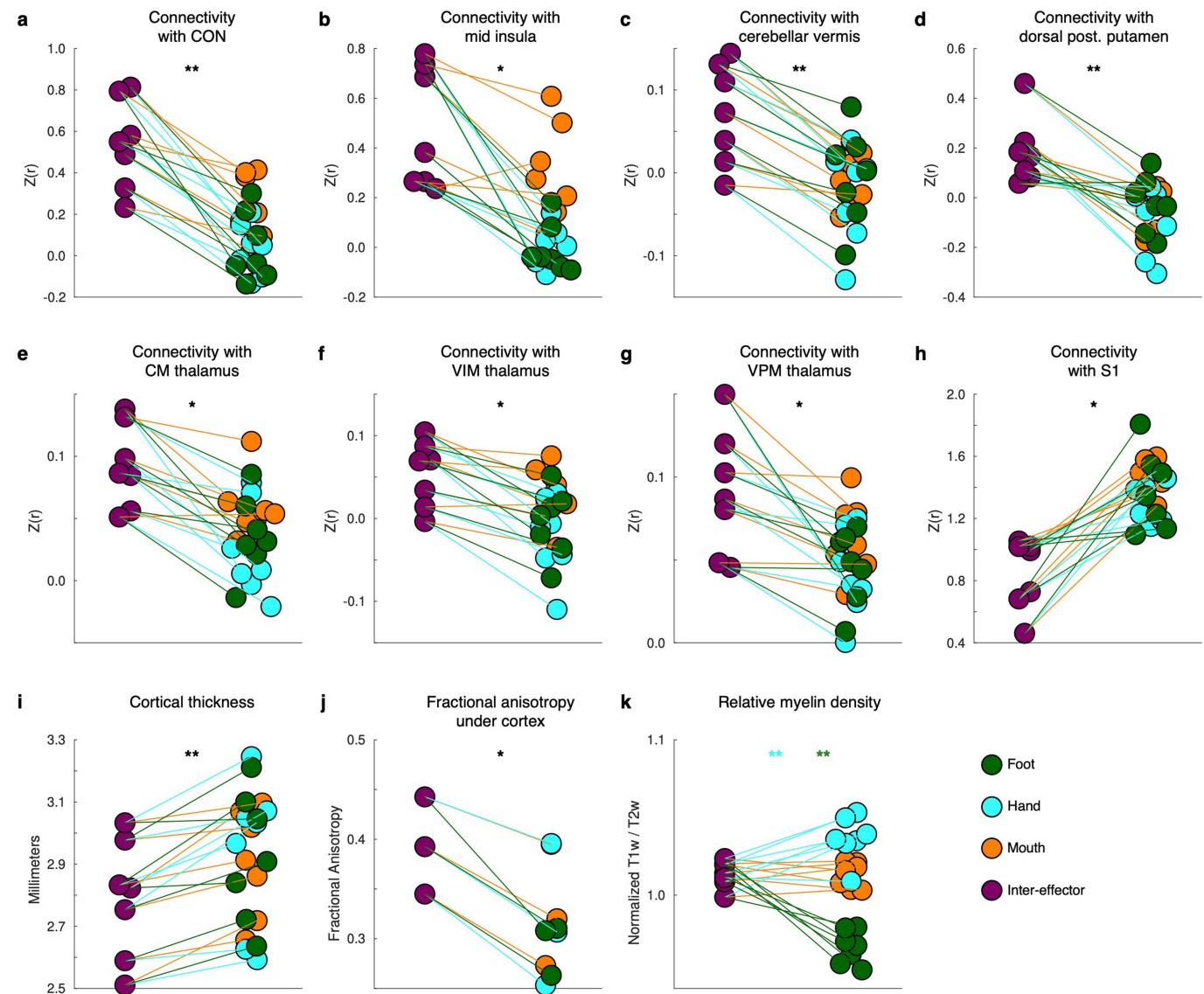

**Extended Data Fig. 4 | Functional connectivity and structural MRI metrics of motor cortex regions.** In each individual participant, measures derived from each of the foot, hand, mouth, and inter-effector motor regions. Colored lines connect the same participant's inter-effector and effector-specific regions for ease of comparison. **a**, Functional connectivity strength *Z(r)* between M1 region and individual-specific Cingulo-Opercular Network (CON). **b**, Functional connectivity between M1 region and middle insula. **c**, Functional connectivity with Lobule VIIIa vermis of the cerebellum. **d**, Functional connectivity between M1 region and dorsal posterior putamen. **e–g**, Functional connectivity

between M1 region and nuclei of the thalamus: **e**, Centromedian nucleus; **f**, Ventral Intermediate nucleus; **g**, Ventral Posteromedial nucleus. **h**, Functional connectivity between M1 region and adjacent postcentral gyrus (S1). **i**, Cortical thickness in M1 region. **j**, Fractional Anisotropy within 2 mm below cortex under M1 region. **k**, Intracortical myelin, indexed by the T1/T2 ratio and normalized across cortex, within cortex of M1 region. All significance values reflect significance across three two-sided paired t-tests (inter-effector vs foot, vs hand, and vs mouth). * $P < 0.05$; ** $P < 0.01$; *** $P < 0.001$, FDR-corrected.

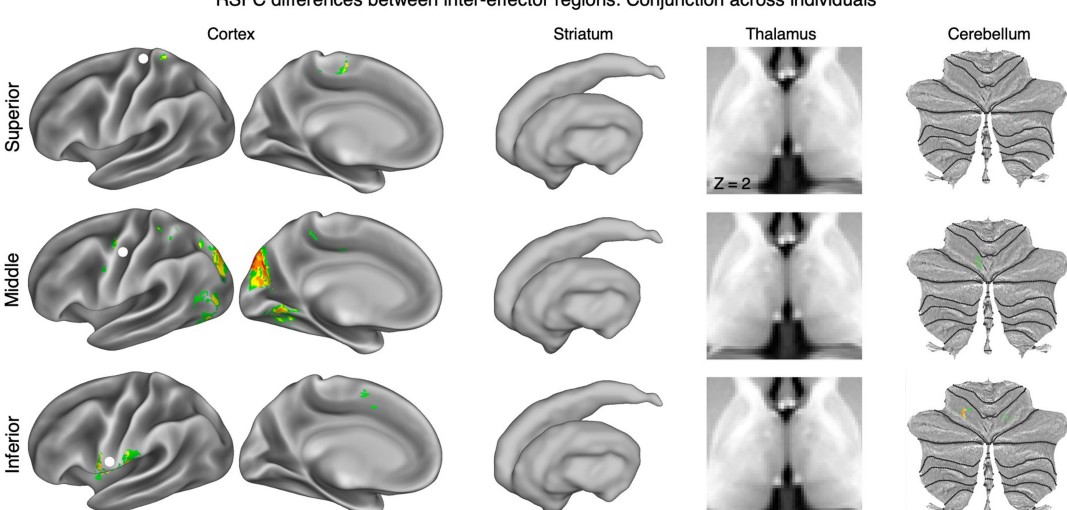

RSFC differences between inter-effector regions: Conjunction across individuals

Cortex    Striatum    Thalamus    Cerebellum

4/7 ▮▮▮ 7/7
Participants with region-preferential FC

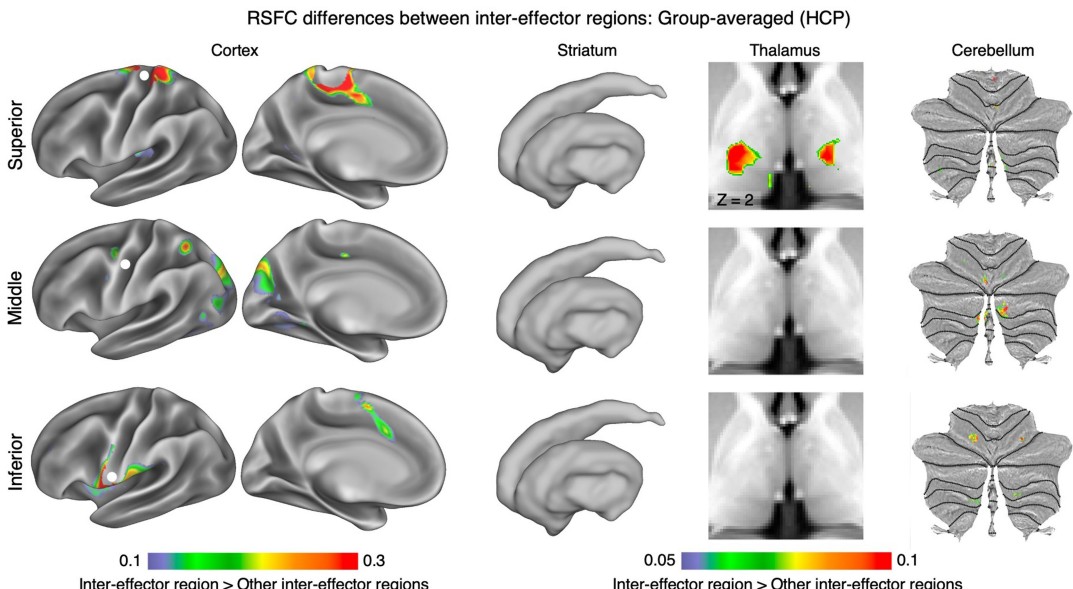

RSFC differences between inter-effector regions: Group-averaged (HCP)

Cortex    Striatum    Thalamus    Cerebellum

0.1 ▮▮▮ 0.3
Inter-effector region > Other inter-effector regions
Δ functional connectivity Z(r)

0.05 ▮▮▮ 0.1
Inter-effector region > Other inter-effector regions
Δ functional connectivity Z(r)

**Extended Data Fig. 5 | Differences in functional connectivity between inter-effector regions.** Brain regions more strongly connected to the superior inter-effector region than to either of the other two (top row); relatively most strongly connected to the middle inter-effector region (middle row); and relatively most strongly connected to the inferior inter-effector region, in cortex (left), striatum, thalamus, and cerebellum (right), **a**, in at least 50% of individuals (*n* = 7) and **b**, in group-averaged data from the Human Connectome Project (HCP; *n* = 812). Thresholds used are the same as in Fig. 2b. Note that central sulcus regions are masked as they exhibit large differences by definition. See Fig. S3 for all individual participants.

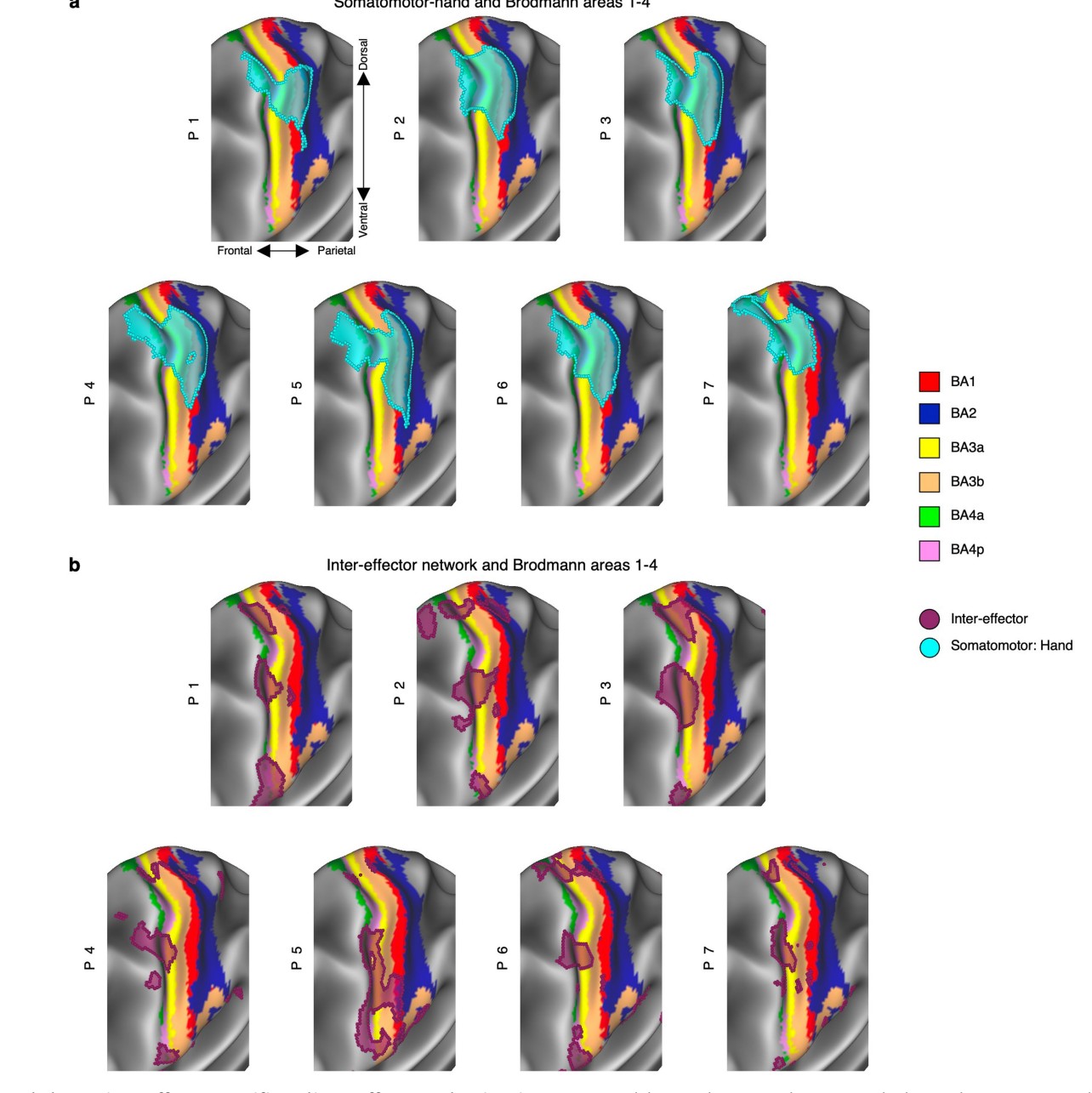

**Extended Data Fig. 6 | Effector-specific and inter-effector and regions in pre- and postcentral gyrus.** In every participant, Brodmann Areas (BAs) in M1 (BAs 4a, 4p) and S1 (BAs 1, 2, 3a, 3b) are displayed on the cerebral cortex, tilted around the Y- and Z-axes to show S1. Overlaid are **a**, the somatomotor-hand region, and **b**, the inter-effector regions.

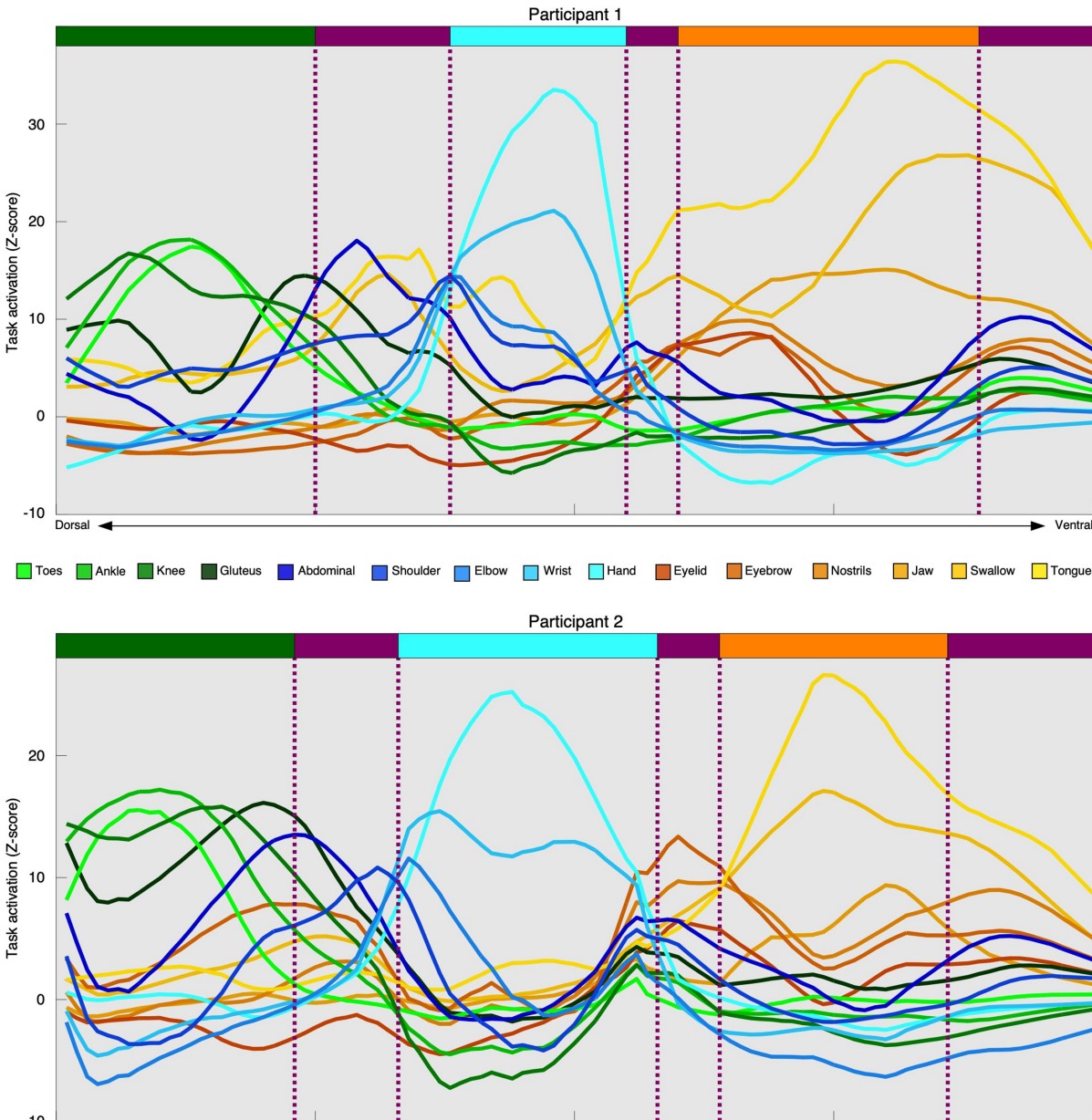

**Extended Data Fig. 7 | Primary motor cortex activation profiles for movement task battery.** In two participants (top, bottom), LOWESS curves were fit to the task activation profiles at each dorsal-ventral point in M1, for each separate movement (colored lines). Colored blocks (top) show the effector-specific foot (green), hand (cyan), and mouth (orange) areas of M1, as well as the inter-effector regions (maroon); dotted maroon lines show the boundaries between regions. The centers of effector-specific regions are characterized by strong activations for movements of the most distal body parts, and deactivations for all other movements. Inter-effector regions, by contrast, exhibited moderate activations for most movements.

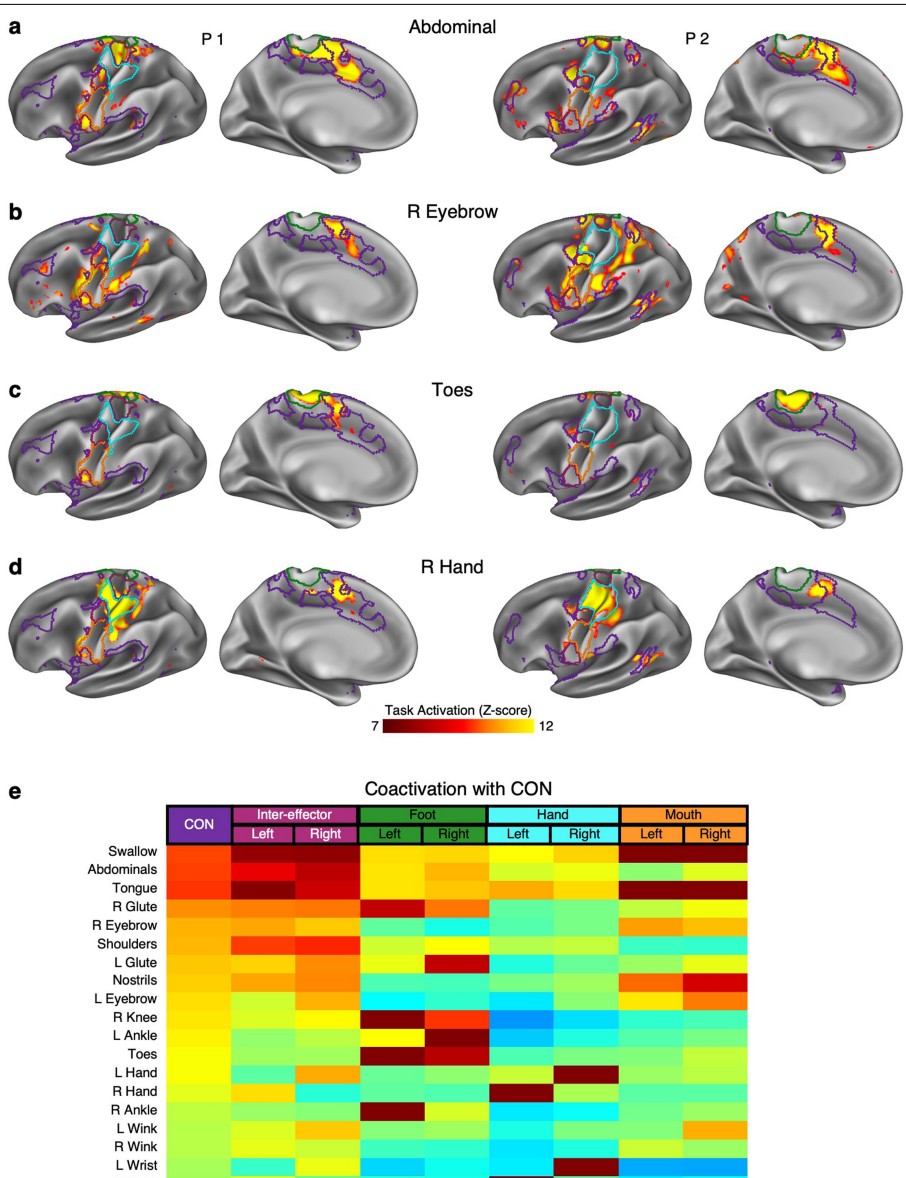

**Extended Data Fig. 8 | Effector-specificity of task fMRI activations.** In each participant, in the **a**, abdominal flexure task and the **b**, eyebrow raising task, the inter-effector regions and cingulo-opercular network (CON) were active. By contrast, in **c**, toe and **d**, hand motion tasks, activation was much more specific to a single region of somatomotor cortex. **e**, Across tasks, the degree of CON activation was consistently similar to the activation of the inter-effector regions (correlation between CON and inter-effector activations: all Pearson's $r > 0.81$, $P < 10^{-5}$, FDR corrected), but not consistently to hand (CON vs hand: Pearson's $r > 0.05$, $P < 0.82$) or foot (CON vs foot: Pearson's $r > 0.33$, $P < 0.13$) regions, and more weakly to mouth regions (CON vs mouth: Pearson's $r > 0.61$, $P < 0.003$). Illustrated activation values are averaged across participants and ordered based on CON activation.

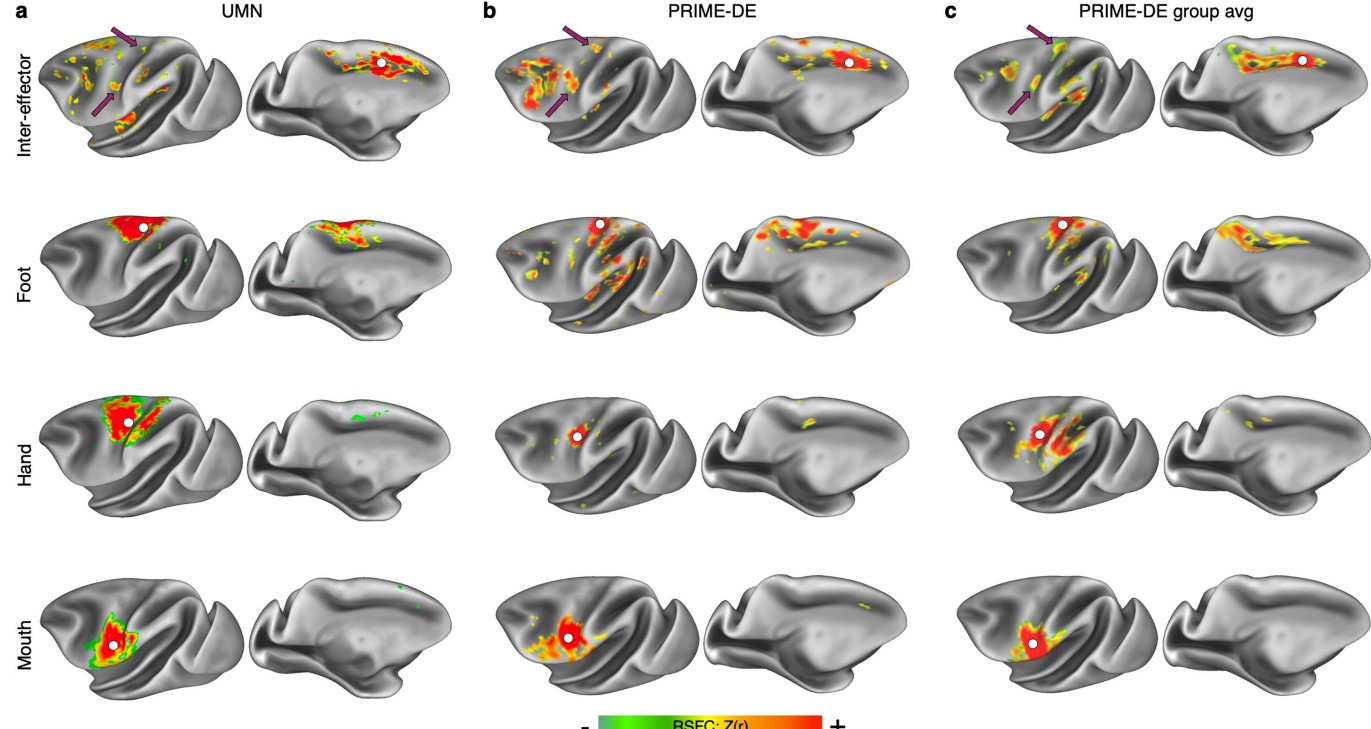

**Extended Data Fig. 9 | Motor Cortex functional connectivity in non-human primates.** Functional connectivity maps were seeded from dorsal anterior cingulate cortex (top row), as well as from a continuous line of points down anterior central sulcus (rows 2-4), in fMRI data from **a**, an individual macaque scanned for 77 min on a 10.5T MRI scanner; **b**, an individual macaque scanned for 53 min on a 3T scanner; and **c**, group-averaged data from eight macaques each scanned for 53 min on a 3T scanner. The dorsal anterior cingulate seed demonstrated connectivity to frontal, insular, and parietal regions homologous with the human CON, as well as with two regions in anterior central sulcus (maroon arrows). These central sulcus regions are thought to correspond to areas that project to internal organs[10] and represent possible macaque homologues of the inter-effector regions. The central sulcus seeds demonstrated connectivity patterns corresponding to the known functional divisions between M1 regions representing the foot (second row), hand (third row), and face (bottom row).

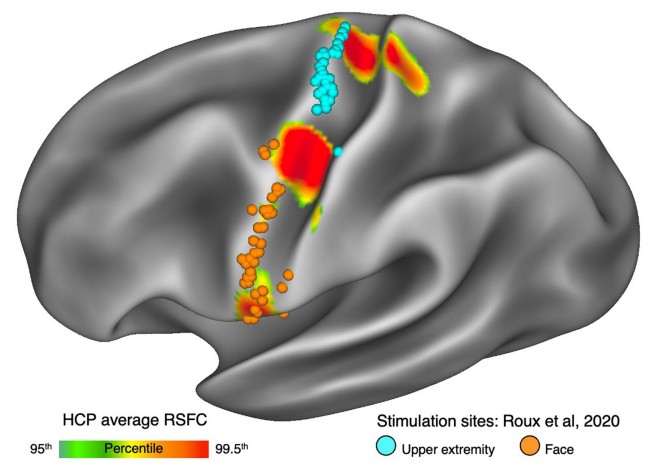

HCP average RSFC

95th [Percentile] 99.5th

Stimulation sites: Roux et al, 2020

◯ Upper extremity  ◯ Face

**Extended Data Fig. 10 | Somato-Cognitive Action Network regions in human cortical surface stimulation data.** The map of somato-cognitive action network (SCAN) regions was compared with published movements evoked by direct cortical surface stimulation[51]. Cortical map: functional connectivity is shown seeded from the middle SCAN region and averaged across all subjects in the HCP dataset ($n$ = 812; see also Extended Data Fig. 1c). Stimulation locations: MNI coordinates of surface stimulation location, and the resulting evoked movement, from 100 patients undergoing awake surgical brain mapping were reported in[51]. Each stimulation location evoking movement was mapped to the nearest cortical vertex on a group-averaged pial surface. Stimulation sites are colored according to whether they evoked facial movements (orange) or upper extremity movements (cyan). Stimulation sites evoking movement did not overlap with the central inter-effector region.

# Reporting Summary

## Statistics

For all statistical analyses, confirm that the following items are present in the figure legend, table legend, main text, or Methods section.

| n/a | Confirmed | |
|---|---|---|
| ☐ | ☒ | The exact sample size (*n*) for each experimental group/condition, given as a discrete number and unit of measurement |
| ☐ | ☒ | A statement on whether measurements were taken from distinct samples or whether the same sample was measured repeatedly |
| ☐ | ☒ | The statistical test(s) used AND whether they are one- or two-sided *Only common tests should be described solely by name; describe more complex techniques in the Methods section.* |
| ☒ | ☐ | A description of all covariates tested |
| ☐ | ☒ | A description of any assumptions or corrections, such as tests of normality and adjustment for multiple comparisons |
| ☐ | ☒ | A full description of the statistical parameters including central tendency (e.g. means) or other basic estimates (e.g. regression coefficient) AND variation (e.g. standard deviation) or associated estimates of uncertainty (e.g. confidence intervals) |
| ☐ | ☒ | For null hypothesis testing, the test statistic (e.g. *F*, *t*, *r*) with confidence intervals, effect sizes, degrees of freedom and *P* value noted *Give P values as exact values whenever suitable.* |
| ☒ | ☐ | For Bayesian analysis, information on the choice of priors and Markov chain Monte Carlo settings |
| ☒ | ☐ | For hierarchical and complex designs, identification of the appropriate level for tests and full reporting of outcomes |
| ☐ | ☒ | Estimates of effect sizes (e.g. Cohen's *d*, Pearson's *r*), indicating how they were calculated |

*Our web collection on statistics for biologists contains articles on many of the points above.*

## Software and code

Policy information about availability of computer code

| Data collection | Task stimuli were presented using JSpsych toolbox v6.3, available from https://www.jspsych.org/6.3/ |
|---|---|
| Data analysis | Data processing code for the ABCD data, as well as for the Child subject, can be found here: https://github.com/DCAN-Labs/abcd-hcp-pipeline <br> Data processing code for the HCP data can be found here: https://github.com/Washington-University/HCPpipelines <br> Data processing code for P01-03, P08, the Neonate, Infant, and Perinatal Stroke subjects can be found here: https://gitlab.com/DosenbachGreene/ <br> Data processing code for P04-07 can be found here: https://github.com/cjl2007/Liston-Laboratory-MultiEchofMRI-Pipeline <br> Data processing code for the Macaque datasets can be found here: https://github.com/DCAN-Labs/nhp-abcd-bids-pipeline <br> Code specific to the analyses in this manuscript can be found here: https://gitlab.com/DosenbachGreene/SCAN/ <br><br> Software packages incorporated into the above pipelines for data analysis included: <br> Matlab R2020b, https://www.mathworks.com/ <br> Connectome Workbench 1.5, http://www.humanconnectome.org/software/connectome-workbench.html <br> Freesurfer v6.2, https://surfer.nmr.mgh.harvard.edu/ <br> FSL 6.0, https://fsl.fmrib.ox.ac.uk/fsl/fslwiki <br> 4dfp tools, https://4dfp.readthedocs.io/en/latest/ <br> Infomap, www.mapequation.org |

For manuscripts utilizing custom algorithms or software that are central to the research but not yet described in published literature, software must be made available to editors and reviewers. We strongly encourage code deposition in a community repository (e.g. GitHub). See the Nature Portfolio guidelines for submitting code & software for further information.

# Data

Policy information about availability of data

All manuscripts must include a data availability statement. This statement should provide the following information, where applicable:
- Accession codes, unique identifiers, or web links for publicly available datasets
- A description of any restrictions on data availability
- For clinical datasets or third party data, please ensure that the statement adheres to our policy

Data from individual subjects P01-P03 is available in the openneuro repository here: https://openneuro.org/datasets/ds002766/versions/3.0.0

Data from the individual Perinatal Stroke subject is available in the openneuro repository: https://openneuro.org/datasets/ds004498/versions/1.0.0.

Data from the UKB dataset used here is available at https://www.fmrib.ox.ac.uk/ukbiobank/

The ABCD data used in this report came from ABCD collection 3165 and the Annual Release 2.0, DOI 10.15154/1503209.

Data from the HCP dataset used here is available at www.humanconnectome.org. Users must agree to data use terms for the HCP before being allowed access to the data and ConnectomeDB, details are provided at https://www.humanconnectome.org/study/hcp-young-adult/data-use-terms.

Data from the WU120 dataset is available in the openneuro repository here: https://openneuro.org/datasets/ds000243/versions/00001

Data from the PRIME-DE Oxford macaque dataset used in this report are available at https://fcon_1000.projects.nitrc.org/indi/PRIME/oxford.html. Users register with NITRC and with the 1000 Functional Connectomes Project website on NITRC to gain access to the PRIME-DE datasets.

Data from the UMN macaque will be publicly available via the PRIME-DE website (see above) by the end of 2023, after data collection of a larger sample is complete.

Data from individual subjects P04-P07, P08, the individual Neonate, Infant, and Child subjects, as well as from the group average Infant datasets, are available on reasonable request from authors CL, EMG, JRP, CMS, and DJG, and CDS, respectively. They are not yet available through public databases, because data collection is still ongoing.

The DISTAL atlas is available from https://www.lead-dbs.org/helpsupport/knowledge-base/atlasesresources/distal-atlas/
The SUIT atlas is available from https://www.diedrichsenlab.org/imaging/suit.htm

# Human research participants

Policy information about studies involving human research participants and Sex and Gender in Research.

| Reporting on sex and gender | Findings apply to all studied individuals and groups, regardless of sex. |
|---|---|
| | Sex ratios: |
| | P01-03, P08: 3M 1F |
| | P04-07: 4M |
| | Neonate: 1M |
| | Infant: 1M |
| | Child: 1 M |
| | Perinatal Stroke: 1 M |
| | UKB: 47%M, 53%F |
| | ABCD: 49%M, 51%F |
| | HCP: 402M, 410F |
| | WU120: 60M, 60F |
| | eLABE: 141M, 121F |

| Population characteristics | Age ranges: |
|---|---|
| | |
| | P01-03, P08: 25-40y |
| | P04-07: 24-38y |
| | Neonate: 13 days |
| | eLABE: 1-3 weeks |
| | Infant: 11 months |
| | Child: 9y |
| | Perinatal Stroke: 13y |
| | ABCD: 9-10y |
| | HCP: 22-35 |
| | UKB: 40-69 |
| | WU120: 19-32y |

| Recruitment | P01-03, P08, WashU120: Healthy adult participants were recruited from the Washington University community via flyers and word of mouth. |
|---|---|

P04-07: Healthy adult participants were recruited from the Weill Cornell Medical School community via word of mouth.

Infant, Child: Parents were recruited from the St. Louis community via flyers and word of mouth.

Perinatal Stroke: the participant was referred to the neurology clinic of author NUFD (St Louis Children's Hospital) because of noted clumsiness of his right hand.

Neonate and group-averaged Neonate: Mothers were recruited during the 2nd or 3rd trimester from two obstetrics clinics at Washington University.

HCP: Sampling 300–400 young adult sibships of average size 3–4, with most of these sibships including a MZ or DZ twin pair.

ABCD: A very important motivation for the ABCD study is that its sample should reflect, as best as possible, the sociodemographic variation of the US population.The ABCD cohort recruitment emulates a multi-stage probability sample of eligible children: A nationally distributed set of 21 primary stage study sites, a probability sampling of schools within the defined catchment areas for each site, and recruitment of eligible children in each sample school. The major departure from traditional probability sampling of U.S. children originates in how participating neuroimaging sites were chosen for the study. Although the 21 ABCD study sites are well-distributed nationally the selection of collaborating sites is not a true probability sample of primary sampling units (PSUs) but was constrained by the grant review selection process and the requirement that selected locations have both the research expertise and the neuroimaging equipment needed for the study protocol.

UKB: Participants were assessed between 2006 and 2010 in 22 assessment centres throughout the UK, covering a variety of different settings to provide socioeconomic and ethnic heterogeneity and urban–rural mix. This ensured a broad distribution across all exposures to allow the reliable detection of generalisable associations between baseline characteristics and health outcomes.

Summary: While each individual dataset represented here may contain selection biases, particularly including socio-demographic factors, the replication of findings across all datasets provides confidence that the findings do not depend on these factors.

**Ethics oversight**

P01-03, P08, Neonate, Infant, Child, Perinatal Stroke, HCP, WU120, and group average Neonate datasets: The study was approved by the Washington University School of Medicine Human Studies Committee and Institutional Review Board.

P04-07: The study was approved by the Weill Cornell Medicine Institutional Review Board.

ABCD: The ABCD Study obtained centralized institutional review board approval from the University of California, San Diego, and each of the 21 study sites obtained local institutional review board approval.

UKB: Ethical procedures are controlled by a dedicated Ethics and Guidance Council (http://www.ukbiobank.ac.uk/ethics) that has developed with UK Biobank an Ethics and Governance Framework (given in full at http://www.ukbiobank.ac.uk/wp-content/uploads/2011/05/EGF20082.pdf), with IRB approval also obtained from the North West Multi-center Research Ethics Committee.

Note that full information on the approval of the study protocol must also be provided in the manuscript.

# Field-specific reporting

Please select the one below that is the best fit for your research. If you are not sure, read the appropriate sections before making your selection.

☒ Life sciences  ☐ Behavioural & social sciences  ☐ Ecological, evolutionary & environmental sciences

For a reference copy of the document with all sections, see nature.com/documents/nr-reporting-summary-flat.pdf

# Life sciences study design

All studies must disclose on these points even when the disclosure is negative.

**Sample size**

Much of this study was focused on within- rather than across-individual analysis. In these analyses, the relevant factor is ensuring that enough data is collected from each participant to ensure reliable measures. We have previously shown that this requires a minimum of 30 minutes of data per individual, and that reliability continuously improves as more per-individual data is collected (Laumann et al., 2015; Gordon et al., 2017). Therefore we ensured that all participants tested at the individual level had at least that much data, and that in many cases the data quantities were much higher.

For analyses of group-averaged data, we always employed the maximum number of participants available in public datasets.

**Data exclusions**

ABCD: 3,928 participants were selected as the participants with at least 8 minutes of low-motion data, a pre-established criterion.

HCP: 812 participants were selected who completed four 15-minute resting-state fMRI runs and who had their raw data reconstructed using the newer "recon 2" software, a pre-established criterion.

PRIME-DE: Each animal's data was closely visually inspected for quality. Following these inspections, data from eleven animals were excluded

due to the presence of artifact in or near the central sulcus, leaving eight animals in the final data. This criterion was not pre-established but was necessary given the observation of artifact in some macaques.

| | |
|---|---|
| Replication | Experimental findings were replicated in independent data collected from all human individuals (aside from the neonate data) and group-averaged datasets, as well as within-participant (i.e. by repeats) in P01-03. |
| Randomization | N/A: There were no separate experimental groups. |
| Blinding | N/A: There were no separate experimental groups. |

# Reporting for specific materials, systems and methods

We require information from authors about some types of materials, experimental systems and methods used in many studies. Here, indicate whether each material, system or method listed is relevant to your study. If you are not sure if a list item applies to your research, read the appropriate section before selecting a response.

## Materials & experimental systems

| n/a | Involved in the study |
|---|---|
| ☒ | Antibodies |
| ☒ | Eukaryotic cell lines |
| ☒ | Palaeontology and archaeology |
| ☐ | ☒ Animals and other organisms |
| ☒ | Clinical data |
| ☒ | Dual use research of concern |

## Methods

| n/a | Involved in the study |
|---|---|
| ☒ | ChIP-seq |
| ☒ | Flow cytometry |
| ☐ | ☒ MRI-based neuroimaging |

## Animals and other research organisms

Policy information about studies involving animals; ARRIVE guidelines recommended for reporting animal research, and Sex and Gender in Research

| | |
|---|---|
| Laboratory animals | The study included one adult female macaca fascicularis (age 6 y) and eight adult male macaca mulatta (age 4.1 +/- 0.98 years) |
| Wild animals | The study did not include wild animals |
| Reporting on sex | The UMN macaque was female and the PRIME-DE macaques were male. Sex was not considered as a factor here because it was confounded with site. |
| Field-collected samples | The study did not include samples collected from the field |
| Ethics oversight | UMN macaque: Experimental procedures were carried out in accordance with the University of Minnesota Institutional Animal Care and Use Committee and the National Institute of Health standards for the care and use of non-human primates.

PRIME-DE macaques: Protocols for animal care, magnetic resonance imaging, and anaesthesia were carried out under authority of personal and project licenses in accordance with the UK Animals (Scientific Procedures) Act 1986. |

Note that full information on the approval of the study protocol must also be provided in the manuscript.

## Magnetic resonance imaging

### Experimental design

| | |
|---|---|
| Design type | Resting state, block design task, and event-related task |
| Design specifications | Resting state: 172 – 1,813 total minutes of data acquired per participant

Block design movement task battery: 10 15.4-second movement blocks across five conditions plus 3 15.4-second rest blocks per run; 64 runs per subject.

Block design laryngeal mapping task: 12 15.0-second movement blocks across six conditions plus 2 15.0-second rest blocks per run; 10 runs per subject.

Even related action control task: . The participant is cued to prepare the movement(s) when they see one or two movement symbols placed on a body shape in a grey color (planning phase), and is then cued to execute the movement(s) when the grey symbol or symbols turn green (execution phase). Using a pseudorandom jitter, the planning phase can last from 2 to 6.5s followed by 4 to 8.5s of movement execution. Each movement trial (planning and |

execution) is followed by a jittered fixation of up to 5s. A rest block of 8.6s is implemented every 12 movements. 48 trials were collected in each run; twelve runs were acquired in each participant.

Behavioral performance measures | Behavioral outputs were not recorded.

## Acquisition

Imaging type(s) | Structural (T1-w and T2w), Diffusion, Functional

Field strength | 3.0T

Sequence & imaging parameters

P01, P03, P08: A high-resolution T1-weighted MP-RAGE (TE=2.22ms, TR=2400ms, flip angle=8°, 208 slices with 0.8x0.8x0.8mm voxels) and a T2-weighted spin-echo image (TE=563ms, TR=3200ms, flip angle=120°, 208 slices with 0.8x0.8x0.8mm voxels) were collected.

P02: four T1-weighted images (sagittal, 224 slices, 0.8 mm isotropic resolution, TE=3.74 ms, TR=2400 ms, TI=1000 ms, flip angle = 8°) and four T2-weighted images (sagittal, 224 slices, 0.8 mm isotropic resolution, TE=479 ms, TR=3200 ms). P01-03: fMRI scans were collected as a blood oxygen level-dependent (BOLD) contrast sensitive gradient echo-planar sequence (TE=33ms, flip angle=84°, resolution=2.6 mm isotropic, TR=1100ms, multiband 4 acceleration). Diffusion imaging was collected using a single-shot echo planar diffusion-weighted sequence consisting of 75 contiguous axial slices, isotropic (2x2x2 mm^3) resolution, TR/TE 3500/83 ms, four shells (b-values 0.25, 0.5, 1.0, and 1.5 ms/mm2) and 96 encoding directions.

P01, P02 (laryngeal mapping fMRI task), P08 (all fMRI): fMRI scans were collected as a multi-band five-echo blood oxygen level-dependent (BOLD) contrast sensitive gradient echo-planar sequence (flip angle = 68°, resolution = 2.0 mm isotropic, TR = 1761ms, multiband 6 acceleration, TE1: 14.20 ms, TE2: 38.93 ms, TE3: 63.66 ms, TE4: 88.39 ms, and TE5: 113.12 ms)

P04-07: High-resolution T1-weighted MP-RAGE images (TE=2.28ms, TR=2400ms, flip angle=90°, 208 slices with 0.8x0.8x0.8mm voxels) and T2-weighted spin-echo images (TE=563ms, TR=3200ms, flip angle=8°, 208 slices with 0.8x0.8x0.8mm voxels) were acquired. A 14.5 minute long multi-echo resting-state fMRI scan was collected as a five-echo blood oxygen level-dependent (BOLD) contrast sensitive gradient echo-planar sequence (flip angle=68°, resolution=2.4 mm isotropic, TR=1355ms, multiband 6 acceleration, TE1: 13.40 ms, TE2: 31.11 ms, TE3: 48.82 ms, TE4: 66.53 ms, and TE5: 84.24 ms).

Neonate: A high-resolution T2-weighted spin-echo image was collected (TE=563ms, TR=3200ms, flip angle=120°, 208 slices with 0.8x0.8x0.8mm voxels). Multi-echo resting-state fMRI runs were collected as a five-echo blood oxygen level-dependent (BOLD) contrast sensitive gradient echo-planar sequence (flip angle=68°, resolution=2.0 mm isotropic, TR=1761ms, multiband 6 acceleration, TE1: 14.20 ms, TE2: 38.93 ms, TE3: 63.66 ms, TE4: 88.39 ms, and TE5: 113.12 ms).

Infant: a high-resolution T1-weighted MP-RAGE (TE=2.24ms, TR=2400ms, flip angle=8°, 208 slices with 0.8x0.8x0.8mm voxels) and a T2-weighted spin-echo image (TE=564ms, TR=3200ms, flip angle=120°, 208 slices with 0.8x0.8x0.8mm voxels) were collected. Resting-state fMRI was collected as a blood oxygen level-dependent (BOLD) contrast sensitive gradient echo-planar sequence (flip angle=52°, resolution=3.0 mm isotropic, TE=30ms, TR=861ms, multiband 4 acceleration).

Child: High-resolution T1-weighted MP-RAGE images (TE=2.90ms, TR=2500ms, flip angle=8°, 176 slices with 1mm isotropic voxels), and T2-weighted spin-echo images (TE=564ms, TR=3200ms, flip angle=120°, 176 slices with 1mm isotropic voxels) were collected. Resting-state fMRI was collected as a blood oxygen level-dependent (BOLD) contrast sensitive gradient echo-planar sequence (flip angle=84°, resolution=2.6mm isotropic, 56 slices, TE=33ms, TR=1100ms, multiband 4 acceleration).

Perinatal stroke: T1-weighted images (sagittal, 224 slices, 0.8 mm isotropic resolution, TE=3.74 ms, TR=2400 ms, TI=1000 ms, flip angle = 8 degrees), and T2-weighted images (sagittal, 224 slices, 0.8 mm isotropic resolution, TE=479 ms, TR=3200 ms) were collected. Resting state fMRI data were collected using a gradient-echo EPI sequence (TR = 2.2 s, TE = 27 ms, flip angle = 90°, voxel size = 4 mm x 4 mm x 4 mm, 36 slices).

UMN macaque: A T1 weighted MP-RAGE was acquired (TR = 3300 ms, TE = 3.56 ms, TI = 1140, flip angle = 5°, slices = 256, matrix = 320×260, acquisition voxel size = 0.5 × 0.5 × 0.5 mm 3, in-plane acceleration GRAPPA = 2). A resolution and FOV matched T2 weighted 3D turbo spin-echo sequence was also acquired. FMRI timeseries, each consisting of 700 continuous 2D multiband EPI89–91 functional volumes (TR = 1110ms; TE = 17.6 ms; flip angle = 60°, slices = 58, matrix = 108×154; FOV = 81×115.5 mm ; acquisition voxel size = 0.75 × 0.75 × 0.75 mm) were acquired with a left-right phase encoding direction using in plane acceleration factor GRAPPA = 3, partial Fourier = 7/8th, and MB factor = 2.

PRIME-DE macaques: A T1-weighted MPRAGE sequence was used to acquire anatomical data (TR = 2500 ms, TE = 4.01 ms, TI = 1100, flip angle = 8°, acquisition voxel size = 0.5 × 0.5 × 0.5 mm, 128 slices). Festing-state fMRI data was acquired at a 2.0 mm isotropic voxel resolution (TR = 2000 ms, TE = 19 ms, Flip angle = 90°).

UKB: T1 - 1.0x1.0x1.0 mm, 208x256x256 3D MPRAGE, sagittal, R=2, TI/TR=880/2000 ms; T2 FLAIR - 1.05x1.0x1.0 mm 192x256x256 FLAIR, 3D SPACE, sagittal, R=2, PF 7/8, fat sat, TI/TR=1800/5000 ms, elliptical; resting-state fMRI - 2.4x2.4x2.4 mm, 88x88x64 TE/TR=39/735 ms, MB=8, R=1, flip angle 52°, fat sat.

HCP: T1w images are acquired using a 3D MPRAGE sequence with 0.7 mm isotropic resolution (FOV = 224 mm, matrix = 320, 256 sagittal slices in a single slab), TR = 2400 ms, TE = 2.14 ms, TI = 1000 ms, FA = 8°, Bandwidth (BW) = 210 Hz per

pixel, Echo Spacing (ES) = 7.6 ms, with a non-selective binomial (1:1) water excitation pulse (a pair of 100 µs hard pulses with 1.2 ms spacing) to reduce signal from bone marrow and scalp fat, phase encoding undersampling factor GRAPPA = 2. T2w images are acquired using the variable flip angle turbo spin-echo sequence (Siemens SPACE;) with 0.7 mm isotropic resolution (same matrix, FOV, and slices as in the T1w), TR = 3200 ms, TE = 565 ms, BW = 744 Hz per pixel, no fat suppression pulse, phase encoding undersampling factor GRAPPA = 2, total turbo factor = 314 (to be achieved with a combination of turbo factor and slice turbo factor, when available), echo train length of 1105 echoes. Resting-state fMRI gradient echo EPI images are 2 mm isotropic resolution (FOV: 208 mm × 180 mm, Matrix: 104 × 90 with 72 slices covering the entire brain), acquired as pairs of R->L and L->R phase encoding directions.

ABCD: T1w images - 3D MPRAGE, 256x256x176, 1.0mm isotropic, TR=2500ms, TE=2.88ms, TI=1060ms, Flip angle=8°, R=2. T2w images - variable flip angle turbo spin-echo, 256x256x176, 1.0mm isotropic, TR=3200ms, TE=565ms, R=2. Resting-state fMRI - gradient echo EPI images, 90x90 with 60 slices, 2.4mm isotropic, TR=800ms, TE=30ms, flip angle = 52°, Multiband factor=6.

WashU120: T1-weighted images (TE=3.08 ms, TR(partition)=2.4 s, TI=1000 ms, flip angle=8°, 176 slices with 1×1×1 mm voxels). A T2-weighted turbo spin echo structural image (TE=84 ms, TR=6.8 s, 32 slices with 1×1×4 mm voxels) was acquired. Resting-state fMRI was performed using a blood oxygenation level-dependent (BOLD) contrast sensitive gradient echo echo-planar sequence (TE=27 ms, flip angle=90°, in-plane resolution=4×4 mm). Whole brain EPI volumes (MR frames) of 32 contiguous, 4 mm-thick axial slices were obtained every 2.5 seconds.

| Area of acquisition | Whole-brain |
|---|---|

**Diffusion MRI**  ☒ Used   ☐ Not used

| Parameters | Diffusion imaging was collected using a single-shot echo planar diffusion-weighted sequence consisting of 75 contiguous axial slices, isotropic (2x2x2 mm^3) resolution, TR/TE 3500/83 ms, four shells (b-values 0.25, 0.5, 1.0, and 1.5 ms/mm2) and 96 encoding directions. |
|---|---|

## Preprocessing

| Preprocessing software | FSL 6.0 software tools used: FAST, Eddy, Topup, DTIFit, FEAT<br>Freesurfer versions 5.0, 5.3, and 6.0, recon-all pipelines for brain segmentation<br>Connectome workbench v1.0 and 1.5<br>4dfp tools (https://4dfp.readthedocs.io/)<br>Processing pipelines used:<br>https://github.com/DCAN-Labs/abcd-hcp-pipelines<br>https://github.com/DCAN-Labs/nhp-abcd-bids-pipeline<br>Smoothing kernels employed: from 2mm to 6mm FWHM in humans and 1.5mm in macaques (geodesic on cortical surface) |
|---|---|
| Normalization | P01-03, P08, Infant, Perinatal Stroke: BOLD->T2 rigid body linear, T2->T1 rigid body linear, T1-> atlas linear<br>WU120: BOLD->T1 rigid body linear, T1-> atlas linear<br>Neonate: BOLD->T2 rigid body linear, T2-> atlas linear<br>All other datasets: BOLD->T2 rigid body linear, T2->T1 rigid body linear, T1-> atlas nonlinear |
| Normalization template | P01-03, Perinatal stroke, and WU120: Talaraich<br>All other human datasets: MNI<br>Macaque dataset: Yerkes 19 |
| Noise and artifact removal | P01-03, P08, Neonate, group Neonate, Infant, Perinatal Stroke:<br>Denoising of resting-state fMRI data was accomplished by regression of nuisance waveforms following a CompCor-like strategy. Regressors included the 6 rigid parameters derived by retrospective motion correction, the global signal averaged over the brain, and orthogonalized waveforms extracted from the ventricles, white matter and extra-cranial tissues (excluding the eyes). Frame censoring (scrubbing) was computed on the basis of both frame-wise displacement (FD) and variance of derivatives (DVARS) measures with thresholds set individually for each participant. Gray plot displays were visually checked to confirm artifact reduction. The data then were temporally band-pass filtered prior to nuisance regression, retaining frequencies between 0.005 Hz and 0.1 Hz. Censored frames were replaced by linearly interpolated values prior to filtering.<br><br>P04-07: Multi-echo ICA (ME-ICA) denoising designed to isolate spatially structured T2\*- (neurobiological; "BOLD-like") and S0-dependent (non-neurobiological; "not BOLD-like") signals was performed using a modified version of the "tedana.py" workflow (https://tedana.readthedocs.io/en/latest/). In short, the preprocessed, ACPC-aligned echoes were first combined according to the average rate of T2\* decay at each voxel across all time points by fitting the monoexponential decay, $S(t) = S0e^{-t/T2*}$, using the "nlinfit.m" function in MATLAB with least-squares optimization and the initial coefficient values obtained from a linear model fit to the log of the data. From these T2\* values, an optimally combined multi-echo (OC-ME) time-series was obtained by combining echoes using a weighted average ($WTE = TE * e^{-TE/T2*}$). The covariance structure of all voxel time-courses was used to identify major signals in the resultant OC-ME time-series using principal component and independent component analysis. Components were classified as either T2\*-dependent (and retained) or S0-dependent (and discarded), primarily according to their decay properties across echoes following the decision tree described in 85. We found that a global influence of respiration (a T2\*-dependent signal that is not of interest per se) was retained after removing S0-dependent components. Mean gray matter time-series regression was subsequently performed to remove this spatially diffuse noise. Finally, temporal masks were generated for censoring high motion time-points using a frame-wise displacement (FD) threshold of 0.3 mm and a backward difference of two TRs (2.77 s), for an effective sampling rate comparable to historical FD measurements (approximately 2 to 4 s). Prior to the FD calculation, head realignment parameters were filtered using a stopband Butterworth filter (0.2 - 0.35 Hz) to attenuate the influence of respiration. |

Child, ABCD: First, a respiratory filter was used to improve FD estimates calculated in the volume ("vol") stage. Second, temporal masks were created to flag motion-contaminated frames using the improved FD estimates. Frames with a filtered FD>0.3mm were flagged as motion-contaminated for nuisance regression only. After computing the temporal masks for high motion frame censoring, the data were processed with the following steps: (i) demeaning and detrending, (ii) interpolation across censored frames using least squares spectral estimation of the values at censored frames so that continuous data can be (iii) denoised via a GLM with whole brain, ventricular, and white matter signal regressors, as well as their derivatives. Denoised data were then passed through (iv) a band-pass filter (0.008 Hz<f<0.10 Hz) without re-introducing nuisance signals or contaminating frames near high motion frames.

Macaque: Nuisance regression using white matter (WM), and cerebrospinal fluid (CSF) signal and Friston-24 parameter models, bandpass filtering (0.01–0.1 Hz), detrending.

HCP, UKB: Resting state fMRI data were denoised for spatially specific temporal artefacts (for example, subject movement, cardiac pulsation, and scanner artefacts) using the ICA+FIX approach, which includes detrending the data and aggressively regressing out 24 movement parameters

WU120: Temporal masks were created to flag motion-contaminated frames. Motion contaminated volumes were identified by frame-by-frame displacement (FD), calculated as the sum of absolute values of the differentials of the 3 translational motion parameters and 3 rotational motion parameters. Volumes with FD>0.2 mm, as well as uncensored segments of data lasting fewer than 5 contiguous volumes, were flagged for removal. After computing the temporal masks for high motion frame censoring, the data were processed with the following steps: (i) demeaning and detrending, (ii), multiple regression including: whole brain, ventricular and white matter signals, and motion regressors derived by Volterra expansion, with censored data ignored during beta estimation, (iii) interpolation across censored frames using least squares spectral estimation of the values at censored frames so that continuous data can be passed through (iv) a band-pass filter (0.009 Hz<f<0.08 Hz) without contaminating frames near high motion frames. Censored frames were then excised from the data for all subsequent analyses.

| Volume censoring | see above |
|---|---|

## Statistical modeling & inference

| Model type and settings | For task data, we employed a mass univariate approach. To compute the overall degree of activation in response to each motion, data from each run was entered as a fixed effect into a first-level analysis within FSL's FEAT in which each condition timecourse was convolved with a hemodynamic response function to form a separate regressor in a GLM analysis testing for the effect of the multiple condition regressors on the timecourse of activity within every vertex/voxel in the brain. Beta value maps for each condition were extracted for each run and entered into a second-level analysis, in which run-level condition betas were tested as random effects. |
|---|---|
| Effect(s) tested | In the movement task battery, run-level condition betas were tested against a null hypothesis of zero activation in a one-sample t-test across runs (within participant).<br><br>In the action control task, a t-test across runs contrasted the run-level planning betas against the run-level execution betas (within participant). |

Specify type of analysis: ☐ Whole brain  ☒ ROI-based  ☐ Both

| Anatomical location(s) | Individual-specific M1 and CON ROIs were created using each subject's resting-state fMRI data. |
|---|---|

| Statistic type for inference (See Eklund et al. 2016) | ROI-wise t-values converted to Z-scores were used for inference. |
|---|---|
| Correction | Multiple comparisons were controlled using FDR correction. |

## Models & analysis

n/a | Involved in the study
☐ ☒ Functional and/or effective connectivity
☐ ☒ Graph analysis
☒ ☐ Multivariate modeling or predictive analysis

| Functional and/or effective connectivity | For each single-participant dataset, a vertex/voxelwise functional connectivity matrix was calculated from the resting-state fMRI data as the Fisher-transformed pairwise correlation of the timeseries of all vertices/voxels in the brain. In the ABCD, WashU120, eLABE, and PRIME-DE datasets, vertex/voxelwise group-averaged functional connectivity matrices were constructed by first calculating the vertex/voxelwise functional connectivity within each participant as the Fisher-transformed pairwise correlation of the timeseries of all vertices/voxels in the brain, and then averaging these values across participants at each vertex/voxel. |
|---|---|
| Graph analysis | To define the somatomotor regions that were visually identified from the seed-based connectivity analysis in an unbiased fashion for further exploration, we entered each individual adult human participant's data into a data-driven network detection algorithm designed to identify network subdivisions that are hierarchically below the level of classic large-scale networks (e.g. that produce hand/foot divisions in somatomotor cortex).<br><br>In each adult participant, this analysis clearly identified network structures corresponding to motor |

representation of the foot, hand, and mouth; and it additionally identified network structures corresponding exactly to the previously unknown connectivity pattern identified from the seed-based connectivity exploration as the inter-effector regions. For simplicity, we manually grouped all inter-effector subnetworks together as a single putative network structure (labeled as inter-effector) for further analysis.

Finally, to identify classic large-scale networks in each participant, we repeated the Infomap algorithm on matrices thresholded at a series of denser thresholds (ranging from 0.2% to 5%), and additionally identified individual-specific networks corresponding to the Default, Medial and Lateral Visual, Cingulo-Opercular, Fronto-Parietal, Dorsal Attention, Language, Salience, Parietal Memory, and Contextual Association networks following procedures described in 32.

