## [Peer Review File · Nature]

Manuscript Title: A Somato-Cognitive Action Network alternates with effector regions in motor cortex

Reviewer Comments & Author Rebuttals

Redactions – unpublished data

Reviewer Reports on the Initial Version: Referee #1 (Remarks to the Author):

This study investigates the organization of human primary motor cortex, often described as a consisting of a motor homunculus, using fMRI. Experts have long been aware that the foot-to-face organization of the motor cortex only superficially, or coarsely, matches that of the somatosensory cortex. Some clues about its more detailed organization come from the the demonstration in the monkey that (1) layer 5 neurons projecting to the ventral spinal cord are segregated into posterior distinct zones, and (2) prolonged stimulation in some portions of the motor cortex elicit multi-effector, coordinated actions, rather than action within muscles from a single body segment. Using primarily resting state functional connectivity in several cohorts of human subjects and one cohort of macaques, the authors demonstrate a reliable discontinuity within M1 motor strip. Namely, moving from dorsomedial to ventrolateral, the show that there are three primary effector areas, corresponding to the foot, hand, and mouth. These structures were homotopically connected with the other hemisphere. However, between these focal points, functional connectivity mapping revealed a very different pattern: the inter-effector regions map to to a much wider and more integrative network in the brain. The authors argue that the human primary motor cortex should be conceived as two parallel systems with fundamentally different roles in guiding action.

There is no question that the findings reported here are of wide interest to both neuroscience research and clinical medicine. As mentioned above, it is not a new idea that the classical homunculus is cartoonish unlikely incorrect at its core. Nonetheless, it has not been clear how to put together the previous evidence challenging this view, much of which has come from studies in the monkey, and how to apply it in a useful way to the human brain. This study in the human is therefore a significant advance. As is often the case, the coverage afforded by fMRI, together the ability to test the brains of many subjects, gives researchers sufficient traction to formulate a new perspective on brain organization. There are, however, a few concerns. First, the new formulation is based principally on resting state functional connectivity, which is itself robust but whose interpretation is always fraught. Second, there is the danger of replacing one imprecise cartoon with another, summarized in the authors' Fig 4b. These limitations and caveats should be apparent to all readers of this study. The authors support their basic finding with both morphometric and task-based imaging data (Figs 2 and 3, respectively) as well as a historical context and ample supplemental data from each individual participant, as well as from the different human and macaque cohorts. I have a few comments.

1. Major clues about the separate organization of “effector” regions, highly elaborated in primates, versus other corticospinal portions of M1, shared widely across mammals, come from anatomical

studies and electrical microstim studies in monkeys. Hence the inclusion of the monkey fMRI data in this paper serves as a potentially important bridge to the human. However, the results in the monkey (Fig S9) are confusing, or at least seem to require more explanation.

In Rathelot and Strick, the separation between “new” and “old” M1, which I believe the authors are taking to be equivalent to the “effector” vs. “inter-effector” regions in the human, are largely restricted to area 4 (and 3a). In particular, while the “old” M1 was more anterior than the “new” M1, there were relatively few cells identified in premotor area 6. The impression one has from Figure S9a, and the summary in Figure S9b, however, is that the area anterior to the effector regions does not reside within M1 at all, but is premotor. It is possible that this is a result of the unfolding / coloration of the maps. However, even though this is only a supplemental figure, I think this difference may lead to significant confusion.

As I understand, if this is truly premotor cortex showing the broad correlation in the macaque, then it is not the same areas identified by Strick and colleagues. The authors need to provide more description of the pattern in the macaque and explain whether it matches the expected results if this result is to serve as a bridge between the present findings in humans and the previous work.

2. The text (line 97) suggest that there are *direct* projections from M1 to the adrenal medulla, citing Dum and Strick (2019). I assume this is a typo. While this reference is certainly relevant, it most certainly does not demonstrate direct projections to the adrenal medulla: in both the monkey and the rat, the minimum retrograde path following adrenal medulla connections was three synaptic connections. The analogous pathway to the kidney had a minimum of four synaptic connections.

Referee #2 (Remarks to the Author):

The paper “A mind-body network...” by Gordon et al. reports the results of high precision fMRI scans of the human motor cortex. The results are compelling, clear, and consistent. The precision of the method, and the large amount of data, combine to provide a most convincing picture of the large-scale organization of the motor cortex running along the central sulcus. I was impressed with the methods, the data, and the clarity of presentation. I am therefore overall very enthusiastic about this paper. I did have major concerns about the way the findings were conceptually couched, and the way prior findings were discussed. But all of these concerns can easily be addressed by revision.

First, what I found compelling:

The authors show that human M1 has the expected pattern of foot, hand, and mouth representations. Each of these zones of motor cortex is more consistent with a concentric organization (e.g. hand at center, wrist wrapped around the hand, elbow wrapped around the wrist), rather than with a more traditional linear organization (elbow more medially, followed by wrist, and finally by hand more laterally). These findings confirm what others have found or suggested.

The authors also show some results that are clear, consistent, and yet divergent from traditional descriptions. Between the major body part representations, were “inter-effector” zones. The presence of these zones was so consistent that they could be found in nearly every individual, and even in children, the pattern stabilizing apparently around the age of 10. These inter-effector zones were preferentially connected to each other, to cortical areas such as the aACC, the pre-SMA, and the cingulo-opercular area, and also had a distinct pattern of connectivity to the basal ganglia, thalamus, cerebellum, and other areas. The authors interpreted these zones as representing all body parts in an integrative manner, and as preferentially involved in cognitive control. A comparison to monkey data suggested that the inter-effector zones may be comparable to the anterior M1 cortex of monkeys.

These findings are very important. I cannot overstate my enthusiasm for them.

Now for my concerns:

1. The inter-effector zones are conceptually lumped together and treated as functionally equivalent.

Figure 4 is not compelling. Why postulate three redundant zones, apparently doing the same thing – a poorly defined, generalized, all-body function? The authors have styled their findings as though two different functional compartments of motor cortex were distinguished: the usual, somatotopic zones which do one thing, and the general-body zones which do a second thing. But presumably the three inter-effector zones have divergent functions. They may share connectional similarities, and they may have preferential connections to each other, but it goes beyond the data to suggest a single general function for all three of them. The authors never discuss or acknowledged that the three may (presumably do) serve different functions. While the authors discuss at length the general connectional properties of the inter-effector zones, there is little or no discussion of how each of the inter-effector zones may be unique or may differ from the others in connectivity. I would like to see the paper address the possible connections or functions unique to each of the three zones.

Indeed, their presence (especially if they are comparable to anterior M1 cortex of monkeys) is strikingly consistent with the findings from the line of Graziano / Kaas / Stepniewska experiments, in which zones of motor cortex emphasized highly integrative, whole-body, complex actions. But in those prior experiments, different zones had very different functions. One zone, for example, seemed more involved in hand-to-mouth interactions, another zone seemed more involved in maintaining a protective buffer of space around the body through defensive actions, and a third zone seemed more involved in the complex coordination of torso, arm, and hand, for reaching. Could it be that the inter-effector zones described here match the zones of complex, ethological actions described by others previously? Maybe. Worth explicitly discussing in the paper. If so, that would explain why there are multiple inter-effector zones. But the authors never really discuss the possibility, and instead conceptualize the inter-effector zones as a functionally single unit.

2. More explicit acknowledgement and discussion of the Graziano / Kaas / Stepniewska findings

My second concern follows directly from the first. I am not suggesting the authors should accept or subscribe to the Graziano / Kaas / Stepniewska description of zones of complex, integrative

movements in motor cortex. But at least that description of motor cortex should be explicitly raised and discussed in comparison to the three integrative zones found here, in a sub-section of the discussion section. After all, the results described here are not as unique as the paper makes it sound. Distinct zones in motor cortex, that do not fit the traditional somatotopy, but instead involve a profound integration of many body parts, have been extensively described before, far beyond the few passing citations the authors give.

3. The “mind-body network” is a misnomer

I urge the authors to rethink the “mind-body” language. It is misleading. It’s too easy to come up with a cute phrase that’s so sticky, it captures people’s imaginations and clouds scientific understanding. These authors should know that, given how they write about Penfield’s homunculus. The homunculus idea was so sticky, that it is still widely accepted, even though the data never supported a clean somatotopy. Just so, I urge the authors not to label a “mind-body” network. “Mind” is not a well-defined concept anyway. Are the authors suggesting they have found the place where the spirit controls the body? I’m sure the authors don’t mean anything spiritual, but that’s what it sounds like. There is no evidence that the inter-effector zones represent the place where the mind communes with the body.

Moreover, the authors can’t even plausibly make the claim that the inter-effector zones are specifically where cognitive control has access to the body. Do they think that cognitive control is not connected to the somatotopic zones? When the subjects are asked to move their fingers, and then do so as a matter of cognitive control, the somatotopic finger area becomes active. So the hypothesis that the inter-effector zones are specifically where cognitive control has access to the body is nonsensical. Clearly the inter-effector zones have a special pattern of connectivity because they must serve special functions. But the paper would be better (and quite remarkable and important enough) without trying to put a grandiose spin on it. Find a more precisely descriptive name for the network.

Clearly the authors have made an important discovery about the human motor cortex, that discovery has specific similarities to at least some previous work that needs to be more explicitly cited and compared, and the exact functional role of the inter-effector zones remains a matter for useful discussion.

Referee #3 (Remarks to the Author):

General Comments:

This is a bold attempt to integrate a breadth of evidence within the motor and premotor cortex into a coherent hypothesis for mind-body integration. The authors are technically sophisticated and are leaders in functional imaging innovation. The basic observation was that resting state functional connectivity (RSFC) revealed that the primary motor cortex (M1) in humans forms an alternating pattern of differential connectivity. The medial to lateral foci of leg, arm and face representations

(termed effector regions) are not interconnected, in concordance with longstanding human and monkey evidence. Surprisingly, the junctions (termed inter-effector regions) between the leg-arm, arm-face and the most lateral extent of the face have interconnections among themselves. This organization has not been reported before. This observation is robust, present in individuals, group data, large collections of group data, and in young children to adults. The question then becomes, how to interpret this data? The authors suggest that the inter-effector regions in M1 form “an integrative system for implementing whole-body action plans, the Mind-Body Network (MBN).”

Two factors raise questions about this interpretation. First, the view that all three inter-effector regions mediate whole body movements depends on their greater connectivity with the cingulate-opercular network (CON) than the connectivity of the leg, arm and face effector regions with CON. This difference is only significant at the $p < 0.05$ level. Ideally, this difference would be evident for each inter-effector region versus each effector region and at a higher significance level. See below for an alternative interpretation of this data. Second, this hypothesis could not be confirmed in the monkey (see below for the justification of this statement). The monkey forms the foundational basis for the Mind-Body hypothesis. Confirmation in the monkey seems to be essential for this line of reasoning. These issues will be detailed in the specific comments below.

One of the cardinal principles of science is the ability to test a hypothesis. How could this mind-body hypothesis be tested? What criteria or experimental tasks would negate this hypothesis?

Specific Comments:

1. Can the inter-effector regions be distinguished from the circuit involved in speech articulation, phonation and respiration?

Correia et al., (Scientific Reports 2020:10:4529) shows speech related activations in sites that appear similar to the top and middle inter-effector sites in this manuscript. In Fig. 3 (Correia), the speech vs rest contrast also activated sites on the medial wall of the hemisphere that are co-extensive with CON. The cerebellar activation includes all the regions activated in the present manuscript. The voiced>whisper contrast (Correia) has a medial and lateral activations in the central sulcus similar in location to the top & middle inter-effector sites in this manuscript. All of Correia's sites are bilateral. Similar results can be seen in Figs. 4, 6, 7 (Correia). Note that this interpretation is based on visual comparisons as the inflated maps of the brains in Correia and Gordon are vastly different. Correia does not provide activation coordinates. The correspondence to the bottom inter-effector region is less clear but in some contrasts, very lateral activations are seen, possibly involving the lips or tongue (Fig. 7).

Other studies that supply coordinates support the speech/respiration interpretation. (Note- there are a multitude of issues in making these comparisons including various MNI vs Talairach coordinate systems, PET vs fMRI, brain inflation techniques, individual vs group averages, task design, activation thresholds, etc.). The Top-effector region appears similar to the location of trunk muscles and inspiration related (diaphragm, external intercostals) activations during breathing (Colebatch et al., 1991, Ramsay et al., 1993; Fink et al. 1996; Evans et al., 2004). This location may overlap with axial/trunk representations as well as adrenal/sympathetic stomach representations (Dum et al.,

2016; Levinthal and Strick, 2020).

Breathing related activations that are similar in location to Gordon et al., top- inter-effector region.

Colebatch et 1991 passive vs active resp

-18 -23 65

18 -18 60

Ramsay et 1993 active volitional respiration vs passive respiration

-14 -24 60

18 -20 60

Evans et 2004 fMRI breathing

-6 -30 70

Fink et 1996- increase inspiratory force

-14 -32 64

10 -30 60

18 -24 56

Gordon 2022 review L- top inter-effector region

-19 -34 59

20 -31 58

The middle inter-effector region (-38, -18, 44) seems to be in a similar location as the dorsal laryngeal motor area in Brown et al., 2020 (-41, -19, 38) (see also Loucks et al., NeuroImage 2007; Brown et al., Cerebral Cortex 2008; Murphy et al., J Appl Physiol 83:1438-1447, 1997). This location is bilateral as is inter-effector region. The lack of evoked movement during cortical stimulation (Fig. S10) from Roux et al., 2020 is entirely consistent with location of the dorsal laryngeal motor cortex of Brown et al., 2020 which is located deep within the central sulcus. This location would be inaccessible to the focal, bipolar stimulation of the cortical surface used in Roux. How can the middle effector region be distinguished from laryngeal motor cortex?

Breathing and speech related activations that are similar in location or potentially analogous to Gordon et al., middle inter-effector region (not all coordinates shown).

Gordon 2022 review L- middle

-38 -18 44

40 -15 43

Brown et al. 2020 L- dorsal Laryngeal

-41 -19 38

Loucks et 2007 L- repetitive vs continuous vocalization

-44 -10 39

exhalation>rest

-44 -12 38

Murphy et 1997

54 -8 36

48 -16 28

Nakayama et 2004

48 -4 47

A major characteristic tying the inter-effector regions together and differentiating them from the effector regions is their stronger association with CON (cingulate opercular network) (Fig. 2). In monkeys and marmosets, the cortical projections to a laryngeal muscle (cricothyroid) have a much greater proportion of disynaptic input from medial wall areas (~CON) than does the hand region (Cerkevich et al., 2022). Thus, one would predict that the laryngeal motor cortex of humans would have stronger connections to medial wall areas (same region as origin of CON) than would the hand region.

What criteria would distinguish these two alternatives? How can this alternative hypothesis be distinguished from the inter-effector global mind-body hypothesis?

2. Why doesn't the monkey functional connectivity (RSFC) data correspond to the known neuroanatomical connectivity of effector related regions? Numerous anatomical tracing experiments have demonstrated multiple premotor areas on the medial wall (i.e., digit/arm- Dum and Strick, 1991, 2002, 2005; Hatanaka et al., 2001; Yang et al., 2001; Gharbawie et al., 2011; face/orofacial- Morecraft et al., 1996; Tokuno et al., 1997; Gharbawie et al. 2011; toes/leg- Hatanaka et al., 2001). This is the basic gold standard of reproducibility. This gold standard has been confirmed by Card & Gharbawie, 2022 where they demonstrated correspondence between resting state functional connectivity and anatomical tracing. If functional connectivity does not reproduce these findings in detail, then do the RSFC methods as applied to monkeys have the precision and sensitivity necessary to validate data?

Which inter-effector regions are the PMd and PMv analogous to? In Fig. S9, the two premotor points (purple) are misplaced relative to Woolsey's map. Further, the PMd and PMv do not have uniform somatotopy. Multiple seeds would need to be placed in each premotor region to get a full picture. According to the hypothesis that these 2 premotor areas correspond to inter-effector regions, each part of them should be connected to each other. In Fig. S9, there is minimal dorsal to ventral connectivity. In contrast, one would expect extensive connectivity if these premotor areas are truly equivalent to the inter-effector regions of humans. Why are the connections between these "inter-effector" regions so weak?

The monkey data seems necessary to validate the whole concept of Mind-Body Network. This is because the adrenal and stomach data are derived from animal models. The adrenal data, in particular, has a major component from M1 (Dum et al., 2016). Thus, the middle inter-effector regions should be present in the precentral gyrus and central sulcus of monkeys.

3. Do the inter-effector regions really have the same functional roles (i.e., "an integrative system for implementing whole-body action plans, the Mind-Body Network (MBN)"?)

The concept of Mind-Body network seems to be rooted in the role of autonomic control- specifically the sympathetic nervous system output located in M1 and the premotor areas. However, the sympathetic system only seems to apply primarily to the top inter-effector region and not the middle and bottom sites. Doesn't this difference distinguish these 3 sites and suggest that they don't have the same functional roles?

Fig. 4a is entirely consistent with Fig. 3 and Penfield's interpretation of motor output in the human (as is also Roux et al., 2020 stimulation based map). Why is a new map needed? Shouldn't the new map be more accurate?

Fig. 4b implies that the 3 inter-effector regions are the same, each representing the whole body (puppet figurine). However, Fig. S7 tells a completely different story. Although it is hard to place the inter-effector regions exactly on the map, the top inter-effector region has leg, abdominal and proximal arm muscles, the middle has proximal arm, maybe abdominal and upper face but no leg and the ventral inter-effector region has only lower face. (Placing a gray box for each inter-effector region would be helpful). How can these all be considered whole body representations as represented by the puppet figurine? Are the results in 2 individuals sufficient evidence to draw a whole new map?

Huber et al., 2018, 2020 has provided high fidelity evidence that there are at least 2 hand representations in M1. According to Huber, a more accurate representation of the hand would have 2 hands, joined at the 5th digit with a larger hand, thumb laterally and a smaller hand, thumb medially. According to Figs. 3 and S7, the medial arm should be more robust and the lateral arm thinner. What evidence suggests that the face has a symmetrical, nested organization? Neither the maps in Bouchard et al., 2013 or in Figs. 3 and S7 seem to support this interpretation.

4. There is a general lack of coordinates to identify regions outside of inter-effector regions.

Fig. 2- What are the coordinates of the preSMA and dACC for each of the 3 inter-effector areas? The preSMA could actually be the SMA.

CON specificity for inter-effector region is only at $P < 0.05$. This is very weak. The reason for this analysis is to figure out what the inter-effector region is doing. What is CON's functional specialty? Whole concept of CON connectivity is somewhat spurious since it is mainly the SMA and cingulate motor areas that provide the bulk of the connectivity with inter-effector regions.

What are the coordinates of the medial wall areas with each inter-effector region? The hypothesis implies that they are quite similar. Is that true? Is there any criteria that would make this distinction unambiguously?

Line 177- extra "c" in middle of description of "b"

line 288- The term "posterior bank of the precentral gyrus" is confusing since many know this region as the anterior bank of the central sulcus.

Author Rebuttals to Initial Comments:

We wish to thank the Reviewers for their high level of enthusiasm for our work. We are very grateful that Reviewer 1 described our findings as “a significant advance” and “a new perspective on brain organization”, and notes that “There is no question that the findings reported here are of wide interest to both neuroscience research and clinical medicine.” Reviewer 2 commented that “The results are compelling, clear, and consistent. The precision of the method, and the large amount of data, combine to provide a most convincing picture of the large-scale organization of the motor cortex running along the central sulcus. I was impressed with the methods, the data, and the clarity of presentation. I am therefore overall very enthusiastic about this paper. [...] These findings are very important. I cannot overstate my enthusiasm for them.” And Reviewer 3 opined that “This is a bold attempt to integrate a breadth of evidence within the motor and premotor cortex into a coherent hypothesis for mind-body integration. The authors are technically sophisticated and are leaders in functional imaging innovation.”

We greatly appreciate the Reviewers’ insightful comments and suggestions, which have substantially improved the manuscript.

Below, we respond to all Reviewers’ comments and suggestions (Reviewers’ comments in black; our responses in blue), and we reproduce all resulting new text and Figures included in the manuscript.

Referee #1

This study investigates the organization of human primary motor cortex, often described as a consisting of a motor homunculus, using fMRI. Experts have long been aware that the foot-to-face organization of the motor cortex only superficially, or coarsely, matches that of the somatosensory cortex. Some clues about its more detailed organization come from the the demonstration in the monkey that (1) layer 5 neurons projecting to the ventral spinal cord are segregated into posterior distinct zones, and (2) prolonged stimulation in some portions of the motor cortex elicit multi-effector, coordinated actions, rather than action within muscles from a single body segment. Using primarily resting state functional connectivity in several cohorts of human subjects and one cohort of macaques, the authors demonstrate a reliable discontinuity within M1 motor strip. Namely, moving from dorsomedial to ventrolateral, the show that there are three primary effector areas, corresponding to the foot, hand, and mouth. These structures were homotopically connected with the other hemisphere. However, between these focal points, functional connectivity mapping revealed a very different pattern: the inter-effector regions map to to a much wider and more integrative network in the brain. The authors argue that the human primary motor cortex should be conceived as two parallel systems with fundamentally different roles in guiding action.

There is no question that the findings reported here are of wide interest to both neuroscience research and clinical medicine. As mentioned above, it is not a new idea that the classical homunculus is cartoonish unlikely incorrect at its core. Nonetheless, it has not been clear how to put together the previous evidence challenging this view, much of which has come from studies in the monkey, and how to apply it in a useful way to the human brain. This study in the human is therefore a significant advance. As is often the case, the coverage afforded by fMRI, together the ability to test the brains of many subjects, gives researchers sufficient traction to formulate a new perspective on brain organization. There are, however, a few concerns.

First, the new formulation is based principally on resting state functional connectivity, which is itself robust but whose interpretation is always fraught. Second, there is the danger of replacing one imprecise cartoon with another, summarized in the authors' Fig 4b. These limitations and caveats should be apparent to all readers of this study.

We have included a caveat about taking the updated motor cortex schematic literally. However, the current illustration, represents a major advance in accuracy, and is needed to help combat the “stickiness” of the homuncular concept. We have also tweaked the illustration to further improve its accuracy. Most notably, there is a dual, symmetric representation of the fingers (as pointed out by R3) that has now been added to the illustration.

Previous illustration:

Updated illustration:

Fig. 4| The interrupted homunculus – integrate/isolate model of action and motor control. a, Penfield’s classical homunculus (adapted from 2) depicting a continuous map of the body in primary motor cortex. b, In the integrate-isolate model of primary motor cortex organization, effector-specific (foot [green], hand [cyan], mouth [orange]) functional zones are represented by concentric rings with proximal body parts surrounding the relatively more isolatable distal ones (toes, fingers, tongue). Inter-effector regions (maroon) sit at the intersecting points of these fields, forming part of a Somato-Cognitive Action Network (SCAN) for integrative, allostatic whole-body control. As with Penfield’s original drawing, this diagram is intended to illustrate organizational principles, and must not be over-interpreted as a precise map.

The authors support their basic finding with both morphometric and task-based imaging data (Figs 2 and 3, respectively) as well as a historical context and ample supplemental data from each individual participant, as well as from the different human and macaque cohorts. I have a few comments.

1. Major clues about the separate organization of “effector” regions, highly elaborated in primates, versus other corticospinal portions of M1, shared widely across mammals, come from anatomical studies and electrical microstim studies in monkeys. Hence the inclusion of the monkey fMRI data in this paper serves as a potentially important bridge to the human. However, the results in the monkey (Fig S9) are confusing, or at least seem to require more explanation.

In Rathelot and Strick, the separation between “new” and “old” M1, which I believe the authors are taking to be equivalent to the “effector” vs. “inter-effector” regions in the human, are largely restricted to area 4 (and 3a). In particular, while the “old” M1 was more anterior than the “new” M1, there were relatively few cells identified in premotor area 6. The impression one has from Figure S9a, and the summary in Figure S9b, however, is that the area anterior to the effector regions does not reside within M1 at all, but is premotor. It is possible that this is a result of the unfolding / coloration of the maps. However, even though this is only a supplemental figure, I think this difference may lead to significant confusion.

As I understand, if this is truly premotor cortex showing the broad correlation in the macaque, then it is not the same areas identified by Strick and colleagues. The authors need to provide more description of the pattern in the macaque and explain whether it matches the expected results if this result is to serve as a bridge between the present findings in humans and the previous work.

Thank you for raising these important points. Unfortunately, macaque BOLD fMRI scanning techniques have not yet achieved the same data quality as in humans, and scanning cannot easily be performed on an extended basis to collect the same data quantities as in humans. Therefore, there is necessarily somewhat more noise in the macaque data, and necessarily more uncertainty in their interpretation. In the data shown in the original manuscript draft, seeds placed in the central sulcus did not identify a clearly homologous network. Instead, the action network we identified in macaques appeared as an intermediate between the CON (Cingulo-Opercular Network) and the SCAN (Somato-Cognitive Action Network). In macaques, the lower overall SNR and larger voxel sizes (relative to the size of the brain) may be impeding clearer separation of inter-effector regions from adjacent effector-specific regions.

To deepen our search for inter-effector regions, we seeded regions of the medial frontal gyrus containing putative CON homologues. Using these seeds, we observed strong connectivity with two regions in the central sulcus that approximated the locations of the top and middle inter-effector region observed in humans (see Extended Data Fig 9a, top).

To validate this finding, we obtained new macaque BOLD data from 8 additional animals. In individual animals (Extended Data Fig 9b, top) and in their group average (Extended Data Fig 9c, top), we replicated this pattern of connectivity from medial prefrontal cortex to two distinct regions in central sulcus. Thus, with additional macaque data and analyses, we were able to identify a connectivity pattern more closely homologous to that found in humans.

Even with the additional data and analyses, the SCAN nodes in macaque M1 do not appear to be as large as in humans; and further, we only identified two candidate homologue SCAN regions. Technical limitations of macaque fMRI are one potential explanation for these differences between the macaque and human SCAN. However, another potential explanation is that the SCAN may be evolutionarily expanded because it is important for coordinating complex actions relatively specific to humans, such as speech and/or tool use. The revised manuscript reflects these additional insights.

Extended Data Figure 9| Functional connectivity in non-human primates. Functional connectivity maps were seeded from dorsal anterior cingulate cortex (top row), as well as from a continuous line of points down anterior central sulcus (rows 2-4), in fMRI data from **a**, an individual macaque scanned for 77 minutes on a 10.5T MRI scanner; **b**, an individual macaque scanned for 53 minutes on a 3T scanner; and **c**, group average data from eight macaques each scanned for 53 minutes on a 3T scanner. The dorsal anterior cingulate seed demonstrated connectivity to frontal, insular, and parietal regions homologous with the human CON, as well as with two small regions in anterior central sulcus (maroon arrows). These central sulcus regions correspond to areas that project to internal organs^{10,53} and represent possible macaque homologues of the inter-effector regions. The central sulcus seeds demonstrated connectivity patterns corresponding to the known functional divisions between M1 regions representing the foot (second row), hand (third row), and face (bottom row).

Added text Results:

To link these neuroimaging findings to decades of detailed motor cortex mapping in non-human primates, we searched for inter-effector homologues in macaques using fMRI. Seeds placed in macaque M1 revealed foot, hand, and mouth effector-specific functional connectivity patterns consistent with those seen in humans³² (Extended Data Fig. 9, rows 2-4). In macaque fMRI, lower overall SNR (signal-to-noise ratio) and larger voxel sizes relative to the size of the brain can impede clear separation of the smaller inter-effectors from adjacent effector-specific regions. Therefore, we also seeded putative CON homologues in dACC, which revealed strong connectivity with lateral frontal cortex, insula, and supramarginal gyrus, similar to the human CON, and with two regions in anterior central sulcus that may be homologous to the superior and middle inter-effectors (Extended Data Fig. 9, row 1).

Added text: Discussion

The apparent relative expansion of SCAN regions in humans could suggest a role in complex actions specific to humans, such as coordinating breathing for speech, and integrating hand, body and eye movement for tool use.

2. The text (line 97) suggest that there are *direct* projections from M1 to the adrenal medulla,

citing Dum and Strick (2019). I assume this is a typo. While this reference is certainly relevant, it most certainly does not demonstrate direct projections to the adrenal medulla: in both the monkey and the rat, the minimum retrograde path following adrenal medulla connections was three synaptic connections. The analogous pathway to the kidney had a minimum of four synaptic connections.

Thank you for noticing. We have corrected the corresponding text.

Referee #2

The paper “A mind-body network...” by Gordon et al. reports the results of high precision fMRI scans of the human motor cortex. The results are compelling, clear, and consistent. The precision of the method, and the large amount of data, combine to provide a most convincing picture of the large-scale organization of the motor cortex running along the central sulcus. I was impressed with the methods, the data, and the clarity of presentation. I am therefore overall very enthusiastic about this paper. I did have major concerns about the way the findings were conceptually couched, and the way prior findings were discussed. But all of these concerns can easily be addressed by revision.

First, what I found compelling:

The authors show that human M1 has the expected pattern of foot, hand, and mouth representations. Each of these zones of motor cortex is more consistent with a concentric organization (e.g. hand at center, wrist wrapped around the hand, elbow wrapped around the wrist), rather than with a more traditional linear organization (elbow more medially, followed by wrist, and finally by hand more laterally). These findings confirm what others have found or suggested.

The authors also show some results that are clear, consistent, and yet divergent from traditional descriptions. Between the major body part representations, were “inter-effector” zones. The presence of these zones was so consistent that they could be found in nearly every individual, and even in children, the pattern stabilizing apparently around the age of 10. These inter-effector zones were preferentially connected to each other, to cortical areas such as the aACC, the pre-SMA, and the cingulo-opercular area, and also had a distinct pattern of connectivity to the basal ganglia, thalamus, cerebellum, and other areas. The authors interpreted these zones as representing all body parts in an integrative manner, and as preferentially involved in cognitive control. A comparison to monkey data suggested that the inter-effector zones may be comparable to the anterior M1 cortex of monkeys.

These findings are very important. I cannot overstate my enthusiasm for them.

Now for my concerns:

1. The inter-effector zones are conceptually lumped together and treated as functionally equivalent.

Figure 4 is not compelling. Why postulate three redundant zones, apparently doing the same thing – a poorly defined, generalized, all-body function? The authors have styled their findings as though two different functional compartments of motor cortex were distinguished: the usual,

somatotopic zones which do one thing, and the general-body zones which do a second thing. But presumably the three inter-effector zones have divergent functions. They may share connectional similarities, and they may have preferential connections to each other, but it goes beyond the data to suggest a single general function for all three of them. The authors never discuss or acknowledged that the three may (presumably do) serve different functions. While the authors discuss at length the general connectional properties of the inter-effector zones, there is little or no discussion of how each of the inter-effector zones may be unique or may differ from the others in connectivity. I would like to see the paper address the possible connections or functions unique to each of the three zones.

Indeed, their presence (especially if they are comparable to anterior M1 cortex of monkeys) is strikingly consistent with the findings from the line of Graziano / Kaas / Stepniewska experiments, in which zones of motor cortex emphasized highly integrative, whole-body, complex actions. But in those prior experiments, different zones had very different functions. One zone, for example, seemed more involved in hand-to-mouth interactions, another zone seemed more involved in maintaining a protective buffer of space around the body through defensive actions, and a third zone seemed more involved in the complex coordination of torso, arm, and hand, for reaching. Could it be that the inter-effector zones described here match the zones of complex, ethological actions described by others previously? Maybe. Worth explicitly discussing in the paper. If so, that would explain why there are multiple inter-effector zones. But the authors never really discuss the possibility, and instead conceptualize the inter-effector zones as a functionally single unit.

Thank you for these helpful suggestions. We also have been intensely interested in what might functionally differentiate the three Somato-Cognitive Action Network (SCAN) nodes. As you pointed out, we did not discuss this question in the original manuscript, primarily because the most striking observation about them was their great similarity in all the data we examined, which stood in stark contrast to their clear and obvious differences from the foot, hand and mouth M1 regions. The revised manuscript now discusses potential differentiating factors across the three SCAN nodes, including the powerful evidence from macaque stimulation studies suggesting differentiation across action types.

We have also added analyses describing connectivity differences between the SCAN regions in M1 to the revised manuscript. For each region in each participant, we identified locations in the brains of individual participants and in group average data where FC was stronger to that node than to either of the other two SCAN nodes.

We observed that this node selectivity was relatively sparse across the cortex and was not very consistent across individuals (Extended Data Fig 4; Fig S2). However, for each SCAN node, there were small regions on the medial wall that exhibited at least some node selectivity that was consistent across individuals and observed in group-averaged data.

Most notably, the middle SCAN node consistently exhibited stronger FC to extrastriate visual cortex than either of the other two nodes, suggesting closer integration with processing visual information.

We now discuss the possibility, based on foundational work in NHP, that the different SCAN nodes are important for different types of actions, and in particular that the middle node may be critically involved in coordinated actions requiring visual inputs.

Added text: Results

Searching for differences between the three inter-effector regions revealed that the middle inter-effector consistently exhibited stronger functional connectivity to extrastriate visual cortex than either of the other two regions (Extended Data Fig 5; Fig S2 for all participants).

Added text: Discussion

Minor connectivity (Extended Data Fig. 5) and activation (Extended Data Fig. 7) differences between the superior, middle and inferior inter-effector regions likely reflect some degree of functional specialization within this integrated system. The middle region's relatively stronger connectivity to visual cortex, for example, could suggest a potential role in hand-eye coordination during reach-and-grasp motions⁵⁷.

Added text: Discussion

The apparent relative expansion of SCAN regions in humans could suggest a role in complex actions specific to humans, such as coordinating breathing for speech, and integrating hand, body and eye movement for tool use.

Extended Data Fig. 5 | Differences in functional connectivity between inter-effector regions. Brain regions more strongly connected to the superior inter-effector region than to either of the other two (top row); relatively most strongly connected to the middle inter-effector region (middle row); and relatively most strongly connected to the inferior inter-effector region, in cortex (left), striatum, thalamus, and cerebellum (right). **a**, In at least 50% of individuals ($n = 7$) and **b**, group-averaged data from the Human Connectome Project (HCP; $n = 812$). Thresholds used are the same as in Fig. 2b. Note that central sulcus regions are masked as they exhibit large differences by definition. See Fig. S2 for all individual participants.

Fig. S2| Functional connectivity differences among inter-effector regions in individual participants. In seven individuals, brain regions more strongly connected to the superior inter-effector region (top row), middle inter-effector region (middle row), and inferior inter-effector region (bottom row) than to either of the other two, in cortex (left), striatum, thalamus, and cerebellum (right). Thresholds used are the same as in Fig. 2b. Note that central sulcus regions are masked as they exhibit large differences by definition. See Extended Data Fig. 5 for overlap across individuals.

2. More explicit acknowledgement and discussion of the Graziano / Kaas / Stepniewska findings

My second concern follows directly from the first. I am not suggesting the authors should accept or subscribe to the Graziano / Kaas / Stepniewska description of zones of complex, integrative movements in motor cortex. But at least that description of motor cortex should be explicitly raised and discussed in comparison to the three integrative zones found here, in a sub-section of the discussion section. After all, the results described here are not as unique as the paper makes it sound. Distinct zones in motor cortex, that do not fit the traditional somatotopy, but instead involve a profound integration of many body parts, have been extensively described before, far beyond the few passing citations the authors give.

We fully agree that the Graziano / Kaas / Stepniewska findings are most relevant here. Indeed, this groundbreaking work motivated and shaped our understanding of the SCAN regions. We have now added a paragraph to the Discussion that addresses these ideas in greater detail:

Added text:

Direct stimulation studies in macaques have evoked complex, multi-effector actions by applying longer stimulation trains (500 ms) to motor cortex^{4,18,56,57}. These actions range from feeding behaviors to climbing to defensive postures—movements which are purposeful and coordinated rather than isolated, involving integration of muscles across the classic foot/hand/mouth divisions. The inter-effector regions, which are connected to action planning areas (Fig. 2) and active during a wide range of foot, hand, and mouth movements (Extended Data Fig. 7, ⁵⁸), represent candidate human homologues to the macaque multi-effector action sites.

3. The “mind-body network” is a misnomer

I urge the authors to rethink the “mind-body” language. It is misleading. It’s too easy to come up with a cute phrase that’s so sticky, it captures people’s imaginations and clouds scientific understanding. These authors should know that, given how they write about Penfield’s homunculus. The homunculus idea was so sticky, that it is still widely accepted, even though the data never supported a clean somatotopy. Just so, I urge the authors not to label a “mind-body” network. “Mind” is not a well-defined concept anyway. Are the authors suggesting they have found the place where the spirit controls the body? I’m sure the authors don’t mean anything spiritual, but that’s what it sounds like. There is no evidence that the inter-effector zones represent the place where the mind communes with the body.

Moreover, the authors can’t even plausibly make the claim that the inter-effector zones are specifically where cognitive control has access to the body. Do they think that cognitive control is not connected to the somatotopic zones? When the subjects are asked to move their fingers, and then do so as a matter of cognitive control, the somatotopic finger area becomes active. So the hypothesis that the inter-effector zones are specifically where cognitive control has access

to the body is nonsensical. Clearly the inter-effector zones have a special pattern of connectivity because they must serve special functions. But the paper would be better (and quite remarkable and important enough) without trying to put a grandiose spin on it. Find a more precisely descriptive name for the network.

We agree that the “mind-body” terminology has its downsides. As suggested, we have renamed this network the Somato-Cognitive Action Network (SCAN).

Referee #3

General Comments:

This is a bold attempt to integrate a breadth of evidence within the motor and premotor cortex into a coherent hypothesis for mind-body integration. The authors are technically sophisticated and are leaders in functional imaging innovation. The basic observation was that resting state functional connectivity (RSFC) revealed that the primary motor cortex (M1) in humans forms an alternating pattern of differential connectivity. The medial to lateral foci of leg, arm and face representations (termed effector regions) are not interconnected, in concordance with longstanding human and monkey evidence. Surprisingly, the junctions (termed inter-effector regions) between the leg-arm, arm-face and the most lateral extent of the face have interconnections among themselves. This organization has not been reported before. This observation is robust, present in individuals, group data, large collections of group data, and in young children to adults. The question then becomes, how to interpret this data? The authors suggest that the inter-effector regions in M1 form “an integrative system for implementing whole-body action plans, the Mind-Body Network (MBN).”

Two factors raise questions about this interpretation. First, the view that all three inter-effector regions mediate whole body movements depends on their greater connectivity with the cingulate-opercular network (CON) than the connectivity of the leg, arm and face effector regions with CON. This difference is only significant at the $p < 0.05$ level. Ideally, this difference would be evident for each inter-effector region versus each effector region and at a higher significance level. See below for an alternative interpretation of this data. Second, this hypothesis could not be confirmed in the monkey (see below for the justification of this statement). The monkey forms the foundational basis for the Mind-Body hypothesis. Confirmation in the monkey seems to be essential for this line of reasoning. These issues will be detailed in the specific comments below.

One of the cardinal principles of science is the ability to test a hypothesis. How could this mind-body hypothesis be tested? What criteria or experimental tasks would negate this hypothesis?

Specific Comments:

1. Can the inter-effector regions be distinguished from the circuit involved in speech articulation, phonation and respiration?

Correia et al., (Scientific Reports 2020:10:4529) shows speech related activations in sites that appear similar to the top and middle inter-effector sites in this manuscript. In Fig. 3 (Correia), the speech vs rest contrast also activated sites on the medial wall of the hemisphere that are co-extensive with CON. The cerebellar activation includes all the regions activated in the present manuscript. The voiced>whisper contrast (Correia) has a medial and lateral activations in the central sulcus similar in location to the top & middle inter-effector sites in this manuscript. All of

Correia's sites are bilateral. Similar results can be seen in Figs. 4, 6, 7 (Correia). Note that this interpretation is based on visual comparisons as the inflated maps of the brains in Correia and Gordon are vastly different. Correia does not provide activation coordinates. The correspondence to the bottom inter-effector region is less clear but in some contrasts, very lateral activations are seen, possibly involving the lips or tongue (Fig. 7).

Other studies that supply coordinates support the speech/respiration interpretation. (Note- there are a multitude of issues in making these comparisons including various MNI vs Talairach coordinate systems, PET vs fMRI, brain inflation techniques, individual vs group averages, task design, activation thresholds, etc.). The Top-effector region appears similar to the location of trunk muscles and inspiration related (diaphragm, external intercostals) activations during breathing (Colebatch et al., 1991, Ramsay et al., 1993; Fink et al. 1996; Evans et al., 2004). This location may overlap with axial/trunk representations as well as adrenal/sympathetic stomach representations (Dum et al., 2016; Levinthal and Strick, 2020).

Breathing related activations that are similar in location to Gordon et al., top- inter-effector region.

Colebatch et 1991 passive vs active resp

-18 -23 65

18 -18 60

Ramsay et 1993 active volitional respiration vs passive respiration

-14 -24 60

18 -20 60

Evans et 2004 fMRI breathing

-6 -30 70

Fink et 1996- increase inspiratory force

-14 -32 64

10 -30 60

18 -24 56

Gordon 2022 review L- top inter-effector region

-19 -34 59

20 -31 58

The middle inter-effector region (-38, -18, 44) seems to be in a similar location as the dorsal laryngeal motor area in Brown et al., 2020 (-41, -19, 38) (see also Loucks et al., NeuroImage 2007; Brown et al., Cerebral Cortex 2008; Murphy et al., J Appl Physiol 83:1438-1447, 1997). This location is bilateral as is inter-effector region. The lack of evoked movement during cortical stimulation (Fig. S10) from Roux et al., 2020 is entirely consistent with location of the dorsal laryngeal motor cortex of Brown et al., 2020 which is located deep within the central sulcus. This location would be inaccessible to the focal, bipolar stimulation of the cortical surface used in Roux. How can the middle effector region be distinguished from laryngeal motor cortex?

Breathing and speech related activations that are similar in location or potentially analogous to Gordon et al., middle inter-effector region (not all coordinates shown).

Gordon 2022 review L- middle

-38 -18 44

40 -15 43

Brown et al. 2020 L- dorsal Laryngeal

-41 -19 38

Loucks et 2007 L- repetitive vs continuous vocalization

-44 -10 39

exhalation>rest
-44 -12 38
Murphy et 1997
54 -8 36
48 -16 28
Nakayama et 2004
48 -4 47

Thank you for these excellent comments. We do agree that the SCAN is likely involved in the coordination of breathing with other actions, especially as part of speech. As a network important for whole-body integrated action control, top-down control of breathing beyond that afforded by the medulla is likely to be one of its functions. We now discuss this point.

Discussion text (added text underlined):

The present work suggests these functions include action implementation, as well as postural and gross motor control of axial muscles, while prior work in humans and non-human primates suggests these circuits may also regulate arousal⁷, coordinate breathing with speech and other complex actions^{58,59}, and control internal processes and organs (i.e., blood pressure⁶, stomach⁵⁴, adrenal medulla⁵³), consistent with circuits for whole-body, metabolic, and physiological control.

Discussion text (added text underlined):

The SCAN provides a substrate for this integration, enabling pre-action anticipatory postural, breathing, cardiovascular, and arousal changes (e.g., shoulder tension, increased heart rate, butterflies in the stomach).

Since posting our preprint, Silva et al. (2022) published a review in the Journal of Neuroscience arguing that a region in the middle precentral gyrus, which may partially overlap with the middle SCAN node, is important for phonological-motoric aspects of speech production. This idea, based on human electrophysiology is now discussed in the revised manuscript.

Discussion text added:

Human speech BCI studies have suggested that the precentral gyrus between the hand and mouth effectors is essential for phonological-motoric aspects of speech planning⁷¹.

The question of whether the SCAN is involved more specifically in laryngeal control is a separate one. Given the prior work cited by the reviewer, we suspected that the dorsal laryngeal motor area is likely ventral to the middle inter-effector region (as the noted coordinates are consistently 4-6 mm more ventral than the middle inter-effector region).

To address this excellent question, we collected new fMRI data in the same two exemplar participants, as well as in one additional novel participant, during performance of a vocalization task (repeatedly articulating the phoneme “ee” in one breath). We found that vocalization-related activity was specific to two mirrored regions in M1, consistent with prior work (Eichert et al., 2020, Cerebral Cortex; Belyk et al., 2021, Philos Trans B; Castellucci et al., 2022, Nature), and that these two regions were directly adjacent to, but not overlapping with, the middle and inferior SCAN nodes. These vocalization-related regions were inside the mouth area, providing more support for, and detail of, the concentric organization of this mouth portion of M1. Because the task did not require inspiration during each phonation block (unlike the Correia task, and in

particular their “Voiced > Whisper” contrast), activation of the top SCAN region in our data was minimal. This result is now included as Figure S4.

Fig. S4| Vocalization task activation is restricted to the mouth effector region. In three participants, a Voice > [Foot + Hand + Tongue] task fMRI contrast identified vocalization-specific activation in two mirrored regions in precentral gyrus (green). Connectivity seeds placed in the middle of the Mouth area revealed that the vocalization-related regions were located within the mouth area, as defined by functional connectivity (red-orange).

Added text:

To verify that inter-effector function is not specific to vocalization^{50,51}, we also collected task fMRI data while participants repeatedly made an ‘ee’ sound, to isolate movement of the larynx while minimizing respirations and jaw and tongue motion. We observed a dual laryngeal representation that was confined to the mouth area rather than extending into the inter-effector regions (Fig S4), consistent with^{52,53} and with a concentric functional zone organization.

A major characteristic tying the inter-effector regions together and differentiating them from the effector regions is their stronger association with CON (cingulate opercular network) (Fig. 2). In monkeys and marmosets, the cortical projections to a laryngeal muscle (cricothyroid) have a much greater proportion of disynaptic input from medial wall areas (~CON) than does the hand

region (Cerkevich et al., 2022). Thus, one would predict that the laryngeal motor cortex of humans would have stronger connections to medial wall areas (same region as origin of CON) than would the hand region.

What criteria would distinguish these two alternatives? How can this alternative hypothesis be distinguished from the inter-effector global mind-body hypothesis?

The vocalization-related regions described above exhibit some connectivity to very small regions on the medial wall, but not to the CON (see Figure R1, below).

Figure R1| Functional connectivity of vocalization-related regions. In Participant 2, seeds placed in the regions identified as vocalization-related (above) connected moderately to small regions on the medial wall, but not to other regions in CON.

2. Why doesn't the monkey functional connectivity (RSFC) data correspond to the known neuroanatomical connectivity of effector related regions? Numerous anatomical tracing experiments have demonstrated multiple premotor areas on the medial wall (i.e., digit/arm- Dum and Strick, 1991, 2002, 2005; Hatanaka et al., 2001; Yang et al., 2001; Gharbawie et al, 2011; face/orofacial- Morecraft et al., 1996; Tokuno et al., 1997; Gharbawie et al. 2011; toes/leg- Hatanaka et al., 2001). This is the basic gold standard of reproducibility. This gold standard has been confirmed by Card & Gharbawie, 2022 where they demonstrated correspondence between resting state functional connectivity and anatomical tracing. If functional connectivity does not reproduce these findings in detail, then do the RSFC methods as applied to monkeys have the precision and sensitivity necessary to validate data?

We apologize for initially omitting the view of the medial wall from our original macaque connectivity figure (original Figure S9) in order to focus on findings in M1. This figure (new

Extended Data Figure 9, shown below) has been remade to show that the foot area exhibits extensive functional connectivity to medial wall regions and the hand area shows smaller areas of connectivity. The mouth area exhibits very small regions of medial wall connectivity in individual macaques, but they are spatially inconsistent and do not emerge in group average macaque data. This may be because the orofacial-connected medial wall regions, as illustrated by Morecraft et al., 1996, NeuroReport and Tokuno et al., 1997, J Comparative Neurology, are very small and difficult to detect in lower-quality macaque fMRI.

Extended Data Fig. 9| Motor Cortex functional connectivity in non-human primates. Functional connectivity maps were seeded from dorsal anterior cingulate cortex (top row), as well as from a continuous line of points down anterior central sulcus (rows 2-4), in fMRI data from **a**, an individual macaque scanned for 77 minutes on a 10.5T MRI scanner; **b**, an individual macaque scanned for 53 minutes on a 3T scanner; and **c**, group-averaged data from eight macaques each scanned for 53 minutes on a 3T scanner. The dorsal anterior cingulate seed demonstrated connectivity to frontal, insular, and parietal regions homologous with the human CON, as well as with two regions in anterior central sulcus (maroon arrows). These central sulcus regions are thought to correspond to areas that project to internal organs^{10,53} and represent possible macaque homologues of the inter-effector regions. The central sulcus seeds demonstrated connectivity patterns corresponding to the known functional divisions between M1 regions representing the foot (second row), hand (third row), and face (bottom row).

The monkey data seems necessary to validate the whole concept of Mind-Body Network. This is because the adrenal and stomach data are derived from animal models. The adrenal data, in particular, has a major component from M1 (Dum et al., 2016). Thus, the middle inter-effector regions should be present in the precentral gyrus and central sulcus of monkeys.

Thank you for raising these important points. Unfortunately, macaque BOLD fMRI scanning techniques have not yet achieved the same data quality as in humans, and scanning cannot easily be performed on an extended basis to collect the same data quantities as in humans. Therefore, there is necessarily somewhat more noise in the macaque data, and necessarily more uncertainty in their interpretation. In the data shown in the original manuscript draft, seeds

placed in the central sulcus did not identify a strictly homologous network. Instead, the action network we identified in macaques appeared as an intermediate between the CON (Cingulo-Opercular Network) and the SCAN (Somato-Cognitive Action Network). In macaques, the lower overall SNR and larger voxel sizes (relative to the size of the brain) may be impeding clearer separation of inter-effector regions from adjacent effector-specific regions.

To deepen our search for inter-effector regions, we seeded regions of the medial frontal gyrus containing putative CON homologues. Using this seed, we observed strong connectivity with two regions in the central sulcus that approximated the locations of the top and middle inter-effector region observed in humans (see Extended Data Fig 9a, top).

To validate this finding, we obtained new macaque BOLD data from 8 additional animals. In individual animals (Extended Data Fig 9b, top) and in their group average (Extended Data Fig 9c, top), we replicated this pattern of connectivity from medial prefrontal cortex to two distinct regions in central sulcus. Thus, with additional macaque data and analyses, we were able to identify a connectivity pattern more closely homologous to that found in humans.

Even with the additional data and analyses, the SCAN nodes in macaque M1 do not appear to be as large as in humans; and further, we only identified two candidate homologue SCAN regions. Technical limitations of macaque fMRI are one potential explanation for these differences between the macaque and human SCAN. However, another potential explanation is that the SCAN may be evolutionarily expanded because it is important for coordinating complex actions relatively specific to humans, such as speech and/or tool use. For example, we have shown that the dual representation of larynx in humans is inside the mouth region (Fig. S4), but they are directly adjacent to the middle and inferior SCAN nodes. This could reflect the integration of breathing control with laryngeal control needed for complex speech. Further, we have shown that the middle SCAN node exhibits strong connectivity to extrastriate visual cortex (Extended Data Fig. 5; see response to next point below), which could reflect the complex reach/grasp hand-eye coordination in tool use. The revised manuscript reflects these additional insights.

Added text: Results

To link these neuroimaging findings to decades of detailed motor cortex mapping in non-human primates, we searched for inter-effector homologues in macaques using fMRI. Seeds placed in macaque M1 revealed foot, hand, and mouth effector-specific functional connectivity patterns consistent with those seen in humans³² (Extended Data Fig. 9, rows 2-4). In macaque fMRI, lower overall SNR (signal-to-noise ratio) and larger voxel sizes relative to the size of the brain can impede clear separation of the smaller inter-effectors from adjacent effector-specific regions. Therefore, we also seeded putative CON homologues in dACC, which revealed strong connectivity with lateral frontal cortex, insula, and supramarginal gyrus, similar to the human CON, and with two regions in anterior central sulcus that may be homologous to the superior and middle inter-effectors (Extended Data Fig. 9, row 1).

Added text: Discussion

The apparent relative expansion of SCAN regions in humans could suggest a role in complex actions specific to humans, such as coordinating breathing for speech, and integrating hand, body and eye movement for tool use.

3. Do the inter-effector regions really have the same functional roles (i.e., “an integrative system for implementing whole-body action plans, the Mind-Body Network (MBN)”)?

Thank you for raising this excellent point. We did not mean to suggest that the three SCAN regions in M1 have identical, redundant functional roles, but rather that they cooperate to conduct integrated processing. The most striking observation about the SCAN regions was their high similarity in all the data we examined, which stood in stark contrast to their clear and obvious differences from the foot, hand and mouth M1 regions. We wanted to first focus on that self-similarity, which might suggest coordination/integration as an important functional principle for SCAN organization.

The revised manuscript now more extensively identifies differentiating factors across the three SCAN nodes. We have added analyses describing connectivity differences between the SCAN regions in M1. For each region in each participant, we identified locations in the brain where FC was stronger to that node than to either of the other two SCAN nodes.

We observed that this node selectivity was relatively sparse across the cortex (Extended Data Fig. 5), and was not very consistent across subjects (Figure S2). However, for each SCAN node, there were small regions on the medial wall that exhibited at least some node selectivity that was consistent across subjects, as you hypothesized.

Most notably, the middle SCAN node consistently exhibited stronger FC to extrastriate visual cortex than either of the other two nodes, suggesting closer integration with processing visual information.

Added text: Results

Searching for differences between the three inter-effector regions revealed that the middle inter-effector consistently exhibited stronger functional connectivity to extrastriate visual cortex than either of the other two regions (Extended Data Fig 5; Fig S2 for all participants).

Added text: Discussion

Minor connectivity (Extended Data Fig. 5) and activation (Extended Data Fig. 7) differences between the superior, middle and inferior inter-effector regions likely reflect some degree of functional specialization within this integrated system. The middle region's relatively stronger connectivity to visual cortex, for example, suggest a potential role in hand-eye coordination during reach-and-grasp motions⁵⁶.

Extended Data Fig. 5| Differences in functional connectivity between inter-effector regions. Brain regions more strongly connected to the superior inter-effector region than to either of the other two (top row); relatively most strongly connected to the middle inter-effector region (middle row); and relatively most strongly connected to the inferior inter-effector region, in cortex (left), striatum, thalamus, and cerebellum (right). **a**, In at least 50% of individuals ($n = 7$) and **b**, group-averaged data from the Human Connectome Project (HCP; $n = 812$). Thresholds used are the same as in Fig. 2b. Note that central sulcus regions are masked as they exhibit large differences by definition. See Fig. S2 for all individual participants.

Fig. S2| Functional connectivity differences among inter-effector regions in individual participants. In seven individuals, brain regions more strongly connected to the superior inter-effector region (top row), middle inter-effector region (middle row), and inferior inter-effector region (bottom row) than to either of the other two, in cortex (left), striatum, thalamus, and cerebellum (right). Thresholds used are the same as in Fig. 2b. Note that central sulcus regions are masked as they exhibit large differences by definition. See Extended Data Fig. 5 for overlap across individuals.

The concept of Mind-Body network seems to be rooted in the role of autonomic control-

specifically the sympathetic nervous system output located in M1 and the premotor areas. However, the sympathetic system only seems to apply primarily to the top inter-effector region and not the middle and bottom sites. Doesn't this difference distinguish these 3 sites and suggest that they don't have the same functional roles?

With regard to the specific example raised here, we note that data from the brilliant study by Dum et al., 2016 suggest that two M1 regions are connected to the adrenal medulla (see Figure 2 from Motor, cognitive, and affective areas of the cerebral cortex influence the adrenal medulla Dum et al., 2016 paper), and these regions might overlap with the macaque homologues of the superior and middle inter-effector regions. Thus it could be that multiple SCAN nodes influence the adrenal medulla, though their functions may not be identical (as illustrated by a difference in their unilateral vs bilateral connectivity in Dum et al. 2016, Fig 3A vs 3C).

IMAGE REDACTED

Fig 3a from Dum et al., 2016 redacted as we cannot publish third party images.

Added text:

Notably, the spatial distribution of adrenal connectivity¹⁰ converges closely with the proposed inter-effector homologues.

Fig. 4a is entirely consistent with Fig. 3 and Penfield's interpretation of motor output in the

human (as is also Roux et al., 2020 stimulation based map). Why is a new map needed? Shouldn't the new map be more accurate?

Fig. 3, and particularly the details of the activation profiles shown in Fig. 3b and Extended Data Fig. 7, are inconsistent with Penfield's homunculus, which does not represent either the concentric activation profiles of each movement or the interconnected inter-effector system. While a schematic illustration can never fully represent the underlying data—a limitation which we now articulate more explicitly—we argue that Fig 4b (now slightly revised) is more accurate in multiple ways than the prior representation.

Text added to Figure 4 legend:

As with Penfield's original drawing, this diagram is intended to illustrate organizational principles, and must not be over-interpreted as a precise map.

Fig. 4b implies that the 3 inter-effector regions are the same, each representing the whole body (puppet figurine). However, Fig. S7 tells a completely different story. Although it is hard to place the inter-effector regions exactly on the map, the top inter-effector region has leg, abdominal and proximal arm muscles, the middle has proximal arm, maybe abdominal and upper face but no leg and the ventral inter-effector region has only lower face. (Placing a gray box for each inter-effector region would be helpful). How can these all be considered whole body representations as represented by the puppet figurine?

Thank you for raising this important point. We have now reworked original Fig. S7 (new Extended Data Fig. 7, shown below), to more comprehensively show all of the activation profiles across the whole range of M1.

Extended Data Fig. 7 | Primary motor cortex activation profiles for a movement task battery. In two participants (top, bottom), LOWESS curves were fit to the task activation profiles at each dorsal-ventral point in M1, for each separate movement (colored lines). Colored blocks (top) show the effector-specific foot (green), hand (cyan), and mouth (orange) areas of M1, as well as the inter-effector regions (maroon); dotted maroon lines show the boundaries between regions. The centers of effector-specific regions are characterized by strong activations for movements of the most distal body parts, and deactivations for all other movements. Inter-effector regions, by contrast, exhibited moderate activations for most movements.

This more complete visualization more clearly illustrates the key differences between the effector-specific and inter-effector regions. The centers of effector-specific regions are strongly activated by distal movement, but deactivated by other movements, indicating strong movement specificity. The inter-effector regions exhibit small to moderate positive activations for most movements. While there is some preference—i.e. the superior inter-effector activates most strongly to abdominal movement, while the middle one may activate more strongly to upper face movements—that preference is much smaller than the preference exhibited by the effector-specific regions (Fig 3d).

Added text:

Unlike effector-specific regions, the inter-effectors exhibited weak movement-specificity, with minimal activation differences between their preferred and non-preferred movements (Fig. 3d) and at least some activation observed across most movements (Extended Data Fig. 7).

Notably, this finding is strongly consistent with a recent preprint by Jensen et al., 2022, biorxiv (<https://www.biorxiv.org/content/10.1101/2022.11.20.517292v3>), following our preprint. This work used intracortical recordings in awake humans to demonstrate that two regions of precentral gyrus, which Jensen et al. believe to correspond to the superior inter-effector regions (labeled RMA in Jensen et al Figure 4, please see Fig 4 in <https://www.biorxiv.org/content/10.1101/2022.11.20.517292v3>), exhibit nonselective activity to voluntary hand, foot, and tongue movements. We now discuss these new findings.

Fig 4 from Jensen et al 2022 redacted as we cannot publish third party images.

Added text:

Following our identification of the SCAN, human depth electrode recordings verified that a portion of M1 between the foot and hand effector regions (superior SCAN) is active during foot, hand and mouth movements, strongly supporting its integrative, whole-body function⁵⁸.

Are the results in 2 individuals sufficient evidence to draw a whole new map?

The existence of the SCAN was demonstrated in all examined individuals at least 11 months old (10 separate participants) and in large group averages constituting thousands of subjects. The concentric organization of M1, which is not represented in Penfield's illustration, has extensive

support from non-human primate studies.

Huber et al., 2018, 2020 has provided high fidelity evidence that there are at least 2 hand representations in M1. According to Huber, a more accurate representation of the hand would have 2 hands, joined at the 5th digit with a larger hand, thumb laterally and a smaller hand, thumb medially.

We agree that Huber's work shows how the hand and finger representations are likely also mirrored (indeed, one participant shows a hint of this in Fig 3b). We have revised the illustration to convey a mirrored hand organization (Fig 4b), and we now discuss this point.

Added text:

It has been suggested that this concentric organization extends to the ordering of fingers within the hand representation⁴⁹.

According to Figs. 3 and S7, the medial arm should be more robust and the lateral arm thinner.

We agree this is likely to be correct. Figure 4's legend now includes a caution against taking the illustration too literally.

Text added to Figure 4 legend:

As with Penfield's original drawing, this diagram is intended to illustrate organizational principles, and must not be over-interpreted as a precise map.

What evidence suggests that the face has a symmetrical, nested organization? Neither the maps in Bouchard et al., 2013 or in Figs. 3 and S7 seem to support this interpretation.

The new results from a vocalization task (Fig S4) recapitulate prior findings showing a dual laryngeal motor representation (Eichert et al., 2020, Cerebral Cortex; Belyk et al., 2021, Philos Trans B; Castellucci et al., 2022, Nature) and further support the observed mirrored symmetrical representations within the mouth/face effector-specific region.

4. There is a general lack of coordinates to identify regions outside of inter-effector regions.

Fig. 2- What are the coordinates of the preSMA and dACC for each of the 3 inter-effector areas?

Thank you for this suggestion. We now provide the coordinates for the full SCAN, including the pre-SMA and dACC locations in Table 1.

The hypothesis implies that they are quite similar. Is that true? Is there any criteria that would make this distinction unambiguously?

As described above, the three SCAN nodes do exhibit minor connectivity differences on the medial wall, suggesting a possible differential connective topography (Extended Data Fig. 5). However, these differences are small enough and inconsistent enough across participants (Fig.

S2) that we do not yet feel comfortable making strong claims about them. Further work is needed to follow up on these important questions.

The preSMA could actually be the SMA.

Editorial Note: the author's response has been redacted at their request to maintain the confidentiality of unpublished data.

The view that all three inter-effector regions mediate whole body movements depends on their greater connectivity with the cingulate-opercular network (CON) than the connectivity of the leg, arm and face effector regions with CON. This difference is only significant at the $p < 0.05$ level.

We apologize for the lack of clarity here. The specificity of SCAN functional connectivity to CON, compared to the connectivity of effector-specific regions to CON, was very high (see Extended Data Figure 5a, reproduced below), with SCAN in all subjects exhibiting stronger connectivity to CON than any effector-specific region in every single subject, and all three separate SCAN vs foot/hand/mouth paired t-tests at $p < 0.01$.

Fig. S5a| In each individual participant, measures derived from each of the foot, hand, mouth, and inter-effector motor regions. Colored lines connect the same participant's inter-effector and effector-specific regions for ease of comparison. **a.** Functional connectivity strength between M1 region and individual-specific Cingulo-Opercular Network.

In a separate set of tests, we compared SCAN connectivity to CON against SCAN connectivity to every other brain network (10 separate paired t-tests). Connectivity to CON was statistically stronger than connectivity to the other networks in every test, with all p values surviving FDR correction for multiple comparisons. We have now remade Figure 2b (reproduced below) to better represent the tests conducted.

Fig. 2b| Functional connectivity and cortical thickness of the motor cortex inter-effector motif. Connectivity was calculated between every network and both inter-effector and effector-specific M1 regions. The plot shows the smallest difference between inter-effector connectivity and any effector-specific connectivity (standard error bars across subjects). This difference was larger for CON than for any other network (*: $P < 0.05$; **: $P < 0.01$).

Ideally, this difference would be evident for each inter-effector region versus each effector region.

As you suggest, we tested the CON connectivity of each of the three SCAN regions against that of each foot/hand/mouth effector-specific region. Across all 9 comparisons for all 7 subjects (63 total value comparisons), SCAN regions had stronger CON connectivity than effector-specific regions in 62 of 63 total comparisons (see Figure R4 below).

Across 9 paired t-tests across subjects, all except one were significant after FDR correction, with p values ranging from 0.01 to 0.0002. One test—the superior SCAN vs Mouth comparison—was only significant at trend level ($p = 0.09$).

Figure R4] Functional connectivity to CON of each separate SCAN and effector-specific region. In each individual participant, functional connectivity strength from each of the foot, hand, mouth, and three separate SCAN regions to the individual-specific Cingular-Opercular network is plotted. Colored lines connect the same participant's inter-effector and effector-specific regions for ease of comparison.

The reason for this analysis is to figure out what the inter-effector region is doing. What is CON's functional specialty?

We first described the CON as a network critical for top-down cognitive control and involved in establishing and maintaining task goals, including task control initiation, task maintenance, and task error response (Dosenbach et al., 2006, 2007, 2008; Petersen & Posner, 2012; Power & Petersen, 2013), across many domains (Gratton et al., 2016; Gratton, Sun, et al., 2018). The original defining feature was that the CON is active whenever human participants engage in a specific task, independent of the nature of the task. However, we now believe that these past analyses only examining one or a few selected task fMRI contrasts inaccurately focused on very specific roles for the CON. Indeed, other lines of work argue that the CON plays a role in processing painful stimuli. The dorsal anterior cingulate and the anterior insula are commonly reported as brain regions most active during application of painful stimuli (Pinto et al., 2022; Reddan & Wager, 2018; Wager et al., 2013). The CON also seems to play a role in the arousal changes that occur at the beginning and end of specific activities (Sadaghiani and D'Esposito, 2015). Finally, some CON regions enable top-down motor control (Lu et al., 1994). In non-human primates, three prefrontal areas critical for motor planning have been identified in the cingulate sulcus; their homologues in humans are consistent with established CON topography (Picard & Strick, 1996, 2001). And in humans, our own neuroimaging work has demonstrated the CON's role in motor aspects of action control (Newbold et al., 2020, 2021).

Together, we posit that the CON plans, initiates, and maintains actions, including translating those action to motor plans and modifying planned actions based on error or painful feedback. The CON encompasses many if not all functions required for successfully executing an action, ranging from abstract planning, to arousal changes, to global allostatic physiological changes, to whole-body motor commands, all the way to processing of pain and visceral feedback.

Whole concept of CON connectivity is somewhat spurious since it is mainly the SMA and cingulate motor areas that provide the bulk of the connectivity with inter-effector regions.

The SCAN is selectively functionally connected to CON midline region, which are thought to be analogous to the pre-SMA and rostral cingulate zone in NHPs. The effector-specific regions in contrast, are functionally connected to more posterior midline regions that also show some effector specificity in task fMRI, albeit much weaker than the effector-specific M1 regions.

While SCAN connectivity to the medial CON regions is certainly the strongest, there is FC to other CON regions, including anterior prefrontal cortex, posterior superior temporal sulcus, and middle insula (see Fig R5 below).

Figure R5j CON regions connected to the SCAN. In participant 2, **a**, Functional connectivity seeded from the anterior prefrontal cortex (aPFC). **b**, Functional connectivity seeded from the posterior superior temporal sulcus (pSTS). **c**, Functional connectivity seeded from the middle insula (seed is located under the fold of the inferior frontal gyrus; see arrow). In each case, SCAN regions exhibited functional connectivity with the seed.

Line 177- extra "c" in middle of description of "b"

Thank you for noticing. This has been corrected.

line 288- The term "posterior bank of the precentral gyrus" is confusing since many know this region as the anterior bank of the central sulcus.

Thank you. As described above, this figure has been modified and this text has been removed.

Reviewer Reports on the First Revision:

Referee #1 (Remarks to the Author):

I have read the responses to my comments, as well as those to the other referees. The authors have done a laudatory job in their responses, and I particularly appreciate the change in nomenclature (to SCAN), prompted by one of the other referees. I believe this paper will have a strong and positive impact on neuroscience and neurology, and I congratulate the authors.

Referee #2 (Remarks to the Author):

All my concerns were addressed. I find the paper excellent in its present form.

Referee #3 (Remarks to the Author):

Overall, the authors have done an outstanding job of responding to my prior comments. They did additional work that has enhanced the significance of this study and clarified some issues relating to the monkey. I will not repeat my prior comments about the importance and validity of the data.

This study is about cortical maps, particularly those involving the motor areas in the frontal lobe of humans and macaque monkeys. There are many kinds of maps with overlapping territories of different scale and all fundamentally depend on the technique used to produce them. Maps are all about location, the more specific the better. Interpretation of a new map should aim to uncover similarities and to resolve conflicts with prior maps. The heavily smoothed version of the brain used in this study makes visual comparison with prior work difficult. As a consequence, I will rely on the coordinates of peak activations to frame many of my comments.

A key question in the present study is the relationship of the inter-effector regions with prior motor maps of the medial wall of the hemisphere. The present study suggests that the inter-effector regions are connected with the preSMA and dACC on the medial wall. However, the coordinates (Table S1) given for the preSMA are located in the SMA (e.g., Picard and Strick, 1996; Strother et al., 2012; Amiez and Petrides, 2014; see Mayka, Corcos, Leurgans, Vaillancourt, 2006 for meta-analysis). Likewise, the coordinates given for the dACC are more consistent with the caudal cingulate zone (CCZ) or possibly the most posterior portion of the posterior rostral cingulate zone (RCZp)(Picard and Strick, 1996; Amiez and Petrides, 2014). The dACC coordinates (Table S1) are 10mm ventral and 10mm posterior to those that form the basis for identifying the CON (Dosenbach et al., 2006; - 1,10,46). Are the SMA and CCZ part of the CON? Does changing the identification of the medial wall areas change the interpretation of the function of the inter-effector regions and its relationship to CON?

From the data presented, I cannot answer these questions. The concept of sympathetic integration (adrenal projections of Dum et al., 2016) between medial wall areas and inter-effector regions does

not seem to depend on the identity of the medial wall areas. The SMA and CMA_d (monkey equivalent of the CCZ) actually have the most adrenal projections of the areas on the medial wall. Likewise, nociceptive information also reaches the CMA_d, CMA_v and CMA_r in the monkey (Dum et al., 2009).

The connectational data is also consistent with the SMA being connected to the inter-effector regions. In monkeys, the SMA (F3) of monkeys receives projections from M1 whereas the pre-SMA (F6) does not (Luppino et al., JCN 1993; Morecraft et al., 2012). The preSMA is not considered to be a premotor area precisely because it lacks both projections to M1 and the spinal cord in monkeys. The connections shown in Figs. 3Ra,b are consistent with those of the SMA. The SMA receives input from the caudal cingulate sulcus but the pre-SMA receives little or no projections from there (Luppino et al., 1993; Morecraft et al 2012).

The dACC is stated to be equivalent to the rostral cingulate zone, which has an anterior and a posterior region (Picard and Strick, 1996). The term dACC is vague and confusing. This term refers to different regions in different studies but also multiple regions within the same study (Vogt, 2016; Cole et al., TINS 2009; Shackman et al., 2011; Heilbronner and Hayden, 2016). The term dACC probably should be retired.

Extended data Fig. 7- For P1, the tongue has the second highest peak in the top inter-effector region. How does this make any sense? There is no evidence that cortical output neurons are located here and there is not evidence in the monkey that the orofacial region is connected directly to the trunk/top inter-effector region. Why is the fMRI activation so non-specific? Is this blurring of localization a function of the task or is it a reflection of some higher order network among widely separated muscles? A similar question could be raised about the abdominal activation seen in the bottom inter-effector region.

At what Z-score do the task activations become significant? Would a horizontal line at the first significant Z-score be useful?

I still don't understand the justification for using the marionette symbol at all three inter-effector regions in Fig. 4. Fig. ED7 shows that each inter-effector region has a different mix of body part representations. Only the top inter-effector region has substantial leg and shoulder representation (a complete marionette) but the lateral are dominated by orofacial muscles (orofacial dominance). Fig. S2 illustrates the same theme, each inter-effector region connects with a different location on the medial wall. Penfield's map has been criticized for glossing over the details. Although each inter-effector region is connected to CON, each has a distinct flavor with the lateral two showing little presence of arm and leg.

Fig. S4- Thank you for testing the possibility that voice related activations accounted for the inter-effector regions. This figure shows that the voice related activations are within the mouth RSFC region. Does the more lateral voice activation correspond to the vLMC of Belyk et al., 2021? Belyk placed this laryngeal area in Brodmann's area 43 whereas Eichert et al., 2020 places it in Brodmann's area 6. Since the most lateral inter-effector zone is lateral to the vLMC, what evidence suggests that it is still in area 4, i.e. primary motor cortex?

Fig. S6 has beautiful maps. Most important, why is the inter-effector network limited to the anterior bank of the central sulcus without any presence on the medial wall? It seems that if the inter-effector large scale network is really important for initiation of “executive action and physiological control, arousal, and processing of errors and pain”, it should appear on the medial wall.

Extended Data Fig. 8- Why is the CON network in P1 different from the CON network for P1 in Fig. S6? Why is the location of the hand on the medial wall different for both P1 & P2 in ED Fig. 8 from the location of the hand in Fig. S6?

Extended Data Fig. 9- Where exactly was the seed in the dorsal anterior cingulate of the monkey? How was it selected? It does seem likely that it was in the CMAr (Morecraft et al. 2012). He reported that an injection into the CMAr (ventral bank of the cingulate sulcus) at the level of the preSMA (case 6) had connections from both medially and laterally in M1. These could correspond to connections with the adrenal region in the CMAr (Dum et al., 2016). Alternatively, the medial region in M1 could represent trunk muscles and the lateral location the cricothyroid muscle which has representations in the CMAr and at a similar location to the lateral one in M1 (ED Fig. 9) (Cerkevich et al., 2022). The location of the seed relative to the genu of the arcuate sulcus would be useful for anyone wanting to reproduce the result.

This seed in the CMAr does not seem to be equivalent to the dACC of the present study. As detailed above, the dACC coordinates appear more consistent with the CMAd of monkeys.

Specific Comments:

lines 58-61- These inter-effector regions exhibit decreased cortical thickness and strong functional connectivity to each other, and to prefrontal, insular, and subcortical regions of the Cingulo-Opercular network (CON), critical for executive action⁵ and physiological control⁶, arousal⁷, and processing of errors⁸ and pain⁹.

As written, this sentence seems to indicate that the inter-effector regions have strong connections with prefrontal and insular cortex but not the CON, only its subcortical regions.

Author Rebuttals to First Revision:

We appreciate Reviewer 3's additional helpful comments and suggestions, which have further improved the manuscript.

Below, we respond to the remaining Reviewer comments and suggestions (Reviewer comments in black; our responses in blue), and we reproduce all resulting new text and Figures included in the manuscript.

Referee #1 (Remarks to the Author):

I have read the responses to my comments, as well as those to the other referees. The authors have done a laudatory job in their responses, and I particularly appreciate the change in nomenclature (to SCAN), prompted by one of the other referees. I believe this paper will have a strong and positive impact on neuroscience and neurology, and I congratulate the authors.

Referee #2 (Remarks to the Author):

All my concerns were addressed. I find the paper excellent in its present form.

We thank Reviewers 1 and 2 for their careful consideration and positive assessment of our work.

Referee #3 (Remarks to the Author):

Overall, the authors have done an outstanding job of responding to my prior comments. They did additional work that has enhanced the significance of this study and clarified some issues relating to the monkey. I will not repeat my prior comments about the importance and validity of the data.

A key question in the present study is the relationship of the inter-effector regions with prior motor maps of the medial wall of the hemisphere. The present study suggests that the inter-effector regions are connected with the preSMA and dACC on the medial wall. However, the coordinates (Table S1) given for the preSMA are located in the SMA (e.g., Picard and Strick, 1996; Strother et al., 2012; Amiez and Petrides, 2014; see Mayka, Corcos, Leurgans, Vaillancourt, 2006 for meta-analysis).

The connectational data is also consistent with the SMA being connected to the inter-effector regions. In monkeys, the SMA (F3) of monkeys receives projections from M1 whereas the pre-SMA (F6) does not (Luppino et al., JCN 1993; Morecraft et al., 2012). The preSMA is not considered to be a premotor area precisely because it lacks both projections to M1 and the spinal cord in monkeys. The connections shown in Figs. 3Ra,b are consistent with those of the

SMA. The SMA receives input from the caudal cingulate sulcus but the pre-SMA receives little or no projections from there (Luppino et al., 1993; Morecraft et al 2012).

The concept of sympathetic integration (adrenal projections of Dum et al., 2016) between medial wall areas and inter-effector regions does not seem to depend on the identity of the medial wall areas. The SMA and CMA_d (monkey equivalent of the CCZ) actually have the most adrenal projections of the areas on the medial wall. Likewise, nociceptive information also reaches the CMA_d, CMA_v and CMA_r in the monkey (Dum et al., 2009).

Thank you for pointing this out. Based on its y-coordinates, the more posterior of the two strongly SCAN connected regions on the midline is indeed more consistent with classic SMA than with preSMA. We also agree the hypothesized link to sympathetic integration is more conceptually coherent if this region is SMA. Thus, we have updated this nomenclature from preSMA to SMA throughout all the submitted materials.

Likewise, the coordinates given for the dACC are more consistent with the caudal cingulate zone (CCZ) or possibly the most posterior portion of the posterior rostral cingulate zone (RCZ_p)(Picard and Strick, 1996; Amiez and Petrides, 2014).

The dACC is stated to be equivalent to the rostral cingulate zone, which has an anterior and a posterior region (Picard and Strick, 1996). The term dACC is vague and confusing. This term refers to different regions in different studies but also multiple regions within the same study (Vogt, 2016; Cole et al., TINS 2009; Shackman et al., 2011; Heilbronner and Hayden, 2016). The term dACC probably should be retired.

Our use of the term “dACC” comes from human fMRI research, which applies this broad label to a relatively wider swath of dorsomedial prefrontal cortex, often centered on the RCZ but also potentially including the CCZ and even the preSMA. The cingulate zone terminology is certainly much more precise; however, it is used less commonly and is not as well known within the human fMRI field. We fully agree based on apparent function and location that the region in question region likely corresponds to the CCZ. Thus, for increased precision, the revised manuscript now articulates that this is likely the CCZ subregion of the dACC.

Added text to Results:

Connectivity was very strong with SMA and a region in dACC (caudal cingulate zone) (Picard and Strick 2001).

The dACC coordinates (Table S1) are 10mm ventral and 10mm posterior to those that form the basis for identifying the CON (Dosenbach et al., 2006; -1,10,46). Are the SMA and CCZ part of the CON? Does changing the identification of the medial wall areas change the interpretation of the function of the inter-effector regions and its relationship to CON?

It is important to note that the CON in the medial wall is not restricted to a single region, but rather extends across multiple contiguous medial wall areas, likely including most of the dACC (at least CCZ plus RCZp) as well as the SMA. Our dACC coordinate from 2006 (-1,10,46) was based on identifying a variety of fMRI responses related to top-down control, most commonly error responses, within group-averaged fMRI data. While this region is certainly in CON, and is likely the CON medial wall region most engaged by top-down cognitive control, it is not the only CON region in the medial wall. The SCAN-connected dACC region is separate and more posterior, but still clearly within the borders of CON.

Extended data Fig. 7- For P1, the tongue has the second highest peak in the top inter-effector region. How does this make any sense? There is no evidence that cortical output neurons are located here and there is not evidence in the monkey that the orofacial region is connected directly to the trunk/top inter-effector region. Why is the fMRI activation so non-specific? Is this blurring of localization a function of the task or is it a reflection of some higher order network among widely separated muscles? A similar question could be raised about the abdominal activation seen in the bottom inter-effector region.

We agree that this finding is surprising. Importantly, these human fMRI data have recently been validated by human depth electrode recording studies, one of which we highlighted in our prior response.

(Jensen et al., BioRxiv, 2022, Fig 4). See Fig 4 in the BioRxiv preprint here: <https://www.biorxiv.org/content/10.1101/2022.11.20.517292v3>.

Fig 4 from Jensen et al 2022 was redacted as we cannot publish unpublished findings.

Recent human BCI work also identified regions in M1 carrying information about a wide range of body movements. For example, face movement related information has been reported in the 'hand knob' (Willett et al., Cell, 2020), while speech has been successfully decoded from a region partially overlapping with middle SCAN node (Silva et al, J Neurosci, 2022), as well as from another region partially overlapping the inferior SCAN node (Willett et al., BioRxiv, 2023). Together, several recent lines of research suggest that higher-order action plans involving the mouth show activity in three distinct M1 regions. These regions also seem to be active when moving other body parts, consistent with our proposal for SCAN's whole-body role. Therefore, we feel that your interpretation of "a higher order network among widely separated muscles" is fairly close to our interpretation of these results.

We agree that the anatomical pathways by which this integration occurs are as yet unclear. We do consider it possible that it may be enabled by indirect connections via higher-order regions.

At what Z-score do the task activations become significant? Would a horizontal line at the first significant Z-score be useful?

This is an excellent question. We chose not to display significance thresholds in Fig. ED7 because of work showing that statistical significance in fMRI is primarily driven by the amount and quality of the data. For example, Gonzalez-Castillo et al. (PNAS, 2012), showed how 95% of the brain can pass statistical significance for a simple visual task, if very large amounts of high-quality data are available. Our individual-specific fMRI approach does have large amounts of high-quality data per participant; thus, much of the brain could be considered statistically activated for each of the contrasts. Indeed, most of the relevant activations, which range from $Z \sim 5$ to $Z \sim 35$ (see y-axis of Fig. ED7), are very far above normal significance thresholds of $Z=1.96$ ($p=0.05$), $Z=3.3$ ($p=0.001$), or even $Z=4.22$ ($p=0.00001$). Findings such as this have suggested that fMRI results may be better interpreted as continuous activation magnitudes, rather than as binary active/not active regions determined by statistical significance. Therefore, visualizing all the data seems to be the most rigorous approach.

Extended Data Fig. 7 | Primary motor cortex activation profiles for a movement task battery. In two participants (top, bottom), LOWESS curves were fit to the task activation profiles at each dorsal-ventral point in M1, for each separate movement (colored lines). Colored blocks (top) show the effector-specific foot (green), hand (cyan), and mouth (orange) areas of M1, as well as the inter-effector regions (maroon); dotted maroon lines show the boundaries between regions. The centers of effector-specific regions are characterized by strong activations for movements of the most distal body parts, and deactivations for all other movements. Inter-effector regions, by contrast, exhibited moderate activations for most movements.

I still don't understand the justification for using the marionette symbol at all three inter-effector regions in Fig. 4. Fig. ED7 shows that each inter-effector region has a different mix of body part representations. Only the top inter-effector region has substantial leg and shoulder representation (a complete marionette) but the lateral are dominated by orofacial muscles (orofacial dominance). Fig. S2 illustrates the same theme, each inter-effector region connects with a different location on the medial wall. Penfield's map has been criticized for glossing over the details. Although each inter-effector region is connected to CON, each has a distinct flavor with the lateral two showing little presence of arm and leg.

In the fMRI data, it is clear that there is greatly reduced movement selectivity in the SCAN nodes relative to effector-specific nodes (validated by human depth electrode recordings in Jensen et al. (BioRxiv, 2022), reproduced above). Therefore, it seems that SCAN regions in M1 are involved in a wide range of body movements from face to foot, perhaps to coordinate/integrate whole-body movement. We thus argue that Penfield's homunculus should be updated because it lacks the SCAN regions, which have very little movement specificity and appear to be fundamentally different from the effector-specific regions in many ways. The marionette icon/symbol was chosen to represent this coordination/integration because it already carries a connotation of integrated action control.

However, the exact significance of the rank order of activation strengths for different movements/body parts (Fig. ED7) is not entirely clear, as that order varies somewhat across nodes and across participants. As such, we hesitate to strongly interpret this ordering in the current work. Towards that end, we have added a warning against overinterpreting the schematic in Fig. 4b to the revised manuscript.

Text added to Figure 4 legend:

As with Penfield's original drawing, this diagram is intended to illustrate organizational principles, and must not be over-interpreted as a precise map.

Fig. S4- Thank you for testing the possibility that voice related activations accounted for the inter-effector regions. This figure shows that the voice related activations are within the mouth RSFC region. Does the more lateral voice activation correspond to the vLMC of Belyk et al., 2021? Belyk placed this laryngeal area in Brodmann's area 43 whereas Eichert et al., 2020 places it in Brodmann's area 6. Since the most lateral inter-effector zone is lateral to the vLMC, what evidence suggests that it is still in area 4, i.e. primary motor cortex?

Great questions. Comparisons with BA maps, which are not without their issues, suggest that the inferior-most SCAN region is partly in BA 4 and partly ventral to BA 4. It does not appear to be in BA 6. It's certainly not in BA 43, which is posterior. In this manuscript we avoid localization to BAs at this level of precision, but future work that enables us to do so with confidence will be very impactful.

Fig. S6 has beautiful maps. Most important, why is the inter-effector network limited to the anterior bank of the central sulcus without any presence on the medial wall? It seems that if the inter-effector large scale network is really important for initiation of "executive action and physiological control, arousal, and processing of errors and pain", it should appear on the medial wall.

The previous Fig. S6 (now Fig. S7 in revised Supplement; see below) was not the correct one, as you pointed out (next comment). Thank you so much for noticing that. The corrected Fig. S7 has been attached to the revised Supplement. Nonetheless, Fig. S7 is not the most optimal one to look for strong functional connectivity to the inter-effectors along the midline. Fig. 2a shows the strong functional connections of the M1 inter-effector regions to the SMA and dACC(/CCZ) on the medial wall better than Fig. S7. Infomap, like other network detection methods, forces each vertex/voxel to only be assigned to a single network. The SMA and dACC

nodes are strongly connected to SCAN (see Fig. 2a), but more strongly connected to the rest of the CON, which is why they are assigned to the CON in Fig. S7.

Extended Data Fig. 8- Why is the CON network in P1 different from the CON network for P1 in Fig. S6? Why is the location of the hand on the medial wall different for both P1 & P2 in ED Fig. 8 from the location of the hand in Fig. S6?

Thank you for catching this! Fig. S6 (now Fig. S7) did not show the correct network maps. We have corrected this.

Fig. S7| Individual-specific large-scale brain networks for repeatedly sampled adult participants. To identify classic large-scale networks in each participant, we ran the Infomap algorithm on matrices thresholded at a series of denser thresholds (ranging from 0.1% to 5%), and identified individual-specific networks corresponding to Somatomotor, Inter-effector, Default, Medial and Lateral Visual, Cingulo-Opercular, Fronto-Parietal, Dorsal Attention, Language, Saliency, Parietal Memory, and Contextual Association networks, following procedures described in ³².

Extended Data Fig. 9- Where exactly was the seed in the dorsal anterior cingulate of the monkey? How was it selected? It does seem likely that it was in the CMAR (Morecraft et al. 2012). He reported that an injection into the CMAR (ventral bank of the cingulate sulcus) at the level of the preSMA (case 6) had connections from both medially and laterally in M1. These could correspond to connections with the adrenal region in the CMAR (Dum et al., 2016). Alternatively, the medial region in M1 could represent trunk muscles and the lateral location the

cricothyroid muscle which has representations in the CMAr and at a similar location to the lateral one in M1 (ED Fig. 9) (Cerkevich et al., 2022). The location of the seed relative to the genu of the arcuate sulcus would be useful for anyone wanting to reproduce the result.

A series of midline seeds were selected running along the cingulate gyrus from the posterior portion of CMA_d to the middle portion of CMA_r. The seed that was relatively most consistent with a macaque homologue of the SCAN and CON was selected for display. The revised manuscript now includes a more comprehensive description of this procedure (see new Fig. S8 below).

The location of the midline seed connected with a potential macaque SCAN/CON homologue varied slightly from monkey to monkey but approximately seemed to lay in the anterior portion of CMA_d ($y = -1$ to -2), just posterior to the genu of the arcuate sulcus. The revised Supplement now also includes a table of coordinates for the macaque data (Table S2).

Text added to Methods:

We placed connectivity seeds continuously along area 4p in the left hemisphere of each macaque, as well as continuously running from the dorsal cingulate motor area to the rostral cingulate motor area in the dorsal anterior cingulate cortex (area 24).

Fig. S8| Functional connectivity in the cingulate gyrus of the macaque. a, Six seeds were placed along the medial prefrontal cortex in posterior dorsal cingulate motor area (CMAd) and rostral cingulate motor area (CMAr). Dotted line denotes the anterior-posterior level of the genu of the arcuate sulcus, taken to be the dividing line between CMAd and CMAr (here at $y=0$). **b**, Functional connectivity from each seed in group-averaged macaque data. Distributed connectivity within M1 could be observed seeded from Seed 4 in anterior CMAd.

This seed in the CMAr does not seem to be equivalent to the dACC of the present study. As detailed above, the dACC coordinates appear more consistent with the CMAd of monkeys.

As discussed above, the human “dACC” is relatively expansive and likely includes all of the cingulate motor zones, equivalent to both the CMAr and CMAd of the macaque. That being

said, the macaque seed described above in CMAAd is a plausible homologue to the human area in posterior dACC/CCZ.

lines 58-61- These inter-effector regions exhibit decreased cortical thickness and strong functional connectivity to each other, and to prefrontal, insular, and subcortical regions of the Cingulo-Opercular network (CON), critical for executive action⁵ and physiological control⁶, arousal⁷, and processing of errors⁸ and pain⁹.

As written, this sentence seems to indicate that the inter-effector regions have strong connections with prefrontal and insular cortex but not the CON, only its subcortical regions.

Thank you very much for catching that. The sentence has been shortened and made clearer.

Revised text:

“These inter-effector regions exhibit decreased cortical thickness and strong functional connectivity to each other, and to the Cingulo-Opercular network (CON), critical for executive action⁵ and physiological control⁶, arousal⁷, and processing of errors⁸ and pain⁹.”

Reviewer Reports on the Second Revision:

Referee #3 (Remarks to the Author):

The authors have done an outstanding job of responding to my prior comments. I appreciate the new figure S8 which provides a much better explanation of their procedures. Thank you for your patience with my many questions. I have no more concerns.